# TASK DESCRIPTORS HELP TRANSFORMERS LEARN LINEAR MODELS IN-CONTEXT

**Ruomin Huang**
Duke University
ruomin.huang@duke.edu

**Rong Ge**
Duke University
rongge@cs.duke.edu

## ABSTRACT

Large language models (LLMs) exhibit strong in-context learning (ICL) ability, which allows the model to make predictions on new examples based on the given prompt. Recently, a line of research (Von Oswald et al., 2023; Akyürek et al., 2023; Ahn et al., 2023; Mahankali et al., 2023; Zhang et al., 2024a; Vladymyrov et al., 2024) considered ICL for a simple linear regression setting and showed that the forward pass of Transformers is simulating some variants of gradient descent (GD) algorithms on the in-context examples. In practice, the input prompt usually contains a task descriptor in addition to in-context examples. We investigate how the task description helps ICL in the linear regression setting. Consider a simple setting where the task descriptor describes the mean of input in linear regression. Our results show that gradient flow converges to a global minimum for a linear Transformer. At the global minimum, the Transformer learns to use the task descriptor effectively to improve its performance. Empirically, we verify our results by showing that the weights converge to the predicted global minimum and Transformers indeed perform better with task descriptors.

## 1 INTRODUCTION

Transformer-based large language models (LLMs) have exhibited surprising abilities. One of their most remarkable abilities is to perform well even on tasks that they are not explicitly trained on. This is partially attributed to in-context learning (ICL) mechanism, where in-context examples are provided to significantly improve the prediction of LLM on a new query input (Brown et al., 2020). For langauge models, a classical example of ICL adapted from Brown et al. (2020) is illustrated as in Figure 1, where we instruct the language model to translate English to French by providing examples.

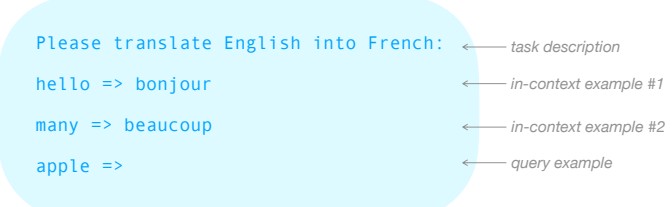

Figure 1: An input with both task descriptions and in-context examples.

To understand ICL mechanism without getting into the difficulty of language modeling, Garg et al. (2022) investigated the simpler problem of learning a function class $\mathcal{H}$ in-context. In this setup, the Transformer is given a sequence $S = (x_1, h(x_1), \ldots, x_n, h(x_n), x_{\text{query}})$. Here $(x_i, h(x_i))(i = 1, 2, \ldots, n)$ are $n$ in-context examples where $h$ is a function in class $\mathcal{H}$, and $x_{\text{query}}$ is new input (the query) that we want to evaluate $h$ on. The goal of the Transformer is to output a prediction that is equal to $h(x_{\text{query}})$. Garg et al. (2022) found that Transformers pretrained to perform this task can successfully in-context learn many concept classes. Later works (Von Oswald et al., 2023; Akyürek

et al., 2023; Ahn et al., 2023; Mahankali et al., 2023; Zhang et al., 2024a;b; Wu et al., 2024) focused more on in-context learning linear models, where $h(x_i) = w^\top x_i$ for an unknown vector $w$ and the goal of in-context learning is to predict $y_{\text{query}} = w^\top x_{\text{query}}$.

These existing theoretical works focuses entirely on leveraging in-context examples. However, as we saw in Figure 1, in practice we often provide an additional *task descriptor* ("Translate English to French."). Such descriptors improve the ICL performance for language models (Brown et al., 2020). While the prefix before in-context examples could contain various information, this work focuses on descriptions containing *distributional* information about inputs. In this paper, we investigate whether such task descriptors can help in-context learning for linear models.

## 1.1 OUR RESULTS

We consider a new family of mean-varying linear regression problems. For these problems, the in-context examples $(x_1, y_1), (x_2, y_2), ..., (x_n, y_n)$ are generated based on two parameters $\mu$ and $w$. Here $w$ is the underlying linear relation and we have $y_i = w^\top x_i$. The vector $\mu$ specifies the *mean* of $x_i$'s, that is, $x_i$'s are sampled from a Gaussian $\mathcal{N}(\mu, I)$ with mean $\mu$. The mean $\mu$ will be given as the task descriptor in the main paper [1]. Given $\mu$ and in context examples $(x_i, y_i)$'s and a query $x_{\text{query}}$, the goal of the Transformer is to estimate $y_{\text{query}} = w^\top x_{\text{query}}$.

We first show that with a specific embedding of the input (see Section 2.1), a 1-layer linear self-attention (LSA) network can indeed leverage the task descriptor. We can formally construct the optimal parameters for 1-layer LSA network for any number of samples $n$. When $n$ is large (going to infinity), the linear Transformer uses the mean $\mu$ straight-forwardly to remove the mean of $x_i$'s, leading to a better solution than previous constructions without task descriptor. When $n$ is small, the optimal solution is more complicated (see Theorem 4.1). In this case, we show that the training loss function can be decomposed into two terms and 1-layer linear Transformers leverage its full capacity to minimize both of them.

For all values of $n$, we also prove that gradient flow training from a reasonable initialization is guaranteed to converge to the global optimal solution. Even though the training loss is nonconvex, we give a detailed characterization for the set of all global optimal solutions, and show that the training dynamics maintain useful invariants that help the algorithm avoid saddle points and converge to a unique global minimum.

Finally, we empirically verify our findings in Section 5. For 1-layer Transformers, we show that the weights indeed converge to the global optimal solution that we have constructed. We also consider different settings where the Transformer may have multiple layers, or use a different embedding. We empirically observe that Transformers with task descriptor always outperform Transformers without task descriptors in all these settings.

## 1.2 RELATED WORKS

**In-context learning for linear regression** Garg et al. (2022) investigated the function classes that Transformers can learn in-context, finding that they can learn various function classes, including linear functions. Later works Von Oswald et al. (2023); Ahn et al. (2023); Mahankali et al. (2023); Bai et al. (2023) proposed that, under certain parameters, one forward pass of a Transformer is equivalent to a single step of some variant of gradient descent on linear models. Specifically, Ahn et al. (2023) showed that Transformers learn to simulate preconditioned GD, which is optimal for one-layer linear Transformers. Vladymyrov et al. (2024) and Fu et al. (2023) demonstrated that preconditioned GD can serve as a second-order optimization algorithm. Zhang et al. (2024b) found that with an appended linear MLP layer, Transformers can learn an initialization for GD. Wu et al. (2024) explored the task complexity bounds for in-context learning in linear regression. Zhang et al. (2024a) investigated how Transformers can be trained for ICL by proving that one-layer linear Transformers with appropriate initialization will converge to the global minimum under gradient flow dynamics. Huang et al. (2023) showed that Transformers with softmax non-linearity can be trained to learn linear regression on distinct features and analyzed the training dynamics. Chen et al. (2024a) investigated the scenario of multiple-head Transformers and multiple-task linear regressions.

---

[1]Experiments in Appendix B.5 explore one-hot encoding as an alternative form of task descriptors.

**Task descriptor**   To the best of our knowledge, there is a lack of theoretical studies on how transformers make use of task descriptors. Many empirical studies demonstrate their effectiveness. For example, adding a token that indicates the domain from which the data originates helps LLMs learn from context more efficiently (Allen-Zhu & Li, 2024). Brown et al. (2020); Honovich et al. (2023) found that incorporating natural language task descriptions helps GPT models improve in-context learning.

**Other in-context learning works**   There is a line of works on in-context learning with latent variable models within a Bayesian framework (Xie et al., 2022; Zhang et al., 2023; Wang et al., 2023; Jiang, 2023). Another line of works focus on mechanisms of induction heads. Olsson et al. (2022) proposed that two-layer Transformers can be induction heads, which complete the prompt $[A, B, A, B, \cdots, A] \to B$. Some generalized versions of induction heads were later studied: Nichani et al. (2024) explored the mechanism and training dynamics of Transformers learning causal graphs in-context, and Chen et al. (2024b) analyzed the setting of in-context learning $n$-gram Markov chains. There are other works showing Transformers can in-context learn various algorithms and tasks, such as sparse token selection (Wang et al., 2024), CFGs (Yao et al., 2021; Allen-Zhu & Li, 2023; Zhao et al., 2023), discriminative scanning (Tian et al., 2023) and Transformers (Panigrahi et al., 2024).

## 2   PRELIMINARIES

In this section we first introduce basic notations, then we describe the mean-varying linear regression problem in Section 2.1. Details about linear self-attention architecture and training are given in Section 2.2. Finally in Section 2.3 we briefly review how previous work understood ICL for linear regression as doing (preconditioned) gradient descent.

**Notations**   We lowercase letters to denote variables and vectors and uppercase letters for matrices. For a matrix $A$, we use $A_{-1,-1}$ to denote the bottom-right entry (the last row and the last column). We use $\| \cdot \|$ to denote $\ell_2$ norm of a vector and and $\| \cdot \|_F$ denotes the Frobenius norm of a matrix. We use $I_d$ to denote the $d \times d$ identity matrix and $0_d, 0_{d \times d}$ to denote the zero vector and the zero matrix of size $d$ and $d \times d$ respectively. We omit the subscripts if the size can be inferred from the context. We denote $\mathrm{Sym}(A) := \frac{1}{2}(A + A^\top)$ the symmetrized version for any real square matrix $A$.

### 2.1   MEAN-VARYING LINEAR REGRESSION

Following the previous line of work (Von Oswald et al., 2023; Akyürek et al., 2023; Ahn et al., 2023; Mahankali et al., 2023; Zhang et al., 2024a), we introduce the *mean-varying linear regressions* problem.

In this problem there are different linear regression tasks. For each linear regression task $\tau$, we first sample a mean $\mu_\tau \sim \mathcal{N}(0, I_d)$ and the linear weight $w_\tau \sim \mathcal{N}(0, I_d)$. We then independently sample in-context examples $x_{\tau,i}$ and the query example $x_{\tau,\text{query}}$ from Gaussian distribution $\mathcal{N}(\mu_\tau, \Lambda)$. The input with task descriptors is $S_\tau = (\mu_\tau, x_{\tau,1}, w_\tau^\top x_{\tau,1}, \ldots, x_{\tau,n}, w_\tau^\top x_{\tau,n}, x_{\tau,\text{query}})$. The goal of the Transformer is to compute $y_{\tau,\text{query}} = w_\tau^\top x_{\tau,\text{query}}$.

**Embedding matrix $E_\tau$.**   There are many ways to encode the input as a series of tokens for the transformer. For most parts of the paper, we consider the following embedding matrix $E_\tau$ which duplicates the task descriptor before each stack of $(x, y)^\top$. That is,

$$E_\tau = \begin{pmatrix} \mu_\tau & \mu_\tau & \cdots & \mu_\tau & \mu_\tau \\ x_{\tau,1} & x_{\tau,2} & \cdots & x_{\tau,n} & x_{\tau,\text{query}} \\ y_{\tau,1} & y_{\tau,2} & \cdots & y_{\tau,n} & 0 \end{pmatrix}. \tag{1}$$

Here we set the last query stack to be $(\mu_\tau, x_{\tau,\text{query}}, 0)^\top$ and the zero entry remains to be filled with the prediction of the model. This particular embedding is chosen to simplify the optimal solution and optimization process. One can of course think of alternative embeddings. For example, we also

consider the following *prefix embedding* $E_\tau^{pre}$ in experiments:

$$E_\tau^{pre} = \begin{pmatrix} 1 & 0 & 0 & \cdots & 0 & 0 \\ 0 & 1 & 1 & \cdots & 1 & 0 \\ \mu_\tau & x_{\tau,1} & x_{\tau,2} & \cdots & x_{\tau,n} & x_{\tau,\text{query}} \\ 0 & y_{\tau,1} & y_{\tau,2} & \cdots & y_{\tau,n} & 0 \end{pmatrix}. \tag{2}$$

In this embedding the task descriptor $\mu_\tau$ is just represented in the first token. The first two rows serve as a simplified version of positional encoding that distinguishes the task descriptor, in-context examples and the query example.

## 2.2 TRAINING DETAILS

To do training on the mean-varying linear regressions problem, we first describe the linear self-attention layer, and then give the training loss and initialization details.

**Model Architecture.** A standard self-attention Transformer layer with one head computes the following update to the embedding $E$:

$$f(E; W) = E + W^P W^V E M \cdot \text{softmax}\left(\frac{(W^K E)^\top W^Q E}{\rho}\right).$$

Here $\rho$ is a normalizing factor, $E$ is the input embedding matrix and $M$ is a masking matrix. Similar to previous results(Ahn et al., 2023; Von Oswald et al., 2023; Zhang et al., 2024a)., we consider a simplified version of one-layer linear self-attention (LSA) Transformer. Specifically, the projection matrix and the value matrix are merged into a projection-value matrix $W^{PV} \in \mathbb{R}^{d_E \times d_E}$, and the key matrix and query matrix are merged into a key-query matrix $W^{KQ} \in \mathbb{R}^{d_E \times d_E}$. Here $d_E$ is the embedding dimension. The attention is also restricted to the first $n$ tokens that represent in-context examples (and excludes the query token):

$$f_{\text{LSA}}(E; W) = E + W^{PV} E M \cdot \frac{E^\top W^{KQ} E}{n}, \quad M = \begin{pmatrix} I_n & 0_n \\ 0_n^\top & 0 \end{pmatrix}. \tag{3}$$

Here $W = (W^{KQ}, W^{PV})$ and the normalizing factor is set to be the number of in-context examples $n$. Note that the masking matrix $M$ excludes attentions to the query token.

**Model Prediction.** For input embedding $E = E_\tau$, the prediction is read out from the bottom-right entry of the output

$$\hat{y}_{\tau,\text{query}} = f_{\text{LSA}}(E_\tau; W)_{-1,-1}. \tag{4}$$

**Training Loss.** Let $\ell(W, \tau)$ be the expected least-square error for task $\tau$. That is,

$$\ell(W, \tau) := \frac{1}{2} \mathbb{E}_{x_{\tau,i}, x_{\tau,\text{query}}, w_\tau} \left[ \left( f_{\text{LSA}}(E_\tau; W)_{-1,-1} - w_\tau^\top x_{\tau,\text{query}} \right)^2 \right]. \tag{5}$$

Note that different tasks may have different expected loss as they have different $\mu_\tau$. In training we take expectation over all tasks:

$$L(W) := \mathbb{E}_{\mu_\tau \sim \mathcal{N}(0, I_d)} \left[ \ell(W, \tau) \right] \tag{6}$$

This represents the population loss for training. In practice, we can generate $m$ sequences $S_{\tau_1}, S_{\tau_2}, ..., S_{\tau_m}$, and the empirical loss is just the mean-squared error for all the sequences

$$\hat{L}(W) := \frac{1}{m} \sum_{i=1}^m \left[ \left( f_{\text{LSA}}(E_{\tau_i}; W)_{-1,-1} - w_{\tau_i}^\top x_{\tau_i,\text{query}} \right)^2 \right]. \tag{7}$$

**Initialization.** We make the following assumption on the initialization. The assumption is motivated by the initialization in Zhang et al. (2024a).

**Assumption 2.1** (Initialization). We assume the initialization of the Transformer satisfies

$$W^{KQ}(0) = \begin{pmatrix} \Sigma_{11} & \Sigma_{12} & 0_d \\ \Sigma_{21} & \Sigma_{22} & 0_d \\ 0_d^\top & 0_d^\top & 0 \end{pmatrix}, W^{PV}(0) = \begin{pmatrix} 0_{d \times d} & 0_{d \times d} & 0_d \\ 0_{d \times d} & 0_{d \times d} & 0_d \\ 0_d^\top & 0_d^\top & \sigma \end{pmatrix}$$

where $\Sigma_{11}, \Sigma_{22}, \Sigma_{12}, \Sigma_{21}$ are PSD matrices and $\sigma$ satisfies the equation:

$$\sigma := \left( \|\Sigma_{11}\|_F^2 + \|\Sigma_{12}\|_F^2 + \|\Sigma_{21}\|_F^2 + \|\Sigma_{22}\|_F^2 \right)^{\frac{1}{2}} > 0.$$

A simple way to satisfy the requirement is to take $\Sigma_{11} = \Sigma_{12} = \Sigma_{21} = \Sigma_{22} = I_d$ and $\sigma = 2\sqrt{d}$. As we will later see in Section 3.2, setting $\sigma$ this way ensures that thoughout training $W^{KQ}$ and $W^{PV}$ matrices are always *balanced* – they always have the same Frobenius norm. This invariant is important for our analysis since, as demonstrated in Section 3.2, among the infinite global optimal solutions, we need only work with those that are balanced.

**Training procedure.** We run gradient flow on the population loss $L(W)$ from the initialization above.

$$\frac{dW}{dt} = -\nabla L(W). \tag{8}$$

It is possible to use standard techniques to discretize the process and run gradient descent, and it is also possible to use a polynomial number of samples to estimate the gradient. For simplicity in this paper we only work with the population loss and gradient flow.

## 2.3 1-LAYER LSA PERFORMING PRECONDITIONED GRADIENT DESCENT

As Ahn et al. (2023) observed, in the standard linear regression setting where $\mu_\tau$ is always set to 0, if the input data just consists of $E = \begin{pmatrix} x_1 & x_2 & \cdots & x_n & x_{\text{query}} \\ y_1 & y_2 & \cdots & y_n & 0 \end{pmatrix}$, then the optimal one layer LSA network computes a preconditioned gradient descent step. That is, they construct weights for 1-layer LSA network such that the predicted $\hat{y}_{\text{query}}$ can be computed by

$$\hat{y}_{\text{query}} = \langle x_{\text{query}}, \Lambda^{-1} \sum_{i=1}^{n} y_i x_i \rangle.$$

Here the vector $\sum_{i=1}^{n} y_i x_i$ is just a multiple of the gradient of a least squares objective $f(\hat{w}) = \frac{1}{2} \sum_{i=1}^{n} (y_i - \hat{w}^\top x_i)^2$. The matrix $\Lambda^{-1}$ is related to the covariance matrix of $x_i$'s and serves as a preconditioner. As we will see later, with task descriptors 1-layer LSA networks can discover more complicated strategies for the mean-varying linear regression problem.

## 3 WARM UP: LARGE SAMPLE SIZE

The global optimal solution and analysis for the general case are complicated. To highlight our main ideas we first describe our results in the limit of infinitely many samples $n \to \infty$.

### 3.1 MAIN RESULTS

Our main results for infinitely many samples are summarized in the following theorem. As we will see, trained Transformers learn to use task descriptors to "standardize" keys by removing the mean.

**Theorem 3.1** (Main result). *Using initialization as in Assumption 2.1, if the number of samples $n \to \infty$ and $\sigma$ satisfies $0 < \sigma < \alpha$ for some constant $\alpha$ [2], then the gradient flow (8) will converge[3] to the global minimizer $W_* = (W_*^{KQ}, W_*^{PV})$ and the corresponding loss $\lim_{n\to\infty} L(W_*) = 0$. Here we have*

$$W_*^{KQ} = \frac{1}{u^*} \begin{pmatrix} 0_{d\times d} & -\Lambda^{-1} & 0_d \\ 0_{d\times d} & \Lambda^{-1} & 0_d \\ 0_d^\top & 0_d^\top & 0 \end{pmatrix} \text{ and } W_*^{PV} = \begin{pmatrix} 0_{d\times d} & 0_{d\times d} & 0_d \\ 0_{d\times d} & 0_{d\times d} & 0_d \\ 0_d^\top & 0_d^\top & u^* \end{pmatrix} \tag{9}$$

*where $u^* = \left(2\|\Lambda^{-1}\|_F^2\right)^{\frac{1}{4}}$.*

**Keys standardization.** To understand what the Transformer is doing in this case, we notice that the $W_*^{KQ}$ matrix can be decomposed as the product of two matrices. Let $C$ and $\tilde{W}_*^{KQ}$ be

$$C = \begin{pmatrix} 0_{d\times d} & 0_{d\times d} & 0_d \\ -I_d & I_d & 0_d \\ 0_d^\top & 0_d^\top & 1 \end{pmatrix}, \tilde{W}_*^{KQ} = \begin{pmatrix} 0_{d\times d} & 0_{d\times d} & 0_d \\ 0 & \frac{1}{u^*}\Lambda^{-1} & 0_d \\ 0_d^\top & 0_d^\top & 1 \end{pmatrix}. \tag{10}$$

---

[2]Please see Lemma A.4 in the appendix for the value of $\alpha$.
[3]Here the gradient flow becomes $\frac{dW}{dt} = -\nabla \lim_{n\to\infty} L(W)$.

Then we have $W_*^{KQ} = C^\top \tilde{W}_*^{KQ}$. Here $\tilde{W}_*^{KQ}$ is in fact the optimal solution for 1-layer LSA without task descriptors (see Zhang et al. (2024a)) and $C$ is the standardization operator. The effect of matrix $C$ suggests that when computing the key for the attention, the optimal solution will first standardize the $x_i$ by removing its mean component to get $CE_\tau$, which has form

$$CE_\tau = \begin{pmatrix} 0 & 0 & \cdots & 0 & 0 \\ z_1 & z_2 & \cdots & z_n & z_{\text{query}} \\ y_1 & y_2 & \cdots & y_n & 0 \end{pmatrix}. \tag{11}$$

Here we call $z_i = x_i - \mu_\tau$ the standardized versions of $x_i$. Doing this helps remove the spurious correlations introduced by the nonzero mean $\mu_\tau$. More precisely, we compute the output $\hat{y}_{\text{query}}$ as:

$$\begin{aligned} \hat{y}_{\text{query}} &= x_{\text{query}}^\top \Lambda^{-1} \frac{1}{n} \sum_{i=1}^n z_i y_i \\ &= x_{\text{query}}^\top \Lambda^{-1} \left( \frac{1}{n} \sum_{i=1}^n z_i x_i^\top \right) w \\ &\to x_{\text{query}}^\top w \quad \text{as } n \to \infty. \end{aligned} \tag{12}$$

The last step uses the fact that $\frac{1}{n} \sum_{i=1}^n z_i x_i^\top$ converges to $\Lambda$ as $n$ goes to infinity, so we see that the transformer outputs the correct estimate and achieves 0 loss. On the other hand, if we did not have the $C$ matrix (and hence did not use the task descriptor), the corresponding matrix would be $\frac{1}{n} \sum_{i=1}^n x_i x_i^\top$ which only converges to $\Lambda + \mu_\tau \mu_\tau^\top$. This matrix is different for different $\mu_\tau$ and it is impossible to invert it by changing the weight matrix (the weight matrices need to work simultaneously for all tasks).

### 3.2 PROOF SKETCH FOR THEOREM 3.1

Now we discuss how to prove the convergence result in Theorem 3.1. Several steps of our proof are similar to the proof in Zhang et al. (2024a) for the case without task descriptors, but as we shall see having task descriptors introduces additional challenges to the proof.

We first characterize the set of global minimum and give a lowerbound of the gradient when the current solution is not globally optimal. As we shall see, there are actually infinitely many global optimal solutions and it would be difficult to lowerbound the norm of the gradient by the distance to the particular optimal solution in Theorem 3.1. We get around this issue by showing that the training dynamics maintain several invariant properties, and the only global optimal solution that satisfies all these invariant properties is the solution in Theorem 3.1.

**Characterizing global minima and gradient lowerbound** Our main lemma below gives a strong characterization of global minima and show that the gradient can be lowerbounded if we are away from the set of global minima.

**Lemma 3.2.** *If our initialization satisfies Assumption 2.1 and $n \to \infty$, then we have*

$$\begin{aligned} \|\nabla L(W)\|_F^2 \geq c \Bigg( &\left\| \text{Sym}(W_{11}^{KQ} + W_{12}^{KQ} + W_{22}^{KQ} + W_{21}^{KQ}) \right\|_F^2 + \left\| W_{22}^{KQ} + W_{21}^{KQ} - \frac{\Lambda^{-1}}{u} \right\|_F^2 \\ &+ \|W_{12}^{KQ} + \frac{\Lambda^{-1}}{u}\|_F^2 + \|W_{22}^{KQ} - \frac{\Lambda^{-1}}{u}\|_F^2 \Bigg) \end{aligned} \tag{13}$$

*for some constant $c > 0$.*

All four terms in the lemma above needs to be equal to 0 in order to achieve a global optimal solution. After examining the four equations, we can show that there are two types of symmetry for the loss function. First, if $W^{KQ}$ is scaled by factor $\kappa$ and $W^{PV}$ is scaled by factor $1/\kappa$, then the function computed by the LSA layer does not change. Second, if we add skew-symmetric matrices $U$ ($U^\top = -U$) to the 11 block of $W^{KQ}$, it also doesn't change the solution. The second type of symmetry is unique to the setting with task descriptors. Every global minimum is equivalent to the minimum we constructed in Theorem 3.1 up to these transformations.

To make the training process easier to analyze and allow use to focus on the particular global minimum in Theorem 3.1, we compliment Lemma 3.2 with the following invariant result of training dynamics:

**Lemma 3.3.** *If the initialization follows Assumption 2.1, then throughout gradient flow training, the following invariants are maintained:*

1. *Balancing condition:*

$$u^2 = \|W_{11}^{KQ}\|_F^2 + \|W_{12}^{KQ}\|_F^2 + \|W_{21}^{KQ}\|_F^2 + \|W_{22}^{KQ}\|_F^2. \tag{14}$$

2. $W^{KQ}$ *is a symmetric matrix.*

3. $u, W_{11}^{KQ}, W_{12}^{KQ}, W_{21}^{KQ}$ *and* $W_{22}^{KQ}$ *are the only non-zero weights.*

4. *If $u$ is smaller than some positive constant $\alpha$ at initialization, then $u > \beta$ for another positive constant $\beta$ throughout training.*

This shows that with the initialization in Assumption 2.1, we never need to worry about the blocks except for $u, W_{11}^{KQ}, W_{12}^{KQ}, W_{21}^{KQ}$ and $W_{22}^{KQ}$. The first symmetry of scaling is now fixed because $u$ and $W_{11}^{KQ}, W_{12}^{KQ}, W_{21}^{KQ}, W_{22}^{KQ}$ are always balanced. The second symmetry also cannot happen because $W_{11}^{KQ}$ remains symmetric. By combining invariants from training trajectory (Lemma 3.3) and the landscape result of lowerbounding the gradient (Lemma 3.2) we can get Theorem 3.1.

## 4 OPTIMAL SOLUTION FOR FINITE NUMBER OF SAMPLES

Surprisingly a 1-layer LSA network comes up with a much more complicated strategy when the sample size $n$ is finite. In fact the optimal solutions differ significantly from that of infinitely many samples when $n$ is not much larger than $d$. In this section for simplicity we assume that the covariance matrix for $x_i$'s are identity ($\Lambda = I_d$), and summarize the results in theorem below:

**Theorem 4.1** (Main theorem, finite sample). *Under Assumption 2.1, if the number of samples $n \geq 2$ and $\sigma$ satisfies $0 < \sigma < \alpha$ for some constant $\alpha$, then the gradient flow (8) will converge to a global minimizer $W_* = (W_*^{KQ}, W_*^{PV})$. If the covariance matrix $\Lambda = I_d$, then the global minimizer $W_* = (W_*^{KQ}, W_*^{PV})$ satisfies*

$$W_*^{KQ} = \begin{pmatrix} a_{11}I_d & a_{12}I_d & 0_d \\ a_{21}I_d & a_{22}I_d & 0_d \\ 0_d^\top & 0_d^\top & 0 \end{pmatrix}, W_*^{PV} = \begin{pmatrix} 0_{d\times d} & 0_{d\times d} & 0_d \\ 0_{d\times d} & 0_{d\times d} & 0_d \\ 0_d^\top & 0_d^\top & b \end{pmatrix}. \tag{15}$$

*Here $b, a_{11}, a_{21}, a_{12}, a_{22}$ are all numbers depending only on $n, d$. For $z \in \{b, a_{11}, a_{21}, a_{12}, a_{22}\}$ we denote $z = f_z(n, d)$, the exact formulas for $f_z$'s are given Theorem A.8 in Appendix.*

As we can see, the optimal solution here makes use of two new blocks in $W_*^{KQ}$ corresponding to $a_{11}I_d$ and $a_{21}I_d$. Intuitively, these two blocks leverage the descriptors $\mu_\tau$ in queries, which is equal to $\mathbb{E}_{z_{\text{query}}}[x_{\tau,\text{query}}]$ and thus can be help reduce the *bias* introduced by $z_{\text{query}}$. To see this, we need a careful way to decompose the training loss function.

**Decomposition of the training loss**   Taking expectation over $z_{\text{query}}$ and decomposing the loss into bias and variance terms, we have

$$L(W) = \underbrace{\frac{1}{2}\mathbb{E}_{x_i,\mu_\tau,w_\tau}\left[\left(\mathbb{E}_{z_{\text{query}}}[\hat{y}_{\tau,\text{query}} - y_{\tau,\text{query}}]\right)^2\right]}_{\text{bias}:L_2}$$
$$+ \underbrace{\frac{1}{2}\mathbb{E}_{x_i,\mu_\tau,w_\tau}\left[\mathbb{E}_{z_{\text{query}}}[(\hat{y}_{\tau,\text{query}} - y_{\tau,\text{query}})^2] - \left(\mathbb{E}_{z_{\text{query}}}[\hat{y}_{\tau,\text{query}} - y_{\tau,\text{query}}]\right)^2\right]}_{\text{variance}:L_1}. \tag{16}$$

We will show that the $L_1$ and $L_2$ terms can achieve their individual optimal value independently. To do that we rely on a reparametrization trick to separate the variables used in $L_1$ and $L_2$.

**Simplify the loss in the new parametrization.** First we note that $w_{31}^{KQ} = w_{32}^{KQ} = w_{31}^{PV} = w_{32}^{PV} = 0_d$ is the critical point and they are zero at initialization. We show that this is maintained throughout training. This allows us to simplify the prediction $\hat{y}_{\tau,\text{query}}$. Denote $\hat{\mu}_\tau = \frac{1}{n}\sum_{i=1}^n x_i$ and $\hat{\Lambda} = \frac{1}{n}\sum_{i=1}^n x_i x_i^\top$. We can then simplify the prediction to be

$$\hat{y}_{\tau,\text{query}} = b w_\tau^\top \left( a_{11}\|\mu_\tau\|^2 w_\tau^\top \hat{\mu}_\tau + a_{21}\hat{\Lambda}\mu_\tau + a_{12}\mu_\tau^\top \hat{\mu}_\tau x_{\tau,\text{query}} + a_{22}\hat{\Lambda}x_{\tau,\text{query}} \right). \tag{17}$$

Intuitively, for this to predict $w_\tau^\top x_{\tau,\text{query}}$, it makes sense to use $a_{12}$ and $a_{22}$ – this is exactly what happens in the infinite sample regime because there $\hat{\Lambda} = \mu_\tau \mu_\tau^\top + I_d$ and $\hat{\mu}_\tau \mu_\tau^\top = \mu_\tau \mu_\tau^\top$, so it is possible to get $x_{\tau,\text{query}}$ just by combining these two terms. However this is no longer true for the finite sample case as the empirical mean $\hat{\mu}$ and second moment $\hat{\Lambda}$ are different from their expectations, and the first two terms provide useful information about them that helps reducing the loss.

Recall $z_i = x_{\tau,i} - \mu_\tau$ is the standardized version of the $i$-th sample. We can in turn write the empirical variance matrix $\hat{Z} := \frac{1}{n}\sum z_i z_i^\top$ and standardized mean $\hat{z} := \frac{1}{n}\sum_{i=1}^n z_i$. Substitute the simplified prediction $\hat{y}_{\tau,\text{query}}$ back to (16) and we reach

$$L(W) = \underbrace{\frac{1}{2}\mathbb{E}_{z_i,\mu_\tau}\left[\left\|\left(b(a_{11}+a_{12}+a_{21}+a_{22})\|\mu_\tau\|^2\hat{\mu}_\tau + b(a_{21}+a_{22})\left(\mu_\tau\hat{z}^\top + \hat{Z}\right)\mu_\tau - \mu_\tau\right)\right\|^2\right]}_{L_2}$$

$$+ \underbrace{\frac{1}{2}\mathbb{E}_{z_i,\mu_\tau}\left[\left\|b(a_{12}+a_{22})\hat{\mu}_\tau\mu_\tau^\top + ba_{22}\left(\mu_\tau\hat{z}^\top + \hat{Z}\right) - I_d\right\|_F^2\right]}_{L_1}.$$

$$\tag{18}$$

Therefore we can write the loss $L(W)$ in the new parametrization $\theta$:

$$\begin{cases} \theta_1 = b(a_{11}+a_{12}+a_{21}+a_{22}) \\ \theta_2 = b(a_{21}+a_{22}) \\ \theta_3 = b(a_{12}+a_{22}) \\ \theta_4 = ba_{22}. \end{cases} \tag{19}$$

Note that for any values of $\theta_1,\theta_2,\theta_3,\theta_4$, there are always values of $b,a_{11},a_{12},a_{21},a_{22}$ that can achieve these $\theta$-values (indeed, the $\theta$ parameters are just a full rank linear transformation of $ba_{11},ba_{12},ba_{21},ba_{22}$). Therefore for the optimal solution it suffices to consider the parametrization of $\theta_1,\theta_2,\theta_3,\theta_4$. Since $L_1$ only depends on $\theta_3$ and $\theta_4$ and $L_2$ only depends on $\theta_1$ and $\theta_2$, we can simply consider the optimal solution for the following two functions separately:

$$L_1(\theta_3,\theta_4) = \frac{1}{2}\mathbb{E}_{z_i,\mu_\tau}\left[\left\|\theta_3\hat{\mu}_\tau\mu_\tau^\top + \theta_4\left(\mu_\tau\hat{z}^\top + \hat{Z}\right) - I_d\right\|_F^2\right],$$

$$L_2(\theta_1,\theta_2) = \frac{1}{2}\mathbb{E}_{z_i,\mu_\tau}\left[\left\|\left(\theta_1\|\mu_\tau\|^2\hat{\mu}_\tau + \theta_2\left(\mu_\tau\hat{z}^\top + \hat{Z}\right)\mu_\tau - \mu_\tau\right)\right\|^2\right].$$

**No need of non-zero $a_{11}$ and $a_{21}$ if $n = \infty$.** In the infinite sample regime, notice that $\hat{\mu}_\tau \to \mu_\tau$, $\hat{Z} \to I_d$ and $\hat{z} \to 0_d$ as $n \to \infty$, the decomposition becomes even simpler

$$\lim_{n\to\infty} L(\theta) = \underbrace{\frac{1}{2}\mathbb{E}_{z_i,\mu_\tau}\left[\left\|\left(\theta_1\|\mu_\tau\|^2\mu_\tau + \theta_2\mu_\tau - \mu_\tau\right)\right\|^2\right]}_{L_2} + \underbrace{\frac{1}{2}\mathbb{E}_{z_i,\mu_\tau}\left[\left\|\theta_3\mu_\tau\mu_\tau^\top + (\theta_4-1)I_d\right\|_F^2\right]}_{L_1}.$$

$$\tag{20}$$

To minimize both $L_1$ and $L_2$ here, it suffices to choose $\theta_2 = \theta_4 = 1$ and $\theta_1 = \theta_3 = 0$, which corresponds to the optimal solution in Theorem 3.1 where $a_{11} = a_{21} = 0$.

**Non-zero $a_{11}$ and $a_{21}$ to minimize bias for finite $n$.** When $n$ is finite, $\hat{Z}$ and $\hat{z}$ deviate from their limits hence just setting $\theta_2 = \theta_4 = 1$ and $\theta_1 = \theta_3 = 0$ are not sufficient to minimize $L_1$ and $L_2$

at the same time. That is, there will be a tradeoff between bias and variance if only using $ba_{12}$ and $ba_{22}$. To avoid the tradeoff and minimize both, Transformers need to learn a more delicate structure where $ba_{11}$ and $ba_{21}$ are also used to better estimate $\mu_\tau$. These requirements give us the closed form formulas for $\theta_i$'s and then we can solve for the optimal solutions for $b, a_{11}, a_{12}, a_{21}, a_{22}$. The final result is in Section A.2 Appendix.

The optimization of the finite sample case follows from similar ideas as the infinite sample case, except that the calculations are more complicated. We leave the details in Section A.2 in Appendix.

## 5 EXPERIMENTS

In this section we show that Transformers can indeed leverage the task descriptor in the mean-varying linear regression problem. For 1-layer Transformers with descriptor embedding, the trained Transformer converges to the optimal solution as our theory predicts. Deeper Transformers can further improve the performance both with or without task descriptors, but the Transformers with task descriptors can always leverage that information to achieve lower loss.

### 5.1 EXPERIMENT SETUP

**Model architecture.** We train $L$-layer LSA Transformers parametrized as below:

$$H_{\ell+1} = H_\ell + W_\ell^{PV} M H_\ell \cdot \frac{H_\ell^\top W_\ell^{KQ} H_\ell}{n} \quad \text{for } \ell = 0, 1, \ldots, L-1.$$

Here $H_0$ is the input matrix, set to be the embedding matrix and the prediction is read out from the bottom-right entry of the output $\hat{y}_{\tau,\text{query}} = [H_L]_{-1,-1}$. Recall $M$ is the masking for attention that restricts attention to in-context tokens (excluding the query). We have also done experiments with separate $W^K, W^Q, W^P, W^V$ matrices in Section B.1 at Appendix and results are similar.

**Embedding matrix.** Our weight matrices $W^{KQ}, W^{PV}$ are both $d_E \times d_E$ matrices. We use three types of input embedding matrices: embedding with task descriptor $E_\tau$ in (1), prefix embedding $E_\tau^{pre}$ in (2) and embedding without task descriptor

$$E = \begin{pmatrix} 1 & 1 & \cdots & 1 & 0 \\ x_{\tau,1} & x_{\tau,2} & \ldots & x_{\tau,n} & x_{\tau,\text{ query}} \\ y_{\tau,1} & y_{\tau,2} & \cdots & y_{\tau,n} & 0 \end{pmatrix}. \tag{21}$$

The first row of $E$ and first two rows of $E_\tau^{pre}$ serve as a simplified version of positional encoding.

**Data hyperparameters.** We generate 4096 i.i.d. input sequences for each episode of training. For all experiments in this section, the data dimension $d = 5$ and the covariance matrix $\Lambda = I_d$. We leave experiments for different $d$ and $\Lambda$ to Section B.2 at Appendix.

**Training algorithms.** For all experiments, we use Adam optimizer (Kingma & Ba, 2015) to train Transformers. We also use $\ell_2$ gradient clipping to stabilize training.

### 5.2 EXPERIMENT RESULTS

We first show that for different number of layers and embeddings, Transformers can find ways to leverage the additional information in task descriptor.

We plot the ICL training loss curves in Figure 2. First we note that embedding with task descriptors have smaller loss values than embedding without descriptors for $1, 2, 3$-layer Transformers.

Single-layer Transformers trained on prefix embeddings and embeddings without descriptors exhibit comparable performance. However, as the depth of the Transformer increases, models trained on prefix embeddings achieve lower loss. Surprisingly, 3-layer Transformers outperform those trained on embeddings without descriptors and even surpass those trained on embeddings with duplicated descriptors. This suggests that with prefix embeddings, deeper Transformers may be able to use the descriptor more efficiently. We explore some possibilities by observing the attention matrices in Section B.3 at Appendix.

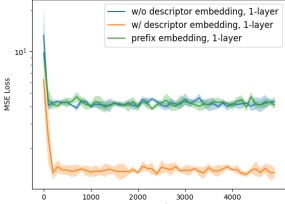 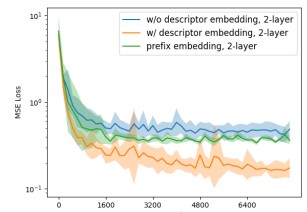 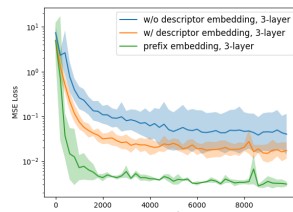

Figure 2: The MSE loss curves for Transformers in different depths. We display mean and std of 5 random seeds. The number of samples in each sequence $n = 50$ and data dimension $d = 5$.

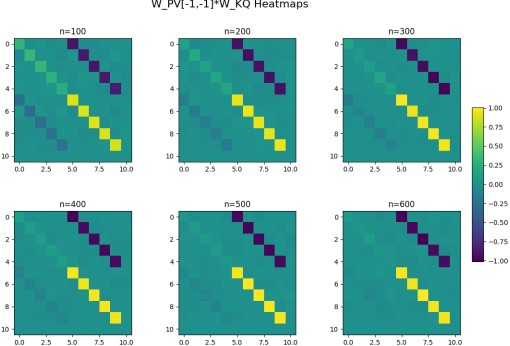

Figure 3: The heat map of $W^{PV}_{-1,-1} \cdot W^{KQ}$ for a well-trained Transformer with task descriptors $\mu_\tau$ in the training sequences. The number of in-context examples $n$ vary from 100 to 600.

Next we show that 1-layer LSA network indeed finds the optimal solution as our theory predicts. To do that, we plot heat maps of the product $W^{PV}_{-1,-1} \cdot W^{KQ}$ of trained Transformers in Figure 3. As Theorem 4.1 suggests, in the heat maps there are four blocks in forms of multiples of $I_d$. The two left blocks fade while the two right blocks remain prominent as $n$ grows large, which is consistent with the infinite sample regime in Theorem 3.1. Detailed formulas for parameters in Theorem 4.1 (appearing in Theorem A.8 in appendix) suggest that $ba_{11}$ will converge to 0 from above as $n \to \infty$ and $ba_{21}$ will converge to 0 from below as $n \to \infty$. The two other values $ba_{12}$ and $ba_{22}$ will be approximately equal to $-1$ and 1 respectively as long as $n$ is much larger than $d$. These trends are all observed in Figure 3.

## 6 CONCLUSIONS

In this work, we investigate how Transformers leverage task descriptions for in-context learning. Specifically, we consider the mean-varying linear regression problem where the task descriptors can be set to be the mean $\mu_\tau$ for each task $\tau$. We give a global convergence result for Transformers trained with task descriptors. Our theoretical result shows that even 1-layer linear Transformers can discover interesting ways to leverage the task descriptor. We empirically show that Transformers can achieve much lower loss for ICL when task descriptors are provided. We also find a clear pattern in the parameters of well-trained Transformers, which verifies our theoretical result. Immediate open problems include understanding how Transformers can make use of task descriptors when the embedding does not duplicate the descriptor, as well as what happens for multi-layer and/or nonlinear versions of Transformers. As future work, it would also be interesting to explore how Transformers leverage more general task descriptors across a broader range of ICL tasks.

## 7 ACKNOWLEDGEMENTS

Rong Ge and Ruomin Huang are supported by NSF Award DMS-2031849.

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

CONTENTS

# A    OMITTED PROOFS

In this section, we assume $\mathbb{E}[\cdot]$ is taking expectation over all randomness without specifying. We denote $\lambda_{\min}(A)$ and $\lambda_{\max}(A)$ the smallest and largest eigenvalue for any matrix $A$. For any PSD matrix $A$, denote $\sqrt{A}$ the unique PSD matrix such that $(\sqrt{A})^2 = A$. Sometimes we also write $\sqrt{A}$ as $A^{\frac{1}{2}}$. We denote $[k] = \{1, 2, \cdots, k\}$.

We first prove the main result for infinite number of samples in Section A.1, then we extend the result to the more complicated finite sample setting in Section A.2.

## A.1    PROOF FOR THEOREM 3.1

In this section we proof Theorem 3.1, which we restate here.

**Theorem A.1.** *Using initialization as in Assumption 2.1, if the number of samples $n \to \infty$ and $\sigma$ satisfies $0 < \sigma < \alpha$ for some constant $\alpha$ [4], then the gradient flow (8) will converge to the global minimizer $W_* = (W_*^{KQ}, W_*^{PV})$ and the corresponding loss $\lim_{n \to \infty} L(W_*) = 0$. Here we have*

$$W_*^{KQ} = \frac{1}{u^*} \begin{pmatrix} 0_{d \times d} & -\Lambda^{-1} & 0_d \\ 0_{d \times d} & \Lambda^{-1} & 0_d \\ 0_d^\top & 0_d^\top & 0 \end{pmatrix} \text{ and } W_*^{PV} = \begin{pmatrix} 0_{d \times d} & 0_{d \times d} & 0_d \\ 0_{d \times d} & 0_{d \times d} & 0_d \\ 0_d^\top & 0_d^\top & u^* \end{pmatrix} \tag{22}$$

*where $u^* = \left(2\|\Lambda^{-1}\|_F^2\right)^{\frac{1}{4}}$.*

### A.1.1    PROOF SKETCH

Here we give the sketch of our proof to Theorem 3.1, which follows the proof framework in Zhang et al. (2024a). Before we start, let's write $W^{PV}$ and $W^{KQ}$ into blocks:

$$W^{PV} = \begin{pmatrix} W_{11}^{PV} & W_{12}^{PV} & w_{13}^{PV} \\ W_{21}^{PV} & W_{22}^{PV} & w_{23}^{PV} \\ (w_{31}^{PV})^\top & (w_{32}^{PV})^\top & w_{33}^{PV} \end{pmatrix} \tag{23}$$

and

$$W^{KQ} = \begin{pmatrix} W_{11}^{KQ} & W_{12}^{KQ} & w_{13}^{KQ} \\ W_{21}^{KQ} & W_{22}^{KQ} & w_{23}^{KQ} \\ (w_{31}^{KQ})^\top & (w_{32}^{KQ})^\top & w_{33}^{KQ} \end{pmatrix} \tag{24}$$

where all the $W_{11}, W_{12}, W_{21}, W_{22} \in \mathbb{R}^{d \times d}$, $w_{13}, w_{23}, w_{31}, w_{32} \in \mathbb{R}^d$ and $w_{33} \in \mathbb{R}$. By expanding the prediction $\hat{y}_{\tau,\text{query}} = f_{\text{LSA}}(E_\tau; W)_{2d+1,n+1}$, we know the prediction only depends on the weight blocks $W_{11}^{KQ}, W_{12}^{KQ}, w_{31}^{PV}, W_{21}^{KQ}, W_{22}^{KQ}, w_{32}^{PV}, w_{31}^{KQ}, w_{32}^{KQ}$ and $w_{33}^{PV}$. Therefore we will only consider the training dynamics of these relevant blocks. To simplify notation we gather all the relevant parameters in the following block matrix $U$.

$$U = \begin{pmatrix} U_{11} & U_{12} & u_{13} \\ U_{21} & U_{22} & u_{23} \\ u_{31}^\top & u_{32}^\top & u_{-1} \end{pmatrix} := \begin{pmatrix} W_{11}^{KQ} & W_{12}^{KQ} & w_{31}^{PV} \\ W_{21}^{KQ} & W_{22}^{KQ} & w_{32}^{PV} \\ (w_{31}^{KQ})^\top & (w_{32}^{KQ})^\top & w_{33}^{PV} \end{pmatrix}. \tag{25}$$

As we mentioned in Section 3, the proof of Theorem 3.1 relies on Lemma 3.3, which we restate here

**Lemma A.2.** *If the initialization follows Assumption 2.1, then throughout gradient flow training, the following invariants are maintained:*

1. *Balancing condition:*

$$u^2 = \|W_{11}^{KQ}\|_F^2 + \|W_{12}^{KQ}\|_F^2 + \|W_{21}^{KQ}\|_F^2 + \|W_{22}^{KQ}\|_F^2. \tag{26}$$

2. *$W^{KQ}$ is a symmetric matrix.*

3. *$u, W_{11}^{KQ}, W_{12}^{KQ}, W_{21}^{KQ}$ and $W_{22}^{KQ}$ are the only non-zero weights.*

---

[4]Please see Lemma A.4 for the value of $\alpha$.

    *4. If $u$ is smaller than some positive constant $\alpha$ at initialization, then $u > \beta$ for another positive constant $\beta$ throughout training.*

We split Lemma 3.3 into Lemma A.3 and Lemma A.4. We start with the dynamics of $u_{13}, u_{23}, u_{31}$ and $u_{32}$, which shows that these parameters stick to 0 during the training phase so the dynamics of $U$ could be simplified. Then we show that there is a balance between $u_{-1}$ and $U_{11}, U_{12}, U_{21}, U_{22}$. Specifically, we have the following lemma.

**Lemma A.3.** *If our initialization satisfies Assumption 2.1, then we have both*

$$u_{13}(t) = u_{23}(t) = u_{31}(t) = u_{32}(t) = 0 \tag{27}$$

*and*

$$u_{-1}(t)^2 = \|U_{11}(t)\|_F^2 + \|U_{12}(t)\|_F^2 + \|U_{21}(t)\|_F^2 + \|U_{22}(t)\|_F^2 \tag{28}$$

*for all $t \geq 0$. Additionally, we have $U_{11}$ remains symmetric throughout the training.*

Given the balanced condition, we can prove $u_{-1}$ could be lower bounded by some positive constant during the training phase in Lemma A.4, which suggests the trajectory of $u_{-1}$ is away from the saddle point $u_{-1} = 0$.

**Lemma A.4.** *If our initialization satisfies Assumption 2.1, $n \to \infty$ and $\sigma$ satisfies $0 < \sigma < \alpha$ where $\alpha$ is equal to*

$$\left( \frac{d+2}{2\|\Lambda\|_F \left( \|\Lambda\|_F^2 + 2\mathrm{tr}\,(\Lambda) + 3d^2 \right) + 28d\mathrm{tr}\,(\Lambda) + 60d^3} \right)^{\frac{1}{2}}, \tag{29}$$

*then we have $u_{-1}(t) \geq \beta > 0$ for all $t \geq 0$. Here*

$$\beta = \left( \frac{(d+2)\sigma^2}{\left( 4 + 2\sqrt{2} \right) \left( \|\Lambda\|_F^2 + 2\mathrm{tr}(\Lambda) + d^2 + 2d \right)} \right)^{\frac{1}{2}}. \tag{30}$$

Combining Lemma A.3 and Lemma A.4, we have the Lemma 3.3 in the main paper.

With the lower bound $\beta$ of $u_{-1}$, we are finally able to give an error bound (Luo & Tseng, 1993) of our loss $L(U)$ in Lemma A.5, which is the Lemma 3.2 in the main paper.

**Lemma A.5.** *If our initialization satisfies Assumption 2.1 and $n \to \infty$, then we have*

$$\|\nabla L(U)\|_F^2$$
$$\geq c \left( \|\mathrm{Sym}\,(U_{11} + U_{12} + U_{22} + U_{21})\|_F^2 + \left\| U_{22} + U_{21} - \frac{\Lambda^{-1}}{u_{-1}} \right\|_F^2 \right.$$
$$\left. + \|U_{12} + \frac{\Lambda^{-1}}{u_{-1}}\|_F^2 + \|U_{22} - \frac{\Lambda^{-1}}{u_{-1}}\|_F^2 \right) \tag{31}$$

*where*

$$c = \beta^2 \min \left( \frac{\lambda_{\min}(\Lambda)^2}{270d^3}, \frac{1}{270d^3}, \frac{\lambda_{\min}(\Lambda)^4}{90d^2}, \frac{1}{90d^2} \right).$$

With Lemma A.5 in hand, we can finally prove Theorem 3.1.

Given these supporting lemmas, we are now ready to prove the main theorem.

*Proof of Theorem 3.1.* Since $L(U) \geq 0$ is bounded below, we know $L(U_t)$ the loss over gradient flow will converge. Any stationary point $U^*$ of the gradient flow must satisfy $\nabla L(U^*) = 0$. Therefore, combining with the error bound (31) we have $\|U_{22}^* + U_{21}^* - \frac{\Lambda^{-1}}{u_{-1}^*}\|_F^2 = \|\mathrm{Sym}(U_{11}^* + U_{12}^* + U_{22}^* + U_{21}^*)\|_F^2 = \|U_{12}^* + \frac{\Lambda^{-1}}{u_{-1}^*}\|_F^2 = \|U_{22}^* - \frac{\Lambda^{-1}}{u_{-1}^*}\|_F^2 = 0$, which implies that $U_{22}^* = \frac{\Lambda^{-1}}{u_{-1}^*}, U_{12}^* = -\frac{\Lambda^{-1}}{u_{-1}^*}, U_{21}^* = 0_{d \times d}$ and $\mathrm{Sym}(U_{11}^*) = 0_{d \times d}$. By lemma A.3 we know this implies $U_{11}^* = 0_{d \times d}$. Finally by direct computation we know the corresponding loss is $L(U^*) = 0$, which implies that $U^*$ is a global minimizer. Combining (28), we have $u_{-1}^* = \left( 2\|\Lambda^{-1}\|_F^2 \right)^{\frac{1}{4}}$. Translating $U$ back to $W$ according to (25), we obtain Theorem 3.1. $\square$

The rest of this section will give detailed proofs for the supporting lemmas.

### A.1.2 Proofs of Lemma A.3, Lemma A.4 and Lemma A.5

The proof of Lemma A.3 mostly relies on calculations of gradients. We will show that (i) gradients are always $0$ for certain blocks and (ii) the growths of $W^{KQ}$ and $W^{PV}$ are balanced in training dynamics.

*Proof of Lemma A.3.* We first prove (27) by showing $0$ is the critical points for certain blocks.

**Part 1: critical points at zero.** The gradient of the loss is

$$\frac{\partial \ell(U, \tau)}{\partial U} = \mathbb{E}[(\hat{y}_{\tau,\text{query}} - w_\tau^\top x_{\tau,\text{query}}) \frac{\partial \hat{y}_{\tau,\text{query}}}{\partial U}].$$

To give the detailed gradient formulation, we need to expand $\hat{y}_{\tau,\text{query}}$ in terms of $U$ first. Denote $\hat{\Lambda} = \frac{1}{n} \sum_{i=1}^n x_{\tau,i} x_{\tau,i}^\top$ and $\hat{\mu}_\tau = \frac{1}{n} \sum_{i=1}^n x_{\tau,i}$. Then we have

$$
\begin{aligned}
\hat{y}_{\tau,\text{query}} = & \begin{pmatrix} u_{13}^\top & u_{23}^\top & u_{-1} \end{pmatrix} \begin{pmatrix} \mu_\tau \mu_\tau^\top & \mu_\tau \hat{\mu}_\tau^\top & \mu_\tau \cdot w_\tau^\top \hat{\mu}_\tau \\ \hat{\mu}_\tau \mu_\tau^\top & \hat{\Lambda} & \hat{\Lambda} w_\tau \\ \mu_\tau^\top \cdot w_\tau^\top \hat{\mu}_\tau & w_\tau^\top \hat{\Lambda} & w_\tau^\top \hat{\Lambda} w_\tau \end{pmatrix} \begin{pmatrix} U_{11} & U_{12} \\ U_{21} & U_{22} \\ u_{31}^\top & u_{32}^\top \end{pmatrix} \begin{pmatrix} \mu_\tau \\ x_{\tau,\text{query}} \end{pmatrix} \\
= & u_{13}^\top \left( \mu_\tau \mu_\tau^\top U_{11} + \mu_\tau \hat{\mu}_\tau^\top U_{21} + w_\tau^\top \hat{\mu}_\tau \mu_\tau u_{31}^\top \right) \mu_\tau \\
& + u_{13}^\top \left( \mu_\tau \mu_\tau^\top U_{12} + \mu_\tau \hat{\mu}_\tau^\top U_{22} + w_\tau^\top \hat{\mu}_\tau \mu_\tau u_{32}^\top \right) x_{\tau,\text{query}} \\
& + u_{23}^\top \left( \hat{\mu}_\tau \mu_\tau^\top U_{11} + \hat{\Lambda} U_{21} + \hat{\Lambda} w_\tau u_{31}^\top \right) \mu_\tau \\
& + u_{23}^\top \left( \hat{\mu}_\tau \mu_\tau^\top U_{12} + \hat{\Lambda} U_{22} + \hat{\Lambda} w_\tau u_{32}^\top \right) x_{\tau,\text{query}} \\
& + u_{-1} \cdot (\mu_\tau^\top w_\tau^\top \hat{\mu}_\tau U_{11} \mu_\tau + w_\tau^\top \hat{\Lambda} U_{21} \mu_\tau + w_\tau^\top \hat{\Lambda} w_\tau u_{31}^\top \mu_\tau) \\
& + u_{-1} \cdot (\mu_\tau^\top w_\tau^\top \hat{\mu}_\tau U_{12} x_{\tau,\text{query}} + w_\tau^\top \hat{\Lambda} U_{22} x_{\tau,\text{query}} + w_\tau^\top \hat{\Lambda} w_\tau u_{32}^\top x_{\tau,\text{query}}).
\end{aligned}
\tag{32}
$$

If letting $u_{13} = u_{23} = u_{31} = u_{32} = 0$, then we have

$$\hat{y}_{\tau,\text{query}} = u_{-1}(\mu_\tau^\top w_\tau^\top \hat{\mu}_\tau U_{11} \mu_\tau + w_\tau^\top \hat{\Lambda} U_{21} \mu_\tau + \mu_\tau^\top w_\tau^\top \hat{\mu}_\tau U_{12} x_{\tau,\text{query}} + w_\tau^\top \hat{\Lambda} U_{22} x_{\tau,\text{query}}). \tag{33}$$

The gradient on $u_{13}$ is

$$
\begin{aligned}
\frac{\partial \ell(U, \tau)}{\partial u_{13}} = & \mathbb{E}\left[ (\hat{y}_{\tau,\text{query}} - w_\tau^\top x_{\tau,\text{query}}) \left( \mu_\tau \mu_\tau^\top U_{11} + \mu_\tau \hat{\mu}_\tau^\top U_{21} \right) \mu_\tau \right] \\
& + \mathbb{E}\left[ (\hat{y}_{\tau,\text{query}} - w_\tau^\top x_{\tau,\text{query}}) \left( \mu_\tau \mu_\tau^\top U_{12} + \mu_\tau \hat{\mu}_\tau^\top U_{22} \right) x_{\tau,\text{query}} \right].
\end{aligned}
$$

Note that

$$\hat{y}_{\tau,\text{query}} - w_\tau^\top x_{\tau,\text{query}} = u_{-1} w_\tau^\top \cdot (\hat{\mu}_\tau \mu_\tau^\top U_{11} \mu_\tau + \hat{\Lambda} U_{21} \mu_\tau + \hat{\mu}_\tau \mu_\tau^\top U_{12} x_{\tau,\text{query}} + \hat{\Lambda} U_{22} x_{\tau,\text{query}} - \frac{x_{\tau,\text{query}}}{u_{-1}})$$

is degree-1 with respect to $w_\tau$. Also note that $\left( \mu_\tau \mu_\tau^\top U_{11} + \mu_\tau \hat{\mu}_\tau^\top U_{21} \right) \mu_\tau$ and $\left( \mu_\tau \mu_\tau^\top U_{12} + \mu_\tau \hat{\mu}_\tau^\top U_{22} \right) x_{\tau,\text{query}}$ do not contain $w_\tau$. Since $\mathbb{E}[w_\tau] = 0$ and $w_\tau$ is independent with all other random variables, we have $\frac{\partial \ell(U, \tau)}{\partial u_{13}} = 0$.

Similarly, we have $\frac{\partial \ell(U, \tau)}{\partial u_{23}} = 0$ given that $u_{13} = u_{23} = u_{31} = u_{32} = 0$.

Let $\Delta := (\hat{\mu}_\tau \mu_\tau^\top U_{11} \mu_\tau + \hat{\Lambda} U_{21} \mu_\tau + \hat{\mu}_\tau \mu_\tau^\top U_{12} x_{\tau,\text{query}} + \hat{\Lambda} U_{22} x_{\tau,\text{query}} - \frac{x_{\tau,\text{query}}}{u_{-1}}) \mu_\tau$. Then the gradient on $u_{31}$ is

$$
\begin{aligned}
\frac{\partial \ell(U, \tau)}{\partial u_{31}} &= \mathbb{E}\left[ u_{-1} w_\tau^\top \hat{\Lambda} w_\tau (\hat{y}_{\tau,\text{query}} - w_\tau^\top x_{\tau,\text{query}}) \mu_\tau \right] \\
&= \mathbb{E}\left[ u_{-1}^2 w_\tau^\top \hat{\Lambda} w_\tau w_\tau^\top \Delta \right] \\
&= 0.
\end{aligned}
$$

Similarly, we have $\frac{\partial \ell(U,\tau)}{\partial u_{32}} = 0$ given that $u_{13} = u_{23} = u_{31} = u_{32} = 0$. Taking expectation over $\mu_\tau$, we have $\frac{\partial L(U)}{\partial u_{13}} = \frac{\partial L(U)}{\partial u_{23}} = \frac{\partial L(U)}{\partial u_{31}} = \frac{\partial L(U)}{\partial u_{32}} = 0$ given that $u_{13} = u_{23} = u_{31} = u_{32} = 0$.

Now we prove (28) by showing the dynamics of $u_{-1}^2$ and $\|U_{11}\|_F^2 + \|U_{12}\|_F^2 + \|U_{21}\|_F^2 + \|U_{22}\|_F^2$ are balanced.

**Part 2: Balancing condition.** We can simplify the prediction $\hat{y}_{\tau,\text{query}}$ by letting $u_{13} = u_{23} = u_{31} = u_{32} = 0$ in (32), which gives the prediction

$$\hat{y}_{\tau,\text{query}} = u_{-1}w_\tau^\top \left( \left( \hat{\mu}_\tau \mu_\tau^\top U_{11} + \hat{\Lambda} U_{21} \right) \mu_\tau + \left( \hat{\mu}_\tau \mu_\tau^\top U_{12} + \hat{\Lambda} U_{22} \right) x_{\tau,\text{query}} \right).$$

This implies that

$$\hat{y}_{\tau,\text{query}} - y_{\tau,\text{query}} = u_{-1}w_\tau^\top \left( \left( \hat{\mu}_\tau \mu_\tau^\top U_{11} + \hat{\Lambda} U_{21} \right) \mu_\tau + \left( \hat{\mu}_\tau \mu_\tau^\top U_{12} + \hat{\Lambda} U_{22} - \frac{1}{u_{-1}} I_d \right) x_{\tau,\text{query}} \right).$$
(34)

Now we can compute the dynamics of $U$ by the chain rule $\frac{\partial \ell(U,\tau)}{\partial U} = \mathbb{E}\left[ (\hat{y}_{\tau,\text{query}} - y_{\tau,\text{query}}) \frac{\partial(\hat{y}_{\tau,\text{query}} - y_{\tau,\text{query}})}{\partial U} \right]$.

Therefore, we have the dynamics of $U_{11}, U_{12}, U_{21}, U_{22}$ and $u_{-1}$ as follows:

- 
$$\frac{\partial \ell(U,\tau)}{\partial U_{11}} = \mathbb{E}\left[ (\hat{y}_{\tau,\text{query}} - y_{\tau,\text{query}})u_{-1}w_\tau^\top \hat{\mu}_\tau \mu_\tau \mu_\tau^\top \right];$$
(35)

- 
$$\frac{\partial \ell(U,\tau)}{\partial U_{21}} = \mathbb{E}\left[ (\hat{y}_{\tau,\text{query}} - y_{\tau,\text{query}})u_{-1}\hat{\Lambda} w_\tau \mu_\tau^\top \right]$$
(36)

- 
$$\frac{\partial \ell(U,\tau)}{\partial U_{12}} = \mathbb{E}\left[ (\hat{y}_{\tau,\text{query}} - y_{\tau,\text{query}})u_{-1}w_\tau^\top \hat{\mu}_\tau \mu_\tau x_{\tau,\text{query}}^\top \right]$$
(37)

- 
$$\frac{\partial \ell(U,\tau)}{\partial U_{22}} = \mathbb{E}\left[ (\hat{y}_{\tau,\text{query}} - y_{\tau,\text{query}})u_{-1}\hat{\Lambda} w_\tau x_{\tau,\text{query}}^\top \right]$$
(38)

- 
$$\frac{\partial \ell(U,\tau)}{\partial u_{-1}} = \mathbb{E}\left[ (\hat{y}_{\tau,\text{query}} - y_{\tau,\text{query}}) w_\tau^\top \left( M_2 \mu_\tau + \left( M_1 + \frac{1}{u_{-1}} I_d \right) x_{\tau,\text{query}} \right) \right].$$
(39)

Here $M_1 := \hat{\mu}_\tau \mu_\tau^\top U_{12} + \hat{\Lambda} U_{22} - \frac{1}{u_{-1}} I_d$ and $M_2 := \hat{\mu}_\tau \mu_\tau^\top U_{11} + \hat{\Lambda} U_{21}$. Therefore we have

$$\frac{\partial \ell(U,\tau)}{\partial u_{-1}} \cdot u_{-1} = \text{tr}\left( U_{11}^\top \frac{\partial \ell(U,\tau)}{\partial U_{11}} + U_{12}^\top \frac{\partial \ell(U,\tau)}{\partial U_{12}} + U_{21}^\top \frac{\partial \ell(U,\tau)}{\partial U_{21}} + U_{22}^\top \frac{\partial \ell(U,\tau)}{\partial U_{22}} \right).$$
(40)

Taking expectation over $\mu_\tau$, we have the same thing holds for $L(U)$

$$\frac{\partial L(U)}{\partial u_{-1}} \cdot u_{-1} = \text{tr}\left( U_{11}^\top \frac{\partial L(U)}{\partial U_{11}} + U_{12}^\top \frac{\partial L(U)}{\partial U_{12}} + U_{21}^\top \frac{\partial L(U)}{\partial U_{21}} + U_{22}^\top \frac{\partial L(U)}{\partial U_{22}} \right).$$
(41)

This implies that

$$\frac{du_{-1}^2(t)}{dt} = \frac{d}{dt}\text{tr}\left( U_{11}(t)U_1^\top(t) + U_{12}(t)U_{22}^\top(t) + U_{21}(t)U_{21}^\top(t) + U_{22}(t)U_{22}^\top(t) \right).$$
(42)

Therefore if we set $u_{-1}(0)^2 = \|U_{11}(0)\|_F^2 + \|U_{12}(0)\|_F^2 + \|U_{21}(0)\|_F^2 + \|U_{22}(0)\|_F^2$ at initialization, we have

$$u_{-1}(t)^2 = \|U_{11}(t)\|_F^2 + \|U_{12}(t)\|_F^2 + \|U_{21}(t)\|_F^2 + \|U_{22}(t)\|_F^2$$
(43)

for all $t \geq 0$.

Finally, the gradient of $U_{11}$ as in (35) is obviously symmetric. Given that we also initialize $U_{11}$ to be symmetric, $U_{11}$ will remain symmetric throughout the training. $\qquad\square$

Now we prove Lemma A.4, which shows that $u_{-1}$ is actually away from the saddle point at origin with appropriate initialization. This proof relies on a careful analysis on loss $L$ as a function of $u_{-1}$. We can rewrite the loss $L_t(u_{-1}) = a(t)u_{-1}^2 - b(t)u_{-1} + c$ as a time-dependent degree-2 function of $u_{-1}$. Note that $a$ and $b$ depend on $U_{11}, U_{12}, U_{21}, U_{22}$ and are in turn time-dependent, while $c$ is a constant independent of time. If we set $u_{-1}(0)$ small enough (smaller than the $b/2a$ but greater than 0) at initialization, $L_0(u_{-1}(0))$ will be dominated by the degree-1 term, which has a negative coefficient at time 0. Specifically, we will have an upper bound $L_0(u_{-1}(0)) < L_0(0) - \epsilon = c - \epsilon$ for some positive number $\epsilon < c$. Another important observation is that the gradient flow of $U$ guarantees that $L$ will be no larger than $L_0(u_{-1}(0))$ throughout the training, which actually implies that $L_t(u_{-1}(t)) < c - \epsilon$ for all time $t$. For any time $t$, the set of $u_{-1}$ that gives loss value at most $c - \epsilon$ ($\{u_{-1} : L_t(u_{-1}) < c - \epsilon\}$) will always have a support that is always positive and bounded away from 0. We take a lower bound of $L_t(u_{-1})$ over all time $t$ to obtain the desired time-independent lower bound of $u_{-1}$. It is worth noting that to obtain time-independent lower and upper bounds of $L_t(u_{-1})$, we need the balancing condition between $u_{-1}$ and $U_{11}, U_{12}, U_{21}, U_{22}$ to eliminate the dependence of $L$ on $U_{11}, U_{12}, U_{21}, U_{22}$.

*Proof of Lemma A.4.* We first rewrite loss function $\ell(U, \tau)$ as a function of $u_{-1}$. By taking expectation over $\mu_\tau$, we can obtain $L(u_{-1})$ as a function of $u_{-1}$.

**Step 1: Rewrite loss as a degree-2 function of $u_{-1}$.** We will decompose the loss $\ell(U, \tau)$ into $\ell(U, \tau) = \ell_1(U, \tau) + \ell_2(U, \tau)$ and rewrite $\ell_1$ and $\ell_2$ separately. Recall $M_1 = \hat{\mu}_\tau \mu_\tau^\top U_{12} + \hat{\Lambda} U_{22} - \frac{1}{u_{-1}} I_d$ and $M_2 = \hat{\mu}_\tau \mu_\tau^\top U_{11} + \hat{\Lambda} U_{21}$. Then we have

$$\ell(U, \tau) = \frac{1}{2} \mathbb{E}_{z_i, z_{\text{query}}} \left[ (\hat{y}_{\tau, \text{query}} - y_{\tau, \text{query}})^2 \right] \tag{44}$$

$$= \frac{u_{-1}^2}{2} \left( \mathbb{E}_{z_i} \left[ \text{tr}(M_2^\top M_2 \mu_\tau \mu_\tau^\top) \right] + \mathbb{E}_{z_i} \left[ \text{tr} \left( M_1^\top M_1 \left( \Lambda + \mu_\tau \mu_\tau^\top \right) \right) \right] \right.$$
$$\left. + 2 \mathbb{E}_{z_i} \left[ \text{tr} \left( M_1^\top M_2 \mu_\tau \mu_\tau^\top \right) \right] \right) \tag{45}$$

$$= \underbrace{\frac{u_{-1}^2}{2} \mathbb{E}_{z_i} \left[ \text{tr} \left( M_1^\top M_1 \Lambda \right) \right]}_{\ell_1(U, \tau)} + \underbrace{\frac{u_{-1}^2}{2} \mathbb{E}_{z_i} \left[ \text{tr} \left( (M_2 + M_1)^\top (M_2 + M_1) \mu_\tau \mu_\tau^\top \right) \right]}_{\ell_2(U, \tau)}. \tag{46}$$

As we can see in (45), when taking expectation over $z_{\text{query}}$ given $\mu_\tau$, the variance term is $\frac{u_{-1}^2}{2} \mathbb{E} \left[ \text{tr} \left( M_1^\top M_1 \Lambda \right) \right]$, which is $\ell_1$. This implies that $\mathbb{E}[\ell_1] = L_1$ and $\mathbb{E}[\ell_2] = L_2$ where $L_2$ and $L_1$ are the bias and variance decomposition of the loss function. Now we compute the expectation in $\ell_1$ and $\ell_2$. Define a positive value $\gamma = \|\mu_\tau\|^2 + \frac{1}{n} \text{tr}(\Lambda)$ and a positive definite matrix $\Gamma = \frac{n+1}{n} \left( \Lambda + \mu_\tau \mu_\tau^\top \right) + \frac{1}{n} \left( \text{tr}(\Lambda) + \|\mu_\tau\|^2 \right) I_d$. By direct computation we have

$$\ell_1(U, \tau) = \frac{1}{2} u_{-1}^2 \text{tr} \left( \gamma U_{12}^\top \mu_\tau \mu_\tau^\top U_{12} \Lambda + U_{22}^\top \Gamma \left( \Lambda + \mu_\tau \mu_\tau^\top \right) U_{22} \Lambda + 2 U_{12}^\top \mu_\tau \mu_\tau^\top \left( \Gamma - \frac{2}{n} \mu_\tau \mu_\tau^\top \right) U_{22} \Lambda \right)$$

$$- u_{-1} \text{tr} \left( \left( \mu_\tau \mu_\tau^\top U_{12} \Lambda + \left( \Lambda + \mu_\tau \mu_\tau^\top \right) U_{22} \Lambda \right) \right) + \frac{1}{2} \text{tr}(\Lambda)$$

$$:= -c_{1,1} u_{-1} + c_{1,2} u_{-1}^2 + \frac{1}{2} \text{tr}(\Lambda). \tag{47}$$

Here $c_{1,1} = \text{tr} \left( \left( \mu_\tau \mu_\tau^\top U_{12} \Lambda + \left( \Lambda + \mu_\tau \mu_\tau^\top \right) U_{22} \Lambda \right) \right)$ is the coefficient of degree-1 term $-u_{-1}$ and $c_{1,2} = \frac{1}{2} \text{tr} \left( \gamma U_{12}^\top \mu_\tau \mu_\tau^\top U_{12} \Lambda + U_{22}^\top \Gamma \left( \Lambda + \mu_\tau \mu_\tau^\top \right) U_{22} \Lambda + 2 U_{12}^\top \mu_\tau \mu_\tau^\top \left( \Gamma - \frac{2}{n} \mu_\tau \mu_\tau^\top \right) U_{22} \Lambda \right)$ is the coefficient of degree-2 term $u_{-1}^2$ in $\ell_1$.

Similarly we have

$$
\begin{aligned}
\ell_2(U,\tau) =& \frac{1}{2}u_{-1}^2 \operatorname{tr}\Big(\gamma\left(U_{12}+U_{11}\right)^\top \mu_\tau\mu_\tau^\top\left(U_{12}+U_{11}\right) + \left(U_{22}+U_{21}\right)^\top\Gamma\left(\Lambda+\mu_\tau\mu_\tau^\top\right)\left(U_{22}+U_{21}\right)\mu_\tau\mu_\tau^\top \\
& +2\left(U_{12}+U_{11}\right)^\top\mu_\tau\mu_\tau^\top\left(\Gamma-\frac{2}{n}\mu_\tau\mu_\tau^\top\right)\left(U_{22}+U_{21}\right)\mu_\tau\mu_\tau^\top\Big) \\
& -u_{-1}\operatorname{tr}\left(\mu_\tau\mu_\tau^\top\left(U_{12}+U_{11}\right)\mu_\tau\mu_\tau^\top+\left(\Lambda+\mu_\tau\mu_\tau^\top\right)\left(U_{22}+U_{21}\right)\mu_\tau\mu_\tau^\top\right)+\frac{1}{2}\operatorname{tr}\left(\mu_\tau\mu_\tau^\top\right) \\
:=& -c_{2,1}u_{-1}+c_{2,2}u_{-1}^2+\frac{1}{2}\operatorname{tr}\left(\mu_\tau\mu_\tau^\top\right).
\end{aligned}
$$
(48)

Here $c_{2,1}=\operatorname{tr}\left(\mu_\tau\mu_\tau^\top\left(U_{12}+U_{11}\right)\mu_\tau\mu_\tau^\top+\left(\Lambda+\mu_\tau\mu_\tau^\top\right)\left(U_{22}+U_{21}\right)\mu_\tau\mu_\tau^\top\right)$ is the coefficient of degree-1 term $-u_{-1}$ and

$$
\begin{aligned}
c_{2,2} =& \frac{1}{2}\operatorname{tr}\Big(\gamma\left(U_{12}+U_{11}\right)^\top\mu_\tau\mu_\tau^\top\left(U_{12}+U_{11}\right)+\left(U_{22}+U_{21}\right)^\top\Gamma\left(\Lambda+\mu_\tau\mu_\tau^\top\right)\left(U_{22}+U_{21}\right)\mu_\tau\mu_\tau^\top \\
& +2\left(U_{12}+U_{11}\right)^\top\mu_\tau\mu_\tau^\top\left(\Gamma-\frac{2}{n}\mu_\tau\mu_\tau^\top\right)\left(U_{22}+U_{21}\right)\mu_\tau\mu_\tau^\top\Big)
\end{aligned}
$$

is the coefficient of degree-2 term $u_{-1}^2$ in $\ell_2$.

Note that coefficients $c_{1,1},c_{1,2},c_{2,1},c_{2,2}$ are all time-dependent positive functions of $\mu_\tau$. Taking expectation over $\mu_\tau$, we have

$$
\begin{aligned}
L_t(u_{-1}) &= \mathbb{E}_{\mu_\tau}\left[\ell_1\left(U,\tau\right)+\ell_2\left(U,\tau\right)\right] \\
&= \mathbb{E}_{\mu_\tau}\left[(c_{1,2}+c_{2,2})u_{-1}^2-(c_{2,1}+c_{1,1})u_{-1}\right]+\frac{1}{2}\mathbb{E}_{\mu_\tau}\left[\|\mu_\tau\|^2\right]+\frac{1}{2}\operatorname{tr}\left(\Lambda\right).
\end{aligned}
$$
(49)

**Step 2: bound the coefficients.** Now we give several useful lower and upper bounds on $c_{1,2}+c_{2,2}$ and $c_{2,1}+c_{1,1}$. Denote $c_{i,j}(t)$ as the corresponding coefficient at time $t$ under the gradient flow. The first two bounds yield a upper bound of $L_0(u_{-1}(0))<L_0(0)-\epsilon$ for some $\epsilon>0$. The third bound yields a lower bound for $L_t(u_{-1})$. Specifically, we have the following claim.

**Claim 1.** *We have the following three bounds:*

*1.*
$$
\mathbb{E}[c_{1,1}(0)+c_{2,1}(0)]\geq(d+2)u_{-1}(0),
$$
(50)

*2.*
$$
c_{1,2}+c_{2,2}\leq u_{-1}^2\|\Lambda+\mu_\tau\mu_\tau^\top\|_F^2\left(2\|\mu_\tau\|^2+\|\Lambda\|_F\right),
$$
(51)

*3.*
$$
c_{1,1}+c_{2,1}\leq(2+\sqrt{2})u_{-1}\|\Lambda+\mu_\tau\mu_\tau^\top\|_F^2.
$$
(52)

**Step 3: Upper bound $L_0(u_{-1}(0))$.** Now we can upper bound $L_0(u_{-1}(0))$.

$$
\begin{aligned}
L_0(u_{-1}(0)) &= \mathbb{E}\left[\ell_1\left(U(0),\tau\right)+\ell_2\left(U(0),\tau\right)\right] \\
&= \mathbb{E}\left[(c_{1,2}+c_{2,2})u_{-1}(0)^2-(c_{2,1}+c_{1,1})u_{-1}(0)\right]+\frac{1}{2}\mathbb{E}\left[\|\mu_\tau\|^2\right]+\frac{1}{2}\operatorname{tr}\left(\Lambda\right) \\
&\leq u_{-1}^2(0)\mathbb{E}\left[\|\Lambda+\mu_\tau\mu_\tau^\top\|_F^2\left(\|\Lambda\|_F+2\|\mu_\tau\|^2\right)u_{-1}^2(0)-(d+2)\right] \\
&\quad +\frac{1}{2}\mathbb{E}\left[\|\mu_\tau\|^2\right]+\frac{1}{2}\operatorname{tr}\left(\Lambda\right) \quad (\text{(50) and (51)}) \\
&\leq -\frac{1}{2}(d+2)u_{-1}^2(0)+\frac{1}{2}\mathbb{E}\left[\|\mu_\tau\|^2\right]+\frac{1}{2}\operatorname{tr}\left(\Lambda\right).
\end{aligned}
$$
(53)

The last inequality comes from that $u_{-1}(0)<\alpha=\left(\frac{d+2}{2\mathbb{E}\left[\|\Lambda+\mu_\tau\mu_\tau^\top\|_F^2\left(\|\Lambda\|_F+2\|\mu_\tau\|^2\right)\right]}\right)^{\frac{1}{2}}$ at initialization.

**Step 4: Show $u_{-1}$ is always positive.** Note that when $u_{-1} = 0$, the loss $L_t(0) = \frac{1}{2}\mathbb{E}_{\mu_\tau}\left[\|\mu_\tau\|^2\right] + \frac{1}{2}\mathrm{tr}\,(\Lambda)$. Therefore, $u_{-1}$ is non-zero whenever $L_t(u_{-1}) < \frac{1}{2}\mathbb{E}_{\mu_\tau}\left[\|\mu_\tau\|^2\right] + \frac{1}{2}\mathrm{tr}\,(\Lambda)$. Since $L_t(u_{-1}(t))$ is non-increasing in $t$ and by (53) we know $L_0(u_{-1}(0)) < \frac{1}{2}\mathbb{E}_{\mu_\tau}\left[\|\mu_\tau\|^2\right] + \frac{1}{2}\mathrm{tr}\,(\Lambda)$, we have $L_t(u_{-1}(t)) < \frac{1}{2}\mathbb{E}_{\mu_\tau}\left[\|\mu_\tau\|^2\right] + \frac{1}{2}\mathrm{tr}\,(\Lambda)$ for all $t \geq 0$, which implies that $u_{-1}$ is non-zero for all $t \geq 0$. Further since we have $u_{-1}(0) > 0$ and $u_{-1}(t)$ is continuous on $t$, we have $u_{-1} > 0$ for all $t \geq 0$.

**Step 5: Lower bound $L_t(u_{-1}(t))$ for all $t$.** Now we lower bound $L_t(u_{-1}(t))$.

$$
\begin{aligned}
L_t(u_{-1}(t)) &= \mathbb{E}\left[(c_{1,2} + c_{2,2})u_{-1}^2 - (c_{2,1} + c_{1,1})\,u_{-1}\right] + \frac{1}{2}\mathbb{E}\left[\|\mu_\tau\|^2\right] + \frac{1}{2}\mathrm{tr}\,(\Lambda) \\
&\geq -u_{-1}\mathbb{E}\left[c_{2,1} + c_{1,1}\right] + \frac{1}{2}\mathbb{E}\left[\|\mu_\tau\|^2\right] + \frac{1}{2}\mathrm{tr}\,(\Lambda) \\
&\geq -u_{-1}^2\mathbb{E}\left[(2 + \sqrt{2})\|\Lambda + \mu_\tau\mu_\tau^\top\|_F^2\right] + \frac{1}{2}\mathbb{E}\left[\|\mu_\tau\|^2\right] + \frac{1}{2}\mathrm{tr}\,(\Lambda) \quad (u_{-1} > 0 \text{ and } (52))
\end{aligned}
$$
$$\tag{54}$$

**Step 6: Combine lower and upper bounds together.** Since $L_t(u_{-1}(t)) \leq L_0(u_{-1}(0))$, combining (53) and (54) we have

$$
u_{-1} \geq \left(\frac{(d + 2)u_{-1}^2(0)}{(4 + 2\sqrt{2})\mathbb{E}\left[\|\Lambda + \mu_\tau\mu_\tau^\top\|_F^2\right]}\right)^{\frac{1}{2}} = \beta > 0. \tag{55}
$$

$\square$

Now we prove Claim 1.

*Proof.* Recall that
$$
c_{1,1} = \mathrm{tr}\left(\left(\mu_\tau\mu_\tau^\top U_{12}\Lambda + \left(\Lambda + \mu_\tau\mu_\tau^\top\right)U_{22}\Lambda\right)\right) \tag{56}
$$
and
$$
c_{2,1} = \mathrm{tr}\left(\mu_\tau\mu_\tau^\top\left(U_{12} + U_{11}\right)\mu_\tau\mu_\tau^\top + \left(\Lambda + \mu_\tau\mu_\tau^\top\right)\left(U_{22} + U_{21}\right)\mu_\tau\mu_\tau^\top\right). \tag{57}
$$
Computing the expectation, we have
$$
\mathbb{E}[c_{1,1}] = \mathrm{tr}\left(U_{12}\Lambda + U_{22}\Lambda + U_{22}\Lambda^2\right) \tag{58}
$$

and

$$
\mathbb{E}[c_{2,1}] = (d + 2)\mathrm{tr}(U_{12} + U_{11} + U_{22} + U_{21}) + \mathrm{tr}((U_{22} + U_{21})\,\Lambda). \tag{59}
$$
By Assumption 2.1, at time $t = 0$ we have $U_{12}(0), U_{11}(0), U_{22}(0)$ and $U_{21}(0)$ are PSD matrices. Therefore we have $\mathbb{E}[c_{1,1}(0)] \geq 0$ and $\mathbb{E}[c_{2,1}(0)] \geq (d + 2)\mathrm{tr}\left(U_{12}(0) + U_{11}(0) + U_{22}(0) + U_{21}(0)\right)$, which implies that

$$
\begin{aligned}
&\mathbb{E}[c_{1,1}(0) + c_{2,1}(0)]^2 \\
\geq &(d + 2)^2\left(\|\sqrt{U_{12}(0)}\|_F^2 + \|\sqrt{U_{11}(0)}\|_F^2 + \|\sqrt{U_{22}(0)}\|_F^2 + \|\sqrt{U_{21}(0)}\|_F^2\right)^2 \\
\geq &(d + 2)^2\left(\|\sqrt{U_{12}(0)}\|_F^4 + \|\sqrt{U_{11}(0)}\|_F^4 + \|\sqrt{U_{22}(0)}\|_F^4 + \|\sqrt{U_{21}(0)}\|_F^4\right) \\
\geq &(d + 2)^2\left(\|U_{12}(0)\|_F^2 + \|U_{11}(0)\|_F^2 + \|U_{22}(0)\|_F^2 + \|U_{21}(0)\|_F^2\right) \quad \text{(submultiplicativity)} \\
= &(d + 2)^2 u_{-1}(0)^2 \quad \text{(Assumption 2.1)}
\end{aligned}
$$
$$\tag{60}$$

Therefore we have $\mathbb{E}[c_{1,1}(0) + c_{2,1}(0)] \geq (d + 2)u_{-1}(0)$.

Recall that
$$
c_{1,2} = \frac{1}{2}\mathrm{tr}\left(\gamma U_{12}^\top\mu_\tau\mu_\tau^\top U_{12}\Lambda + U_{22}^\top\Gamma\left(\Lambda + \mu_\tau\mu_\tau^\top\right)U_{22}\Lambda + 2U_{12}^\top\mu_\tau\mu_\tau^\top\left(\Gamma - \frac{2}{n}\mu_\tau\mu_\tau^\top\right)U_{22}\Lambda\right)
$$
$$\tag{61}$$

and

$$c_{2,2} = \frac{1}{2} \operatorname{tr} \Big( \gamma \left( U_{12} + U_{11} \right)^\top \mu_\tau \mu_\tau^\top \left( U_{12} + U_{11} \right) \mu_\tau \mu_\tau^\top$$

$$+ 2 \left( U_{12} + U_{11} \right)^\top \mu_\tau \mu_\tau^\top \left( \Gamma - \frac{2}{n} \mu_\tau \mu_\tau^\top \right) \left( U_{22} + U_{21} \right) \mu_\tau \mu_\tau^\top \Big) \tag{62}$$

$$+ \frac{1}{2} \operatorname{tr} \Big( \left( U_{22} + U_{21} \right)^\top \Gamma \left( \Lambda + \mu_\tau \mu_\tau^\top \right) \left( U_{22} + U_{21} \right) \mu_\tau \mu_\tau^\top \Big)$$

Note that $\Gamma \to \Lambda + \mu_\tau \mu_\tau^\top$ and $\gamma \to \|\mu_\tau\|^2$ if $n \to \infty$. Therefore we have

$$\begin{aligned}
c_{2,2} \le{}& \frac{1}{2} \|\mu_\tau\|^2 \|U_{12} + U_{11}\|_F^2 \|\mu_\tau \mu_\tau^\top\|_F^2 + \frac{1}{2} \|U_{22} + U_{21}\|_F^2 \|\Lambda + \mu_\tau \mu_\tau^\top\|_F^2 \|\mu_\tau \mu_\tau^\top\|_F \\
&+ \|U_{12} + U_{11}\|_F \|U_{22} + U_{21}\|_F \|\mu_\tau \mu_\tau^\top\|_F^2 \|\Lambda + \mu_\tau \mu_\tau^\top\|_F \quad \text{(Cauchy-Schwartz inequality)} \\
\le{}& \frac{1}{2} \left\| \Lambda + \mu_\tau \mu_\tau^\top \right\|_F^2 \left\| \mu_\tau \mu_\tau^\top \right\|_F \left( \|U_{12} + U_{11}\|_F + \|U_{22} + U_{21}\|_F \right)^2 \\
\le{}& 2 \left\| \Lambda + \mu_\tau \mu_\tau^\top \right\|_F^2 \left\| \mu_\tau \mu_\tau^\top \right\|_F \left( \|U_{12}\|_F^2 + \|U_{11}\|_F^2 + \|U_{22}\|_F^2 + \|U_{21}\|_F^2 \right) \quad \text{(Triangle inequality)} \\
={}& 2 \left\| \Lambda + \mu_\tau \mu_\tau^\top \right\|_F^2 \|\mu_\tau\|^2 u_{-1}^2 \quad \text{(Lemma A.3)}
\end{aligned}$$

$$\tag{63}$$

Here the second last inequality comes from $\|\mu_\tau\|^2 = \|\mu_\tau \mu_\tau^\top\|_F \le \|\Lambda + \mu_\tau \mu_\tau^\top\|_F$.

Similarly, for $c_{1,2}$ we have

$$\begin{aligned}
c_{1,2} ={}& \frac{1}{2} \operatorname{tr} \left( \|\mu_\tau\|^2 U_{12}^\top \mu_\tau \mu_\tau^\top U_{12} \Lambda + U_{22}^\top \left( \Lambda + \mu_\tau \mu_\tau^\top \right)^2 U_{22} \Lambda + 2 U_{12}^\top \mu_\tau \mu_\tau^\top \left( \Lambda + \mu_\tau \mu_\tau^\top \right) U_{22} \Lambda \right) \\
\le{}& \frac{1}{2} \|\mu_\tau\|^2 \|U_{12}\|_F^2 \|\Lambda\|_F^2 + \frac{1}{2} \|U_{22}\|_F^2 \|\Lambda + \mu_\tau \mu_\tau^\top\|_F^2 \|\Lambda\|_F \\
&+ \|\mu_\tau \mu_\tau^\top\|_F^2 \|\Lambda\|_F \|U_{11}\|_F \|U_{12}\|_F \quad \text{(Cauchy-Schwartz inequality)} \\
\le{}& \frac{1}{2} \left\| \Lambda + \mu_\tau \mu_\tau^\top \right\|_F^2 \|\Lambda\|_F \left( \|U_{12}\|_F + \|U_{22}\|_F \right)^2 \\
\le{}& \frac{1}{2} \left\| \Lambda + \mu_\tau \mu_\tau^\top \right\|_F^2 \|\Lambda\|_F \left( \|U_{12}\|_F + \|U_{22}\|_F + \|U_{11}\|_F^2 + \|U_{21}\|_F^2 \right)^2 \\
\le{}& \left\| \Lambda + \mu_\tau \mu_\tau^\top \right\|_F^2 \|\Lambda\|_F \left( \|U_{12}\|_F^2 + \|U_{11}\|_F^2 + \|U_{22}\|_F^2 + \|U_{21}\|_F^2 \right) \quad \text{(Cauchy-Schwartz inequality)} \\
={}& \left\| \Lambda + \mu_\tau \mu_\tau^\top \right\|_F^2 \|\Lambda\|_F u_{-1}^2 \quad \text{(Lemma A.3)}
\end{aligned}$$

$$\tag{64}$$

Here the second inequality comes from $\|\mu_\tau\|^2 \le \|\Lambda + \mu_\tau \mu_\tau^\top\|_F$ and $\|\Lambda\|_F \le \|\Lambda + \mu_\tau \mu_\tau^\top\|_F$.

Adding (113) and (112) up, we have

$$c_{1,2} + c_{2,2} \le u_{-1}^2 \|\Lambda + \mu_\tau \mu_\tau^\top\|_F^2 \left( 2\|\mu_\tau\|^2 + \|\Lambda\|_F \right). \tag{65}$$

Recall that

$$c_{1,1} = \operatorname{tr} \left( \left( \mu_\tau \mu_\tau^\top U_{12} \Lambda + \left( \Lambda + \mu_\tau \mu_\tau^\top \right) U_{22} \Lambda \right) \right) \tag{66}$$

and

$$c_{2,1} = \operatorname{tr} \left( \mu_\tau \mu_\tau^\top \left( U_{12} + U_{11} \right) \mu_\tau \mu_\tau^\top + \left( \Lambda + \mu_\tau \mu_\tau^\top \right) \left( U_{22} + U_{21} \right) \mu_\tau \mu_\tau^\top \right). \tag{67}$$

We have

$$\begin{aligned}
c_{1,1} ={}& \operatorname{tr} \left( \left( \mu_\tau \mu_\tau^\top U_{12} \Lambda + \left( \Lambda + \mu_\tau \mu_\tau^\top \right) U_{22} \Lambda \right) \right) \\
\le{}& \|\mu_\tau \mu_\tau^\top\|_F \|\Lambda\|_F \|U_{12}\|_F + \|\Lambda + \mu_\tau \mu_\tau^\top\|_F \|\Lambda\|_F \|U_{22}\|_F \quad \text{(Cauchy-Schwartz inequality)} \\
\le{}& \left( \|U_{12}\|_F + \|U_{22}\|_F \right) \left\| \Lambda + \mu_\tau \mu_\tau^\top \right\|_F^2 \\
\le{}& \sqrt{2 \left( \|U_{12}\|_F^2 + \|U_{22}\|_F^2 \right)} \left\| \Lambda + \mu_\tau \mu_\tau^\top \right\|_F^2 \quad \text{(Cauchy-Schwartz inequality)} \\
\le{}& \sqrt{2 \left( \|U_{12}\|_F^2 + \|U_{22}\|_F^2 + \|U_{11}\|_F^2 + \|U_{21}\|_F^2 \right)} \left\| \Lambda + \mu_\tau \mu_\tau^\top \right\|_F^2 \\
={}& \sqrt{2} u_{-1} \left\| \Lambda + \mu_\tau \mu_\tau^\top \right\|_F^2 \quad \text{(Lemma A.3)}.
\end{aligned}$$

$$\tag{68}$$

Here the second inequality comes from $\|\mu_\tau \mu_\tau^\top\|_F \leq \|\Lambda + \mu_\tau \mu_\tau^\top\|_F$ and $\|\Lambda\|_F \leq \|\Lambda + \mu_\tau \mu_\tau^\top\|_F$.

Similarly we have

$$
\begin{aligned}
c_{2,1} &= \mathrm{tr}\left(\mu_\tau \mu_\tau^\top (U_{12} + U_{11}) \mu_\tau \mu_\tau^\top + \left(\Lambda + \mu_\tau \mu_\tau^\top\right) (U_{22} + U_{21}) \mu_\tau \mu_\tau^\top\right) \\
&\leq \|\mu_\tau \mu_\tau^\top\|_F^2 \|U_{12} + U_{11}\|_F + \|\Lambda + \mu_\tau \mu_\tau^\top\|_F \|\mu_\tau \mu_\tau^\top\|_F \|U_{22} + U_{21}\|_F \quad \text{(Cauchy-Schwartz inequality)} \\
&\leq \left(\|U_{12} + U_{11}\|_F + \|U_{22} + U_{21}\|_F\right) \left\|\Lambda + \mu_\tau \mu_\tau^\top\right\|_F^2 \\
&\leq \left(\|U_{12}\|_F + \|U_{11}\|_F + \|U_{22}\|_F + U_{21}\|_F\right) \left\|\Lambda + \mu_\tau \mu_\tau^\top\right\|_F^2 \quad \text{(Triangle inequality)} \\
&\leq 2\sqrt{\|U_{12}\|_F^2 + \|U_{22}\|_F^2 + \|U_{11}\|_F^2 + \|U_{21}\|_F^2} \left\|\Lambda + \mu_\tau \mu_\tau^\top\right\|_F^2 \quad \text{(Cauchy-Schwartz inequality)} \\
&= 2u_{-1} \left\|\Lambda + \mu_\tau \mu_\tau^\top\right\|_F^2. \quad \text{(Lemma A.3)}
\end{aligned}
\tag{69}
$$

Here the second inequality comes from $\|\mu_\tau \mu_\tau^\top\|_F \leq \|\Lambda + \mu_\tau \mu_\tau^\top\|_F$.

Adding (68) and (69) up, we have

$$
c_{1,1} + c_{2,1} \leq (2 + \sqrt{2}) u_{-1} \|\Lambda + \mu_\tau \mu_\tau^\top\|_F^2.
\tag{70}
$$

$\square$

Finally we prove the gradient lower bound of $\|\nabla L(U)\|_F$. The high-level idea of this proof is to expand $L_1$ and $L_2$ under new parametrizations $\widetilde{U} = (u_{-1} U_{12}, u_{-1} U_{22})$ and $\hat{U} = (u_{-1}\mathrm{Sym}\left(U_{11} + U_{12} + U_{21} + U_{22}\right), u_{-1}\left(U_{12} + U_{22}\right))$ respectively, where the two terms admit gradient lower bounds respectively. We can merge these two gradient lower bounds by merging these two parametrizations into one parametrization $\Theta = (\widetilde{U}, \hat{U})$ to obtain a lower bound on $\|\nabla L(\Theta)\|_F$. We then write out the Jacobian of the parameters transformation $(u_{-1} U_{12}^\top, u_{-1} U_{22}^\top, u_{-1} U_{21}^\top, u_{-1} U_{11}^\top)^\top \to \Theta$ and find its spectral norm is bounded by some constant. Therefore, by combining the lower bound of $u_{-1}$ we obtained in Lemma A.4, we can translate the gradient lower bound of $\|\nabla L(\Theta)\|_F$ back to $\|\nabla U\|_F$ up to some constant. Specifically, we use the following lemma to relate the gradient norm bounds after reparametrization.

**Lemma A.6** (Gradient lowerbound with reparametrization). *Suppose $f : \mathbb{R}^p \to \mathbb{R}^p$ and $g : \mathbb{R}^p \to \mathbb{R}$ are differentiable functions. Denote $y = f(x)$ where $f$ is a transformation between two parametrizations $y$ and $x$. Suppose $g(y)$ has a gradient lower bound $\|\nabla_y g(y)\| \geq \delta(y)$. Denote $J_f(x) \in \mathbb{R}^{p \times p}$ the Jacobian matrix of $f$ at $x$. If $J_f(x)$ is invertible and the spectral norm of the inverse $\|J_f^{-1}(x)\|_2 \leq a$ for some constant $a$, then we have $\|\nabla_x g(f(x))\| \geq a^{-1}\delta(y)$ for all $y = f(x)$.*

*Proof of Lemma A.6.* Easy to see through chain rule that $\nabla_x g(f(x)) = J_f(x)\nabla_y g(y)$. Since $J_f(x)$ is invertible, we obtain $\delta(y) \leq \|\nabla_y g(y)\| = \|J_f(x)^{-1}\nabla_x g(f(x))\| \leq a\|\nabla_x g(f(x))\|$, which completes the proof. $\square$

With the above lemma in hand, we can prove Lemma A.5.

*Proof of Lemma A.5.* We first develope a gradient lower bound on $L_1$.

**Step 1: gradient lower bound on $L_1$.** We take a new parameterization $\widetilde{U}_{12} := u_{-1} U_{12}$ and $\widetilde{U}_{22} := u_{-1} U_{22}$. Denote $L_1(U) = \mathbb{E}[\ell_1(U, \tau)]$ and $L_2(U) = \mathbb{E}[\ell_2(U, \tau)]$. Then we can simply $L_1(U)$ in the new parameterization $\widetilde{U} = (\widetilde{U}_{12}, \widetilde{U}_{22})$ as $L_1(\widetilde{U})$. Specifically, we have

$$
\begin{aligned}
L_1(U) &= L_1(\widetilde{U}) \\
&= \frac{1}{2}\mathbb{E}\left[\mathrm{tr}\left(\|\mu_\tau\|^2 \widetilde{U}_{12}^\top \mu_\tau \mu_\tau^\top \widetilde{U}_{12}\Lambda + \widetilde{U}_{22}^\top \left(\Lambda + \mu_\tau \mu_\tau^\top\right) \widetilde{U}_{22}\Lambda + \Lambda \right.\right. \\
&\qquad \left.\left. + 2\widetilde{U}_{12}^\top \mu_\tau \mu_\tau^\top \left(\Lambda + \mu_\tau \mu_\tau^\top\right) \widetilde{U}_{22}\Lambda - 2\mu_\tau \mu_\tau^\top \widetilde{U}_{12}\Lambda - 2\left(\Lambda + \mu_\tau \mu_\tau^\top\right) \widetilde{U}_{22}\Lambda\right)\right].
\end{aligned}
\tag{71}
$$

First we want to show that

$$
\left\|\nabla L_1(\widetilde{U})\right\|_F \geq \frac{1}{10}\lambda_{\min}(\Lambda) \min\{\lambda_{\min}(\Lambda)^3, 1\}\|\widetilde{U} - \widetilde{U}_*\|_F
\tag{72}
$$

for all $t \geq 0$, where $\widetilde{U}_* := (-\Lambda^{-1}, \Lambda^{-1})$.

By direct computation, we have the gradients

- $\frac{\partial L_1(\widetilde{U})}{\partial \widetilde{U}_{12}} = \left( (d+2)\widetilde{U}_{12} + (d+2)\widetilde{U}_{22} + \Lambda\widetilde{U}_{22} - I \right)\Lambda$;

- $\frac{\partial L_1(\widetilde{U})}{\partial \widetilde{U}_{22}} = \left( (d+2)\widetilde{U}_{22} + \left( \Lambda^2 + 2\Lambda \right)\widetilde{U}_{22} + (d+2)\widetilde{U}_{12} + \Lambda\widetilde{U}_{12} - I - \Lambda \right)\Lambda$.

Therefore we have

$$
\begin{aligned}
&\|\nabla L_1(\widetilde{U})\|_F \cdot \|\widetilde{U} - \widetilde{U}_*\|_F \\
\geq& \left\langle \left( \frac{\partial L_1(\widetilde{U})}{\partial \widetilde{U}_{12}}, \frac{\partial L_1(\widetilde{U})}{\partial \widetilde{U}_{22}} \right), \left( \widetilde{U}_{12} + \Lambda^{-1}, \widetilde{U}_{22} - \Lambda^{-1} \right) \right\rangle \\
\geq& \operatorname{tr}\left( \Lambda \left( 2\left( \widetilde{U}_{12} + \widetilde{U}_{22} \right)^\top \left( \widetilde{U}_{12} + \widetilde{U}_{22} \right) + \left( \widetilde{U}_{22} - \Lambda^{-1} \right)\Lambda\left( \widetilde{U}_{12}^\top + \Lambda^{-1} \right) \right. \right. \\
&\left. \left. + (\Lambda + 2I)\left( \widetilde{U}_{22} - \Lambda^{-1} \right)\Lambda\left( \widetilde{U}_{22}^\top - \Lambda^{-1} \right) + \left( \widetilde{U}_{12} + \Lambda^{-1} \right)\Lambda\left( \widetilde{U}_{22}^\top - \Lambda^{-1} \right) \right) \right) \\
=& \operatorname{tr}\left( \Lambda \left( VV^\top + \left( \widetilde{U}_{12} + \Lambda^{-1} \right) \underbrace{\frac{1}{2}\Lambda^2\left( \Lambda^2 + 2\Lambda + 2I \right)^{-1}}_{P_1} \left( \widetilde{U}_{12} + \Lambda^{-1} \right)^\top \right. \right. \\
&\left. \left. + \left( \widetilde{U}_{22} - \Lambda^{-1} \right)\underbrace{\left( \Lambda^2 + 2\Lambda + 2I - (\Lambda + 2I)^2\left( 2I - \frac{1}{2}\Lambda^2\left( \Lambda^2 + 2\Lambda + 2I \right)^{-1} \right) \right)}_{P_2} \right) \right).
\end{aligned}
\tag{73}
$$

In the last equation the matrix $V$ is defined as

$$
\begin{aligned}
V :=& \left( \tilde{U}_{12} + \Lambda^{-1} \right)\left( 2I - \frac{1}{2}\Lambda^2\left( \Lambda^2 + 2\Lambda + 2I \right)^{-1} \right)^{\frac{1}{2}} \\
&+ \left( \widetilde{U}_{22} - \Lambda^{-1} \right)(\Lambda + 2I)\left( 2I - \frac{1}{2}\Lambda^2\left( \Lambda^2 + 2 + 2I \right)^{-1} \right)^{-\frac{1}{2}}.
\end{aligned}
\tag{74}
$$

It is easy to see $P_1$ is a PSD matrix. Actually $P_2$ is also PSD. To see this, for any eigenvalue $a$ of $\Lambda$, the corresponding eigenvalue in $P_2$ is $a^2 + 2a + 2 - \frac{2\left( a^2 + 2a + 4 \right)\left( a^2 + 2a + 2 \right)}{3a^2 + 8a + 8} \geq \frac{1}{4}a^2 \geq 0$. Furthermore we have $\lambda_{\min}(P_2) \geq \frac{1}{4}\lambda_{\min}(\Lambda)^2$. Similarly, we have $\lambda_{\min}(P_1) \geq \frac{1}{10}\min\{\lambda_{\min}(\Lambda)^3, 1\}$. Removing the term containing $V$ in (73), we have

$$
\begin{aligned}
&\|\nabla L_1(\widetilde{U})\|_F \cdot \|\widetilde{U} - \widetilde{U}_*\|_F \\
\geq& \operatorname{tr}\left( \Lambda\left( \widetilde{U}_{12} + \Lambda^{-1} \right)P_1\left( \widetilde{U}_{12} + \Lambda^{-1} \right)^\top + \Lambda\left( \widetilde{U}_{22} - \Lambda^{-1} \right)P_2\left( \widetilde{U}_{22} - \Lambda^{-1} \right)^\top \right) \\
\geq& \frac{1}{10}\lambda_{\min}(\Lambda)\min\{\lambda_{\min}(\Lambda)^3, 1\}\|\widetilde{U}_{12} + \Lambda^{-1}\|_F^2 + \frac{1}{4}\lambda_{\min}(\Lambda)^2\|\widetilde{U}_{22} - \Lambda^{-1}\|_F^2 \\
\geq& \frac{1}{10}\lambda_{\min}(\Lambda)\min\{\lambda_{\min}(\Lambda)^3, 1\}\left( \|\widetilde{U}_{12} + \Lambda^{-1}\|_F^2 + \|\widetilde{U}_{22} - \Lambda^{-1}\|_F^2 \right) \\
=& \frac{1}{10}\lambda_{\min}(\Lambda)\min\{\lambda_{\min}(\Lambda)^3, 1\}\|\widetilde{U} - \widetilde{U}_*\|_F^2.
\end{aligned}
\tag{75}
$$

Therefore we have

$$
\|\nabla L_1(\widetilde{U})\|_F \geq \frac{1}{10}\lambda_{\min}(\Lambda)\min\{\lambda_{\min}(\Lambda)^3, 1\}\|\widetilde{U} - \widetilde{U}_*\|_F.
\tag{76}
$$

**Step 2: gradient lower bound on $L_2$.** Now we derive a gradient lower bound for $L_2$. We define a new parameterization $U_1 := u_{-1}\mathrm{Sym}(U_{11} + U_{12} + U_{21} + U_{22})$ and $U_2 := u_{-1}(U_{21} + U_{22})$. By checking the dynamics of $U_{11}$ in (35) and (37), we can find that $U_{11}$ keeps symmetric for all $t \geq 0$. Therefore $U_1 = u_{-1}(U_{11} + \mathrm{Sym}(U_{12} + U_{21} + U_{22}))$. Note that for any $d \times d$ matrix $A$, it holds that $\mu_\tau^\top A \mu_\tau = \mu_\tau^\top \mathrm{Sym}(A)\mu_\tau$. Therefore we can write $L_2$ under the new parameterization as $L_2(U_1, U_2)$:

$$L_2(U) = L_2(U_1, U_2) = \frac{1}{2}\mathbb{E}\left[\left\|\left(\Lambda U_2 - I + \mu_\tau \mu_\tau^\top U_1\right)\mu_\tau\right\|^2\right]. \tag{77}$$

Therefore we know $L_2$ is convex in terms of $U_1$ and $U_2$ since $\Lambda U_2 - I + \mu_\tau \mu_\tau^\top U_1$ is an affine function of $(U_1, U_2)$, $f(X) = \|X\mu_\tau\|^2$ is a convex function and the expectation reserves convexity. By setting $U_2^* = \Lambda^{-1}$ and $U_1^* = 0_{d \times d}$ in (77), we have $L_2(U_1^*, U_2^*) = 0$. Denote $\hat{U} = (U_1, U_2)$, $\hat{U}_* = (U_1^*, U_2^*)$.

By convexity, we have

$$\left\langle \nabla L_2(\hat{U}), \hat{U} - \hat{U}_* \right\rangle + L_2(U_1^*, U_2^*) = \left\langle \nabla L_2(\hat{U}), \hat{U} - \hat{U}_* \right\rangle \geq L_2(\hat{U}). \tag{78}$$

Therefore expanding the expectation in (77), we have

$$
\begin{aligned}
L_2(\hat{U}) &= \|\Lambda U_2 - I\|_F^2 + (d+4)\left(\mathrm{tr}\left(U_1\right)^2 + \mathrm{tr}\left(U_1^2\right) + \|U_1\|_F^2\right) \\
&\quad + 2\left(\mathrm{tr}\left((\Lambda U_2 - I)(U_1)\right) + \mathrm{tr}\left((\Lambda U_2 - I)(U_1)^\top\right) + \mathrm{tr}\left(\Lambda U_2 - I\right)\mathrm{tr}\left(U_1\right)\right) \\
&\geq \left\|\frac{1}{\sqrt{d+4}}\left(\Lambda U_2 + U_2^\top \Lambda - 2I\right) + \sqrt{d+4}\left(U_1\right)\right\|_F^2 + \frac{d}{d+4}\|\Lambda U_2 - I\|_F^2 + (d+4)\|U_1\|_F^2 \\
&\quad + (d+4)\mathrm{tr}\left(U_1\right)^2 + 2\,\mathrm{tr}\left(\Lambda U_2 - I\right)\mathrm{tr}\left(U_1\right) \\
&\geq \left\|\frac{1}{\sqrt{d+4}}\left(\Lambda U_2 + U_2^\top \Lambda - 2I\right) + \sqrt{d+4}\left(U_1\right)\right\|_F^2 \\
&\quad + \left(\frac{d}{d+4} - \frac{d}{d+5}\right)\|\Lambda U_2 - I\|_F^2 + \frac{1}{d+5}\mathrm{tr}\left(\Lambda U_2 - I\right)^2 \\
&\quad + 4\|U_1\|_F^2 + (d+5)\mathrm{tr}\left(U_1\right)^2 + 2\,\mathrm{tr}\left(\Lambda U_2 - I\right)\mathrm{tr}\left(U_1\right) \\
&\quad + \left(\frac{1}{\sqrt{d+5}}\mathrm{tr}\left(\Lambda U_2 - I\right) + \sqrt{d+5}\,\mathrm{tr}\left(U_1\right)\right)^2 \\
&\geq \left(\frac{d}{d+4} - \frac{d}{d+5}\right)\|\Lambda U_2 - I\|_F^2 + 4\|U_1\|_F^2 \\
&\geq \frac{1}{30d}\left(\|\Lambda U_2 - I\|_F^2 + \|U_1\|_F^2\right) \\
&\geq \frac{\min\left(\lambda_{\min}(\Lambda), 1\right)^2}{30d}\left(\|U_1\|_F^2 + \|U_2 - \Lambda^{-1}\|_F^2\right) \\
&= \frac{\min\left(\lambda_{\min}(\Lambda), 1\right)^2}{30d}\|\hat{U} - \hat{U}_*\|_F^2.
\end{aligned}
\tag{79}
$$

Here the first equation comes from that $\|A\|_F^2 \geq \mathrm{tr}(A^2)$ by taking $A = \Lambda U_2 - I$. The second equation comes from that $U_1$ is symmetric hence $\mathrm{tr}\left(U_1^2\right) = \|U_1\|_F^2$. The first inequality comes from that $\|A\|_F^2 \geq \frac{1}{d}\mathrm{tr}(A)^2$ for any $d \times d$ real matrix $A$ by taking $A = \Lambda U_2 - I$ and $A = U_1$. The last inequality comes from that

$$\left\|\begin{pmatrix} U_2 - \Lambda^{-1} \\ U_1 \end{pmatrix}\right\|_F = \left\|\begin{pmatrix} \Lambda^{-1} & 0 \\ 0 & I \end{pmatrix}\begin{pmatrix} \Lambda U_2 - I \\ U_1 \end{pmatrix}\right\|_F \leq \frac{1}{\min\left(\lambda_{\min}(\Lambda), 1\right)}\left\|\begin{pmatrix} \Lambda U_2 - I \\ U_1 \end{pmatrix}\right\|_F.$$

Combining (78) by Cauchy-Schwartz inequality we have

$$\|\nabla L_2(\hat{U})\|_F \|\hat{U} - \hat{U}_*\|_F \geq \left\langle \nabla L_2(\hat{U}), \hat{U} - \hat{U}_* \right\rangle \geq L_2(\hat{U}) \geq \frac{\lambda_{\min}(\Lambda)^2}{30d}\|\hat{U} - \hat{U}_*\|_F^2, \tag{80}$$

which yields that

$$\|\nabla L_2(\hat{U})\|_F \geq \frac{\min\left(\lambda_{\min}(\Lambda), 1\right)^2}{30d}\|\hat{U} - \hat{U}_*\|_F. \tag{81}$$

**Step 3: merge two lower bounds and parametrizations.** Combine two types of parameterizations to get $\Theta = (\widetilde{U}_{12}^\top, \widetilde{U}_{22}^\top, U_2^\top, U_1^\top)^\top$. Let $\mathrm{Vec}(A)$ be the vectorization operator in row-wise order. For example, $\mathrm{Vec}\begin{pmatrix} 1 & 2 \\ 3 & 4 \end{pmatrix} = (1, 2, 3, 4)^\top$. We now give a gradient lower bound of $\|\nabla L(\mathrm{Vec}(\Theta))\|_F$ by adding two gradient lower bounds (76) and (81):

$$\begin{aligned}
&\|\nabla L(\mathrm{Vec}(\Theta))\|_F^2 \\
=&\|\nabla L(\Theta)\|_F^2 \\
=&\|\nabla L_2(\hat{U})\|_F^2 + \|\nabla L_1(\widetilde{U})\|_F^2 \\
\geq& c'\left(\|U_1\|_F^2 + \left\|U_2 - \Lambda^{-1}\right\|_F^2 + \|\widetilde{U}_{12} + \Lambda^{-1}\|_F^2 + \|\widetilde{U}_{22} - \Lambda^{-1}\|_F^2\right).
\end{aligned} \tag{82}$$

where $c' := \min\left(\frac{\lambda_{\min}(\Lambda)^2}{30d}, \frac{1}{30d}, \frac{\lambda_{\min}(\Lambda)^4}{10}, \frac{1}{10}\right)$.

**Step 4: Translate the gradient lower bound to $U$-parametrization.** We define $W = (u_{-1}U_{12}^\top, u_{-1}U_{22}^\top, u_{-1}U_{21}^\top, u_{-1}U_{11}^\top)^\top$. Then we have

$$\mathrm{Vec}(\Theta) = \begin{pmatrix} I_{d^2} & & & \\ & I_{d^2} & & \\ & & I_{d^2} & \\ \frac{1}{2}(I_{d^2} + T) & \frac{1}{2}(I_{d^2} + T) & \frac{1}{2}(I_{d^2} + T) & I_{d^2} \end{pmatrix} \mathrm{Vec}(W) =: J\,\mathrm{Vec}(W). \tag{83}$$

Here $T \in \mathbb{R}^{d^2 \times d^2}$ is the transpose operator. That is $T\,\mathrm{Vec}(A) = \mathrm{Vec}(A^\top)$ for any $d \times d$ matrix $A$. It is easy to see $T$ has exactly a 1 and $d^2 - 1$ of zero in each row. So we have $\|I_{d^2} + T\|_F^2 \leq 4d^2$. We note that $J$ is invertible and

$$J^{-1} = \begin{pmatrix} I_{d^2} & & & \\ & I_{d^2} & & \\ & -I_{d^2} & I_{d^2} & \\ \frac{1}{2}(I_{d^2} + T) & -\frac{1}{2}(I_{d^2} + T) & -\frac{1}{2}(I_{d^2} + T) & I_{d^2} \end{pmatrix}.$$

Therefore we have $\|J^{-1}\|_F^2 \leq 9d^2$, which yields a upper bound of spectral norm $\|J^{-1}\|_2 \leq 3d$. Now we can invoke Lemma A.6 and obtain

$$\begin{aligned}
&\|\nabla L(W)\|_F^2 \\
=&\|\nabla L(\mathrm{Vec}(W))\|_F^2 \\
\geq& \frac{c'}{9d^2}\left(\|U_1\|_F^2 + \left\|U_2 - \Lambda^{-1}\right\|_F^2 + \|\widetilde{U}_{12} + \Lambda^{-1}\|_F^2 + \|\widetilde{U}_{22} - \Lambda^{-1}\|_F^2\right).
\end{aligned} \tag{84}$$

Finally we translate the gradient lower bound back to $U$:

$$\begin{aligned}
&\|\nabla L(U)\|_F^2 \\
\geq& \|\nabla_{U_{11}} L(U)\|_F^2 + \|\nabla_{U_{12}} L(U)\|_F^2 + \|\nabla_{U_{21}} L(U)\|_F^2 + \|\nabla_{U_{22}} L(U)\|_F^2 \\
=& u_{-1}^2\|\nabla L(W)\|_F^2 \\
\geq& \beta^2\|\nabla L(W)\|_F^2 \\
\geq& c\left(\|U_1\|_F^2 + \left\|U_2 - \Lambda^{-1}\right\|_F^2 + \|\widetilde{U}_{12} + \Lambda^{-1}\|_F^2 + \|\widetilde{U}_{22} - \Lambda^{-1}\|_F^2\right).
\end{aligned} \tag{85}$$

Here the second inequality comes from the lower bound $u_{-1} \geq \beta$ in Lemma A.4. $\qquad\square$

## A.2 FINITE SAMPLE PROOF

Now we consider the general setting where $n$ is finite. We first give a global convergence result, of which the proof follows the same framework as infinite samples. There are two differences in the proof: (i) in the infinite-sample case, our converged solution has a population loss of 0, which is automatically the optimal, while the converged loss is positive when $n$ is finite. Hence we need a new Lemma (Lemma A.10) which shows that fixing $u_{13}, u_{23}, u_{31}, u_{32}$ to be zero does not lose optimality; (ii) as we will see later in Theorem A.8, the optimal solution have a much more complicated form when $n$ is finite. Therefore we use the form of gradient lower bound which does not explicitly involve the form of the optimal solution as in Lemma A.11.

**Theorem A.7** (Finite-sample convergence). *Under Assumption 2.1, if the number of samples $n \geq 2$ and $\sigma$ satisfies $0 < \sigma < \alpha$ for some constant $\alpha$, then the gradient flow (8) will converge to a global minimizer $W_* = (W_*^{KQ}, W_*^{PV})$.*

To give a characterization of the global minimizer when $\Lambda = I_d$, we have the following theorem.

**Theorem A.8** (Global minimizer). *If the covariance matrix $\Lambda = I_d$, then the global minimizer $W_* = (W_*^{KQ}, W_*^{PV})$ in Theorem A.7 satisfies*

$$W_*^{KQ} = \begin{pmatrix} a_{11}I_d & a_{12}I_d & 0_d \\ a_{21}I_d & a_{22}I_d & 0_d \\ 0_d^\top & 0_d^\top & 0 \end{pmatrix}, W_*^{PV} = \begin{pmatrix} 0_{d\times d} & 0_{d\times d} & 0_d \\ 0_{d\times d} & 0_{d\times d} & 0_d \\ 0_d^\top & 0_d^\top & b \end{pmatrix}. \tag{86}$$

*where*

$$b^4 = \left( \frac{nd + 2n - n^2}{n^2 + nd^2 + 5dn + 4n + d^2 + d - 1} - \frac{(n - n^2)(d + 1)}{n^2(d + 1) + 2d^2n + 6dn + 2d^2 + d - 1} \right)^2$$

$$+ \left( \frac{n(n - 1)}{n^2 + nd^2 + 5dn + 4n + d^2 + d - 1} + \frac{n^2(d + 1) + n(d - 1)}{n^2(d + 1) + 2d^2n + 6dn + 2d^2 + d - 1} \right)^2$$

$$+ \left( \frac{(n - n^2)(d + 1)}{n^2(d + 1) + 2d^2n + 6dn + 2d^2 + d - 1} \right)^2 + \left( \frac{n^2(d + 1) + n(d - 1)}{n^2(d + 1) + 2d^2n + 6dn + 2d^2 + d - 1} \right)^2$$

$$\tag{87}$$

*and*

$$\begin{cases} a_{11} = \frac{1}{b} \left( \frac{nd + 2n - n^2}{n^2 + nd^2 + 5dn + 4n + d^2 + d - 1} - \frac{(n - n^2)(d + 1)}{n^2(d + 1) + 2d^2n + 6dn + 2d^2 + d - 1} \right) \\ a_{21} = \frac{1}{b} \left( \frac{n(n - 1)}{n^2 + nd^2 + 5dn + 4n + d^2 + d - 1} - \frac{n^2(d + 1) + n(d - 1)}{n^2(d + 1) + 2d^2n + 6dn + 2d^2 + d - 1} \right) \\ a_{12} = \frac{1}{b} \left( \frac{(n - n^2)(d + 1)}{n^2(d + 1) + 2d^2n + 6dn + 2d^2 + d - 1} \right) \\ a_{22} = \frac{1}{b} \left( \frac{n^2(d + 1) + n(d - 1)}{n^2(d + 1) + 2d^2n + 6dn + 2d^2 + d - 1} \right). \end{cases} \tag{88}$$

**Remark A.9** (Non-zero $a_{11}$ and $a_{21}$ to reduce bias under finite samples). *As we see in (18), the Transformer uses some $a_{12}$ and $a_{22}$ to minimize the variance term $L_1$. In the infinite-sample case, these $a_{12}$ and $a_{22}$ also minimize the bias term $L_2$ to be 0. However, when $n$ is finite, these $a_{12}$ and $a_{22}$ fail to minimize the bias term as $\hat{Z}$ deviates from $I_d$ and $\hat{z}$ deviates from $0_d$. Thus the Transformer uses some non-zero $a_{11}$ and $a_{21}$ to better estimate $\mu_\tau$ which in turn reduces the bias term $L_2$.*

Theorem A.8 can be simply proved by

1. Computing the critical point of the reparametrized optimization problem;
2. Translate the critical point condition back to the original parametrization along with the balancing condition.

### A.2.1 PROOF OF THEOREM A.7

Denote $z_i = x_{\tau,i} - \mu_\tau$, $z_{\text{query}} = x_{\tau,\text{query}} - \mu_\tau$, $\hat{z} = \frac{1}{n} \sum_{i=1}^n z_i$, and $\hat{Z} = \frac{1}{n} \sum_{i=1}^n z_i z_i^\top$ the deviation vectors, the empirical deviation vector and the empirical covariance matrix of the in-context samples.

Now we sketch the proof of Theorem A.7.

**Decompose the loss.** Taking expectation over $z_{\text{query}}$ and decomposing the result into bias and variance terms, we have

$$
L(U) = \underbrace{\frac{1}{2}\mathbb{E}_{z_i,\mu_\tau,w_\tau}\left[\left(\mathbb{E}_{z_{\text{query}}}\left[\hat{y}_{\tau,\text{query}} - y_{\tau,\text{query}}\right]\right)^2\right]}_{L_2}
$$
$$
+ \underbrace{\frac{1}{2}\mathbb{E}_{z_i,\mu_\tau,w_\tau}\left[\left(\mathbb{E}_{z_{\text{query}}}[(\hat{y}_{\tau,\text{query}} - y_{\tau,\text{query}})^2] - \left(\mathbb{E}_{z_{\text{query}}}\left[\hat{y}_{\tau,\text{query}} - y_{\tau,\text{query}}\right]\right)^2\right)\right]}_{L_1}.
\tag{89}
$$

**Simplify the loss in the new parametrization.** Noting that Lemma A.3 tells us that $u_{13} = u_{23} = u_{31} = u_{32} = 0_d$ through the training regardless of how large $n$ is, we can simplify the prediction to be

$$
\hat{y}_{\tau,\text{query}} = \mu_\tau^\top w_\tau^\top \hat{\mu}_\tau \tilde{U}_{11}\mu_\tau + w_\tau^\top \hat{\Lambda}\tilde{U}_{21}\mu_\tau + \mu_\tau^\top w_\tau^\top \hat{\mu}_\tau \tilde{U}_{12}x_{\tau,\text{query}} + w_\tau^\top \hat{\Lambda}\tilde{U}_{22}x_{\tau,\text{query}}
\tag{90}
$$

where $\tilde{U} := u_{-1}U$. Therefore, by algebra we have

$$
L_2(\tilde{U}) = \frac{1}{2}\mathbb{E}_{z_i,\mu_\tau}\left[\left\|\left((\mu_\tau + \hat{z})\mu_\tau^\top\left(\tilde{U}_{11} + \tilde{U}_{12} + \tilde{U}_{21} + \tilde{U}_{22}\right)\mu_\tau + \left(\mu_\tau \hat{z}^\top + \hat{Z}\right)\left(\tilde{U}_{21} + \tilde{U}_{22}\right)\mu_\tau - \mu_\tau\right)\right\|^2\right]
\tag{91}
$$

and

$$
L_1(\tilde{U}) = \frac{1}{2}\mathbb{E}_{z_i,\mu_\tau}\left[\left\|(\mu_\tau + \hat{z})\mu_\tau^\top\left(\tilde{U}_{12} + \tilde{U}_{22}\right)\sqrt{\Lambda} + \left(\mu_\tau \hat{z}^\top + \hat{Z}\right)\tilde{U}_{22}\sqrt{\Lambda} - \sqrt{\Lambda}\right\|_F^2\right].
\tag{92}
$$

Note that for any skew-symmetric matrix $A$ and vector $x$ we have $x^\top A x = 0$. Therefore by defining $\text{Sym}(A) = \frac{1}{2}(A + A^\top)$, we can further simplify the loss function to be

$$
L(\tilde{U})
$$
$$
= \underbrace{\frac{1}{2}\mathbb{E}_{z_i,\mu_\tau}\left[\left\|(\mu_\tau + \hat{z})\mu_\tau^\top \text{Sym}\left(\tilde{U}_{11} + \tilde{U}_{12} + \tilde{U}_{21} + \tilde{U}_{22}\right)\mu_\tau + \left(\mu_\tau \hat{z}^\top + \hat{Z}\right)\left(\tilde{U}_{21} + \tilde{U}_{22}\right)\mu_\tau - \mu_\tau\right\|^2\right]}_{L_2}
$$
$$
+ \underbrace{\frac{1}{2}\mathbb{E}_{z_i,\mu_\tau}\left[\left\|(\mu_\tau + \hat{z})\mu_\tau^\top\left(\tilde{U}_{12} + \tilde{U}_{22}\right)\sqrt{\Lambda} + \left(\mu_\tau \hat{z}^\top + \hat{Z}\right)\tilde{U}_{22}\sqrt{\Lambda} - \sqrt{\Lambda}\right\|_F^2\right]}_{L_1}.
$$
$$
\tag{93}
$$

Here we give a lemma ensuring that restricting $u_{13}, u_{23}, u_{31}$ and $u_{32}$ to be $0_d$ does not affect the global minimum of the loss.

**Lemma A.10.** *Denote $S := \{U \in \mathbb{R}^{(2d+1)\times(2d+1)} : u_{13} = u_{23} = u_{31} = u_{32} = 0_d\}$ the set of simplified parameters. We have the global minimum of loss $L$ in $S$ equal to the original global minimum:*

$$
\min_{U \in S} L(U) = \min_{U \in \mathbb{R}^{(2d+1)\times(2d+1)}} L(U).
$$

Now we introduce the new parametrization $\Theta$:

$$
\begin{cases}
\Theta_1 = \text{Sym}(\tilde{U}_{11} + \tilde{U}_{12} + \tilde{U}_{21} + \tilde{U}_{22}) \\
\Theta_2 = \tilde{U}_{21} + \tilde{U}_{22} \\
\Theta_3 = \tilde{U}_{12} + \tilde{U}_{22} \\
\Theta_4 = \tilde{U}_{22}.
\end{cases}
\tag{94}
$$

Therefore $L_1$ is a function of $\Theta_3, \Theta_4$ and $L_2$ is a function of $\Theta_1, \Theta_2$. Denote the new parameter space $\mathcal{P} = \{(\Theta_1, \Theta_2, \Theta_3, \Theta_4) : \Theta_i \in \mathbb{R}^{d \times d} \text{ for } 1 \leq i \leq 4 \text{ and } \Theta_1 \text{ is symmetric}\}$. The following lemma shows that the loss $L(\Theta)$ is actually strongly-convex as a function in the convex parameter space $\mathcal{P}$.

**Lemma A.11.** *If the number of samples $n \geq 2$, there exists constant $c > 0$, such that for any $\Theta', \Theta'' \in \mathcal{P}$, we have*

$$\|\nabla L(\Theta') - \nabla L(\Theta'')\|_F \geq c\|\Theta' - \Theta''\|_F. \tag{95}$$

We can prove $u_{-1}$ could be lower bounded by some positive constant during the training phase in Lemma A.12, which suggests the trajectory of $u_{-1}$ is away from the saddle point $u_{-1} = 0$.

**Lemma A.12.** *If the initialization of one-layer LSA Transformer satisfies Assumption 2.1 and $0 < \sigma < \alpha$ where $\alpha^2$ is equal to*

$$\frac{d+2}{(1+\frac{1}{n})\|\Lambda\|_F^3 + (4 + \frac{d+4+\mathrm{tr}(\Lambda)}{n})\|\Lambda\|_F^2 + (2d+4+\frac{3d+6+3\mathrm{tr}(\Lambda)}{2n})\|\Lambda\|_F + 4(d+4+\frac{\mathrm{tr}(\Lambda)}{n})(d+2)}, \tag{96}$$

*we have*

$$u_{-1} \geq \sigma \left( \frac{\alpha}{(2+\sqrt{2})(\|\Lambda\|_F^2 + 2\mathrm{tr}(\Lambda) + d^2 + 2d)} \right)^{\frac{1}{2}} \tag{97}$$

*for all $t \geq 0$.*

With Lemma A.12 in hand, we can finally prove Theorem A.7.

*Proof of Theorem A.7.* By setting $\Theta' = \Theta$ and $\Theta''$ to be the critical point $\Theta^*$ of $L(\Theta)$ in Lemma A.11, we have

$$\|\nabla L(\Theta)\|_F \geq c\|\Theta - \Theta^*\|_F \tag{98}$$

for any $\Theta \in \mathcal{P}$.

To translate the above gradient lowerbound back to $U$-parametrization, we first note that (35) implies that $U_{11}$ keeps symmetric for all $t \geq 0$. Therefore $\Theta_1 = U_{11} + \mathrm{Sym}(U_{12} + U_{21} + U_{22})$.

Rearrange the parameter matrices in a column to get $\Theta = (\Theta_4^\top, \Theta_3^\top, \Theta_2^\top, \Theta_1^\top)^\top$ and $\tilde{U} = (\tilde{U}_{22}^\top, \tilde{U}_{12}^\top, \tilde{U}_{21}^\top, \tilde{U}_{11}^\top)^\top$. Let $\mathrm{Vec}(A)$ be the vectorization operator in row-wise order. For example, $\mathrm{Vec}\begin{pmatrix} 1 & 2 \\ 3 & 4 \end{pmatrix} = (1,2,3,4)^\top$. It is easy to check that

$$\mathrm{Vec}(\Theta) = \begin{pmatrix} I_{d^2} & & & \\ I_{d^2} & I_{d^2} & & \\ I_{d^2} & & I_{d^2} & \\ \frac{1}{2}(I_{d^2}+T) & \frac{1}{2}(I_{d^2}+T) & \frac{1}{2}(I_{d^2}+T) & I_{d^2} \end{pmatrix} \mathrm{Vec}(\tilde{U}) =: J\,\mathrm{Vec}(\tilde{U}). \tag{99}$$

Here the transformation matrix $J$ is invertible and

$$J^{-1} = \begin{pmatrix} I_{d^2} & & & \\ -I_{d^2} & I_{d^2} & & \\ -I_{d^2} & & I_{d^2} & \\ \frac{1}{2}(I_{d^2}+T) & -\frac{1}{2}(I_{d^2}+T) & -\frac{1}{2}(I_{d^2}+T) & I_{d^2} \end{pmatrix}. \tag{100}$$

It is easy to see $\|J^{-1}\|_2 \leq \|J^{-1}\|_F \leq 3d$. Now we can invoke Lemma A.6 on the gradient lower bound (98) to obtain

$$\|\nabla L(\tilde{U})\|_F = \|\nabla L(\mathrm{Vec}(\tilde{U}))\| \geq \frac{c}{3d}\|\Theta - \Theta^*\|. \tag{101}$$

Finally we translate the gradient lower bound on $\tilde{U}$ back to $U$:

$$\begin{aligned} &\|\nabla L(U)\|_F^2 \\ \geq &\|\nabla_{U_{11}} L(U)\|_F^2 + \|\nabla_{U_{12}} L(U)\|_F^2 + \|\nabla_{U_{21}} L(U)\|_F^2 + \|\nabla_{U_{22}} L(U)\|_F^2 \\ = &u_{-1}^2 \|\nabla L(\tilde{U})\|_F^2. \end{aligned} \tag{102}$$

Therefore using the lower bound of $u_{-1}$ in Lemma A.12, we have

$$\|\nabla L(U)\|_F \geq \frac{c\sigma}{3d} \left( \frac{\alpha}{(2+\sqrt{2})(\|\Lambda\|_F^2 + 2\mathrm{tr}(\Lambda) + d^2 + 2d)} \right)^{\frac{1}{2}} \|\Theta - \Theta^*\|_F. \tag{103}$$

Since $L(U) \geq 0$ is bounded below, we know $L(U(t))$ the loss over gradient flow will converge. Any stationary point $U^*$ of the gradient flow must satisfy $\nabla L(U^*) = 0$, which implies the corresponding $\Theta(U^*)$ is a critical point of $L(\Theta)$. Since $L(\Theta)$ is strongly-convex on $\Theta$, we have that the critical point $\Theta(U^*)$ is a global minimizer of $L$ under the restriction that $u_{13} = u_{23} = u_{31} = u_{32} = 0$. Finally through Lemma A.10 we have the converged $U^*$ is a global minimizer of $L$. $\qquad\square$

### A.2.2 PROOF OF LEMMA A.11 AND LEMMA A.12

The high-level idea of proof for Lemma A.11 is to lower bound the gradients of $L_1$ and $L_2$ respectively and then combine the obtained bounds.

*Proof of Lemma A.11.* Given any $\Theta', \Theta'' \in \mathcal{P}$, we want to show that

$$\|\nabla L(\Theta') - \nabla L(\Theta'')\|_F \geq c\|\Theta' - \Theta''\|_F. \tag{104}$$

Only need to show

$$\langle \nabla L(\Theta') - \nabla L(\Theta''), \Theta' - \Theta'' \rangle \geq c\|\Theta' - \Theta''\|_F^2. \tag{105}$$

With slight abuse of notation, we denote $\Theta := \Theta' - \Theta''$. It is equivalent to show

$$\langle \nabla L(\Theta') - \nabla L(\Theta''), \Theta \rangle \geq c\|\Theta\|_F^2 \tag{106}$$

for some $c > 0$. Recall that we have decomposed the loss function into two parts under $\Theta$-parametrization $L(\Theta) = L_2(\Theta_1, \Theta_2) + L_1(\Theta_3, \Theta_4)$. Therefore it is sufficient to show that

$$\langle \nabla L_1(\Theta_3', \Theta_4') - \nabla L_1(\Theta_3'', \Theta_4''), (\Theta_3, \Theta_4) \rangle \geq c \cdot (\|\Theta_3\|_F^2 + \|\Theta_4\|_F^2).$$

and

$$\langle \nabla L_2(\Theta_1', \Theta_2') - \nabla L_2(\Theta_1'', \Theta_2''), (\Theta_1, \Theta_2) \rangle \geq c \cdot (\|\Theta_1\|_F^2 + \|\Theta_2\|_F^2).$$

**Step 1. Getting the gradient lower bound for $L_1(\Theta_3, \Theta_4)$.** By computation we have

$$\frac{\partial L_1}{\partial \Theta_3'} - \frac{\partial L_1}{\partial \Theta_3''} = (\Theta_3\Lambda + \Lambda\Theta_4\Lambda) + \left(d + 1 + \frac{\mathrm{tr}(\Lambda)}{n}\right)\Theta_3\Lambda + \frac{1}{n}\Lambda\Theta_4\Lambda$$

and

$$\frac{\partial L_1}{\partial \Theta_4'} - \frac{\partial L_1}{\partial \Theta_4''} = \Lambda\Theta_3\Lambda + \Lambda^2\Theta_4\Lambda + \frac{1}{n}\Lambda^2\Theta_4\Lambda + \frac{\mathrm{tr}(\Lambda) + d}{n}\Lambda\Theta_4\Lambda + \frac{1}{n}\Lambda\Theta_3\Lambda.$$

Therefore we have the gradient lower bound

$$\langle \nabla L_1(\Theta_3', \Theta_4') - \nabla L_1(\Theta_3'', \Theta_4''), (\Theta_3, \Theta_4) \rangle$$

$$\geq \mathrm{tr}\left(\left(\frac{19}{100} + \frac{1}{n}\right)\Lambda^2\Theta_4\Lambda\Theta_4^\top + \frac{\mathrm{tr}(\Lambda)}{n}\Lambda\Theta_4\Lambda\Theta_4^\top + \left(d + 2 + \frac{\mathrm{tr}(\Lambda)}{n} - \frac{100\left(1 + \frac{1}{n}\right)^2}{81}\right)\Theta_3\Lambda\Theta_3^\top\right)$$

$$\geq \left(\left(\frac{19}{100} + \frac{1}{n}\right)\lambda_{\min}(\Lambda)^3 + \frac{\mathrm{tr}(\Lambda) + d}{n}\lambda_{\min}(\Lambda)^2\right)\|\Theta_4\|_F^2$$

$$+ \left(d + 2 + \frac{\mathrm{tr}(\Lambda)}{n} - \frac{100\left(1 + \frac{1}{n}\right)^2}{81}\right)\lambda_{\min}(\Lambda)\|\Theta_3\|_F^2.$$

Note that $d + 2 + \frac{\mathrm{tr}(\Lambda)}{n} - \frac{100\left(1 + \frac{1}{n}\right)^2}{81} > d - 1$ if $n \geq 2$. We have the gradient lower bound for $L_1(\Theta_3, \Theta_4)$

$$\langle \nabla L_1(\Theta_3', \Theta_4') - \nabla L_1(\Theta_3'', \Theta_4''), (\Theta_3, \Theta_4) \rangle$$

$$\geq \underbrace{\lambda_{\min}(\Lambda)\min\left\{\frac{19}{100}\lambda_{\min}(\Lambda)^2, d - 1\right\}}_{c_1}\left(\|\Theta_3\|_F^2 + \|\Theta_4\|_F^2\right). \tag{107}$$

**Step 2. Getting the gradient lower bound for** $L_2(\Theta_1, \Theta_2)$**.** Again we first compute the gradient difference

$$\frac{\partial L_2}{\partial \Theta_1'} - \frac{\partial L_2}{\partial \Theta_1''} = 2\left(d + 4 + \frac{\mathrm{tr}(\Lambda)}{n}\right)\Theta_1 + 2\left(1 + \frac{1}{n}\right)\mathrm{Sym}(\Lambda\Theta_2)$$
$$+ \mathrm{tr}\left(\left(1 + \frac{1}{n}\right)\Lambda\Theta_2 + \left(d + 4 + \frac{\mathrm{tr}(\Lambda)}{n}\right)\Theta_1\right)I_d$$

and

$$\frac{\partial L_2}{\partial \Theta_2'} - \frac{\partial L_2}{\partial \Theta_2''} = \left(1 + \frac{1}{n}\right)\Lambda(2\Theta_1 + \Lambda\Theta_2) + \left(1 + \frac{1}{n}\right)\mathrm{tr}(\Theta_1)\Lambda + \frac{d + 2 + \mathrm{tr}(\Lambda)}{n}\Lambda\Theta_2.$$

Therefore we have the gradient lower bound

$$\langle \nabla L_2(\Theta_1', \Theta_2') - \nabla L_2(\Theta_1'', \Theta_2''), (\Theta_1, \Theta_2)\rangle$$

$$\geq \frac{1}{2}\mathrm{tr}(\Theta_1)^2 + \mathrm{tr}\left(\left(\frac{3}{2}d + 8 + \frac{\mathrm{tr}(\Lambda)}{2n}\right)\Theta_1\Theta_1^\top + 2\left(1 + \frac{1}{n}\right)\left(\mathrm{Sym}(\Lambda\Theta_2)\Theta_1^\top + \Lambda\Theta_1\Theta_2^\top\right) + \underbrace{\left(1 + \frac{1}{n}\right)\Lambda\Theta_2\Theta_2^\top\Lambda}_{*}\right)$$

$$+ \mathrm{tr}\left(\left(1 + \frac{1}{n}\right)\mathrm{tr}(\Theta_1)\Lambda\Theta_2^\top + \left(\left(d + 4 + \frac{\mathrm{tr}(\Lambda)}{n}\right)\mathrm{tr}(\Theta_1) + \left(1 + \frac{1}{n}\right)\mathrm{tr}(\Lambda\Theta_2)\right)\Theta_1 + \frac{d + 2 + \mathrm{tr}(\Lambda)}{n\lambda_{\max}(\Lambda)}\Lambda\Theta_2\Theta_2^\top\Lambda\right).$$

The inequality comes from $d\|\Theta_1\|_F^2 \geq \mathrm{tr}(\Theta_1)^2$ and $\mathrm{tr}(\Lambda\Theta_2\Theta_2^\top) \geq \frac{1}{\lambda_{\max}(\Lambda)}\|\Lambda\Theta_2\|_F^2$.

We rewrite the $*$ term as

$$\left(1 + \frac{1}{n}\right)\Lambda\Theta_2\Theta_2^\top\Lambda$$

$$= \left(\frac{d\left(1 + \frac{1}{n}\right)^2}{d + 4 + \xi} + \left(\frac{d\left(1 + \frac{1}{n}\right)^2}{d + 4} - \frac{d\left(1 + \frac{1}{n}\right)^2}{d + 4 + \xi}\right) + \left(1 + \frac{1}{n}\right)\frac{4}{d + 4} - \left(1 + \frac{1}{n}\right)\frac{d}{(d + 4)n}\right)\Lambda\Theta_2\Theta_2^\top\Lambda$$

where $\xi > 0$ will be determined later.

Substituting $\mathrm{tr}\left(\frac{d\left(1 + \frac{1}{n}\right)^2}{d + 4 + \xi}\Lambda\Theta_2\Theta_2^\top\Lambda\right) \geq \frac{\left(1 + \frac{1}{n}\right)^2}{d + 4 + \xi}\mathrm{tr}(\Lambda\Theta_2)^2$ and $\mathrm{tr}\left(\left(1 + \frac{1}{n}\right)\frac{2}{d + 4}\Lambda\Theta_2\Theta_2^\top\Lambda\right) \geq \left(1 + \frac{1}{n}\right)\frac{2}{d + 4}\mathrm{tr}(\Lambda\Theta_2\Lambda\Theta_2)$ into $\mathrm{tr}(*)$ and substituting the rewritten $\mathrm{tr}(*)$ term back, we obtain

$$\langle \nabla L_2(\Theta_1', \Theta_2') - \nabla L_2(\Theta_1'', \Theta_2''), (\Theta_1, \Theta_2)\rangle$$

$$\geq \left(b\,\mathrm{tr}(\Theta_1) + \frac{1 + \frac{1}{n}}{b}\mathrm{tr}(\Lambda\Theta_2)\right)^2 + \left\|\frac{1 + \frac{1}{n}}{a}\Theta_1 + a\left(\Lambda\Theta_2 + \Theta_2^\top\Lambda\right)\right\|_F^2$$

$$+ \frac{\mathrm{tr}(\Lambda)}{n}\mathrm{tr}(\Theta_1)^2 + \left(\frac{3}{2}d + 8 + \frac{\mathrm{tr}(\Lambda)}{2n} - \left(1 + \frac{1}{n}\right)(d + 4)\right)\|\Theta_1\|_F^2$$

$$+ \left(\left(\frac{d + 2 + \mathrm{tr}(\Lambda)}{n\lambda_{\max}(\Lambda)} - \left(1 + \frac{1}{n}\right)\frac{d}{(d + 4)n}\right) + \left(\frac{d\left(1 + \frac{1}{n}\right)^2}{d + 4} - \frac{d\left(1 + \frac{1}{n}\right)^2}{d + 4 + \xi}\right)\right)\|\Lambda\Theta_2\|_F^2$$

where $a := \sqrt{(1 + \frac{1}{n})/(d + 4)}$ and $b := \sqrt{d + 4 + \xi}$. After subtracting the first three positive square terms at RHS and substituting $\|\Lambda\Theta_2\|_F \geq \lambda_{\min}(\Lambda)\|\Theta_2\|_F$, we reach

$$\langle \nabla L_2(\Theta_1', \Theta_2') - \nabla L_2(\Theta_1'', \Theta_2''), (\Theta_1, \Theta_2)\rangle$$

$$\geq \left(\frac{3}{2}d + 8 + \frac{\mathrm{tr}(\Lambda)}{2n} - \left(1 + \frac{1}{n}\right)(d + 4)\right)\|\Theta_1\|_F^2$$

$$+ \left(\left(\frac{d + 2 + \mathrm{tr}(\Lambda)}{n\lambda_{\max}(\Lambda)} - \left(1 + \frac{1}{n}\right)\frac{d}{(d + 4)n}\right) + \left(\frac{d\left(1 + \frac{1}{n}\right)^2}{d + 4} - \frac{d\left(1 + \frac{1}{n}\right)^2}{d + 4 + \xi}\right)\right)\lambda_{\min}(\Lambda)^2\|\Theta_2\|_F^2.$$

Since $n \geq 2$, we can see the coefficient of $\|\Theta_1\|^2$ is positive and bounded below by 2. It remains to pick some $\xi$ such that the coefficient of $\|\Lambda \Theta_2\|_F^2$ is positive. It is easy to see

$$\frac{d + 2 + \mathrm{tr}(\Lambda)}{n \lambda_{\max}(\Lambda)} - \left(1 + \frac{1}{n}\right) \frac{d}{(d+4)n} \geq \underbrace{\frac{d^2 - \frac{1}{n}\lambda_{\max}(\Lambda)d}{n\lambda_{\max}(\Lambda)(d+4)}}_{**}.$$

Now we consider two complementary cases.

Case 1: If $\lambda_{\max}(\Lambda) \leq nd$, then we have $** \geq 0$. So we can simply pick $\xi = 4.5$ and we have

$$\langle \nabla L_2(\Theta_1', \Theta_2') - \nabla L_2(\Theta_1'', \Theta_2''), (\Theta_1, \Theta_2)\rangle \geq \min\{2, \frac{\lambda_{\min}(\Lambda)^2}{55d}\} \left(\|\Theta_1\|_F^2 + \|\Theta_2\|_F^2\right).$$

Case 2: If $\lambda_{\max}(\Lambda) > nd$, we can then pick $\xi = \frac{\mathrm{tr}(\Lambda)}{n}$ and by computation we have

$$\frac{d^2 - \frac{1}{n}\lambda_{\max}(\Lambda)d}{n\lambda_{\max}(\Lambda)(d+4)} + \left(\frac{d\left(1 + \frac{1}{n}\right)^2}{d+4} - \frac{d\left(1 + \frac{1}{n}\right)^2}{d+4+\xi}\right) \geq \frac{d^2}{n(d+4)\lambda_{\max}(\Lambda)}$$

when $n \geq 2$.

Therefore with direct computation with the assumption $\lambda_{\max}(\Lambda) > nd$ we reach

$$\langle \nabla L_2(\Theta_1', \Theta_2') - \nabla L_2(\Theta_1'', \Theta_2''), (\Theta_1, \Theta_2)\rangle \geq \min\{2, \frac{d^3\lambda_{\min}(\Lambda)^2}{(d+4)\lambda_{\max}(\Lambda)^2}\} \left(\|\Theta_1\|_F^2 + \|\Theta_2\|_F^2\right).$$

Combining both cases, we obtain the gradient lower bound of $L_2$

$$\langle \nabla L_2(\Theta_1', \Theta_2') - \nabla L_2(\Theta_1'', \Theta_2''), (\Theta_1, \Theta_2)\rangle \geq \underbrace{\min\{2, \frac{d^3\lambda_{\min}(\Lambda)^2}{(d+4)\lambda_{\max}(\Lambda)^2}, \frac{\lambda_{\min}(\Lambda)^2}{55d}\}}_{c_2} \left(\|\Theta_1\|_F^2 + \|\Theta_2\|_F^2\right).$$

(108)

**Step 3. Combine both gradient lower bounds (107) and (108).** We finish the proof with

$$\langle \nabla L(\Theta') - \nabla L(\Theta''), \Theta\rangle \geq \min\{c_1, c_2\}\|\Theta\|_F^2. \tag{109}$$

$\square$

Now we prove Lemma A.12. The high-level idea of the proof is same as Lemma A.4. We again write the loss $L_t(u_{-1}) = a(t)u_{-1}^2 - b(t)u_{-1} + c$ as a time-dependent degree-2 function of $u_{-1}$. Note that $b(t)$ is independent of $n$ so we can re-use the previous bound on it. Since $a(t)$ is dependent on $n$, we re-calculate the upper bound of the coefficient for degree-2 term in this proof.

*Proof.* **Rewrite loss as a degree-$2$ function of $u_{-1}$.** We first write loss function as a degree-2 polynomial of $u_{-1}$. Recall that

$$L(\Theta) = \underbrace{\frac{1}{2}\mathbb{E}_{z_i, \mu_\tau}\left[\left\|\left((\mu_\tau + \hat{z})\mu_\tau^\top \Theta_1 \mu_\tau + \left(\mu_\tau \hat{z}^\top + \hat{Z}\right)\Theta_2 \mu_\tau - \mu_\tau\right)\right\|^2\right]}_{L_2}$$
$$+ \underbrace{\frac{1}{2}\mathbb{E}_{z_i, \mu_\tau}\left[\left\|(\mu_\tau + \hat{z})\mu_\tau^\top \Theta_3 \sqrt{\Lambda} + \left(\mu_\tau \hat{z}^\top + \hat{Z}\right)\Theta_4 \sqrt{\Lambda} - \sqrt{\Lambda}\right\|_F^2\right]}_{L_1}.$$

Since $\Theta$ is degree-1 w.r.t. $u_{-1}$, we can write $L_2$ and $L_1$ as degree-2 polynomials

$$L_1 =: c_{1,2}u_{-1}^2 - c_{1,1}u_{-1} + \text{constant},$$

$$L_2 =: c_{2,2}u_{-1}^2 - c_{2,1}u_{-1} + \text{constant}.$$

Here constant refers to the term that is constant w.r.t. $u_{-1}$. By computation, we can write out the coefficients $c_{1,1}, c_{2,1}, c_{1,2}, c_{2,2}$. After writing them out, as we did in proof of Lemma A.3, we will give lower and upper bounds on $c_{1,1} + c_{2,1}$ and an upper bound on $c_{1,2} + c_{2,2}$. Now we denote $\tilde{\Theta} := \frac{1}{u_{-1}}\Theta$. We have

$$c_{2,1} = \operatorname{tr}\left(\Lambda\tilde{\Theta}_2\right) + 2\operatorname{tr}\left(\tilde{\Theta}_1\right) + d\operatorname{tr}\left(\tilde{\Theta}_1\right) \text{ and } c_{1,1} = \operatorname{tr}\left(\Lambda\tilde{\Theta}_4\Lambda\right) + \operatorname{tr}\left(\tilde{\Theta}_3\Lambda\right).$$

Note that these two coefficients are independent of $n$, which implies that previous bounds (50) and (52) still holds when $n$ is finite. We restate these bounds here as

$$c_{1,1}(0) + c_{2,1}(0) \geq (d+2)u_{-1}(0) \text{ and } c_{1,1} + c_{2,1} \leq (2+\sqrt{2})u_{-1}\left(\|\Lambda\|_F^2 + 2\operatorname{tr}(\Lambda) + d^2 + 2d\right).$$

The formulas for $c_{1,2}$ and $c_{2,2}$ are more complicated:

$$
\begin{aligned}
2c_{1,2} =& (d+2)\operatorname{tr}\left(\tilde{\Theta}_3\Lambda\tilde{\Theta}_3^\top\right) + \frac{\operatorname{tr}(\Lambda)}{n}\operatorname{tr}\left(\tilde{\Theta}_3\Lambda\tilde{\Theta}_3^\top\right) + \frac{d}{n}\operatorname{tr}\left(\Lambda\tilde{\Theta}_4\Lambda\tilde{\Theta}_4^\top\right) \\
& + \operatorname{tr}\left(\left(\left(1+\frac{1}{n}\right)\Lambda^2 + \frac{\operatorname{tr}\Lambda}{n}\Lambda\right)\tilde{\Theta}_4\Lambda\tilde{\Theta}_4^\top\right) + 2\operatorname{tr}\left(\tilde{\Theta}_3\Lambda\tilde{\Theta}_4^\top\Lambda\right) + \frac{2}{n}\operatorname{tr}\left(\Lambda\tilde{\Theta}_3\Lambda\tilde{\Theta}_4^\top\right),
\end{aligned}
$$
$$(110)$$

$$
\begin{aligned}
2c_{2,2} =& \left(d+4+\frac{\operatorname{tr}(\Lambda)}{n}\right)\cdot\left(\operatorname{tr}\left(\tilde{\Theta}_1\right)^2 + \operatorname{tr}\left(\tilde{\Theta}_1^2\right) + \operatorname{tr}\left(\tilde{\Theta}_1^\top\tilde{\Theta}_1\right)\right) \\
& + \frac{d+2}{n}\operatorname{tr}\left(\Lambda\tilde{\Theta}_2\tilde{\Theta}_2^\top\right) + \left(1+\frac{1}{n}\right)\operatorname{tr}\left(\Lambda^2\tilde{\Theta}_2\tilde{\Theta}_2^\top\right) \\
& + \frac{\operatorname{tr}(\Lambda)}{n}\operatorname{tr}\left(\Lambda\tilde{\Theta}_2\tilde{\Theta}_2^\top\right) + 2\left(1+\frac{1}{n}\right)\left(\operatorname{tr}\left(\tilde{\Theta}_1\Lambda\tilde{\Theta}_2\right) + \operatorname{tr}\left(\tilde{\Theta}_1^\top\Lambda\tilde{\Theta}_2\right) + \operatorname{tr}\left(\Lambda\tilde{\Theta}_2\right)\operatorname{tr}(\tilde{\Theta}_1)\right)
\end{aligned}
$$
$$(111)$$

**Re-calculate the upper bounds for** $c_{1,2} + c_{2,2}$. We first upper bound $c_{2,2}$.

$$
\begin{aligned}
2c_{2,2} \leq & \left(d+4+\frac{\operatorname{tr}(\Lambda)}{n}\right)\left(d\left\|\tilde{\Theta}_1\right\|_F^2 + 2\left\|\tilde{\Theta}_1\right\|_F^2\right) + \frac{d+2}{n}\|\Lambda\|_F\left\|\tilde{\Theta}_2\right\|_F^2 + \left(1+\frac{1}{n}\right)\|\Lambda\|_F^2\left\|\tilde{\Theta}_2\right\|_F^2 \\
& + \frac{\operatorname{tr}(\Lambda)}{n}\|\Lambda\|_F\left\|\tilde{\Theta}_2\right\|_F^2 + 2\left(1+\frac{1}{n}\right)\left(2\|\Lambda\|_F\left\|\tilde{\Theta}_1\right\|_F\left\|\tilde{\Theta}_2\right\|_F + d\|\Lambda\|_F\left\|\tilde{\Theta}_2\right\|_F\left\|\tilde{\Theta}_1\right\|_F\right) \\
\leq & \left(\left(d+4+\frac{\operatorname{tr}(\Lambda)}{n}\right)(d+2) + \left(1+\frac{1}{n}\right)(d+2)\|\Lambda\|_F\right)\cdot\left\|\tilde{\Theta}_1\right\|_F^2 \\
& + \left(\frac{d+2+\operatorname{tr}(\Lambda)}{n}\|\Lambda\|_F + \left(1+\frac{1}{n}\right)\|\Lambda^2\|_F + \left(1+\frac{1}{n}\right)(2+d)\|\Lambda\|_F\right)\left\|\tilde{\Theta}_2\right\|_F^2 \\
\leq & 4\left(\left(d+4+\frac{\operatorname{tr}(\Lambda)}{n}\right)(d+2) + \left(1+\frac{1}{n}\right)(2+d)\|\Lambda\|_F + \frac{d+2+\operatorname{tr}(\Lambda)}{n}\|\Lambda\|_F\right. \\
& + \left.\left(1+\frac{1}{n}\right)\|\Lambda\|_F^2 + \left(1+\frac{1}{n}\right)(2+d)\|\Lambda\|_F\right)\cdot\|U\|_F^2 \\
= & 4\left(\left(d+4+\frac{\operatorname{tr}(\Lambda)}{n}\right)(d+2) + \left(1+\frac{1}{n}\right)(2+d)\|\Lambda\|_F + \frac{d+2+\operatorname{tr}(\Lambda)}{n}\|\Lambda\|_F\right. \\
& + \left.\left(1+\frac{1}{n}\right)\|\Lambda\|_F^2 + \left(1+\frac{1}{n}\right)(2+d)\|\Lambda\|_F\right)\cdot u_{-1}^2.
\end{aligned}
$$
$$(112)$$

The first inequality is because Cauchy-Schwartz inequality and $\operatorname{tr}(\Lambda\tilde{\Theta}_2) \leq \sqrt{d}\|\Lambda\tilde{\Theta}_2\|_F \leq \sqrt{d}\|\Lambda\|_F^2\|\tilde{\Theta}_2\|$ and $\operatorname{tr}(\tilde{\Theta}_1) \leq \sqrt{d}\|\tilde{\Theta}_1\|_F$. The second inequality is because $\|\tilde{\Theta}_2\|_F\|\tilde{\Theta}_1\|_F \leq \frac{1}{2}\left(\|\tilde{\Theta}_1\|_F^2 + \|\tilde{\Theta}_2\|_F^2\right)$. The third inequality is because for $i \in [2]$ we have $\|\tilde{\Theta}_i\|_F^2 \leq (\|U_{11}\|_F + \|U_{12}\|_F + \|U_{21}\|_F + \|U_{22}\|_F)^2 \leq 4\|U\|_F^2$ as $u_{-1}$. The last equality is because $u_{-1}$ is balanced with $U$ matrices.

We next upper bound $c_{1,2}$.

$$
\begin{aligned}
2c_{1,2} \leq & \left(d + 2 + \frac{\operatorname{tr}(\Lambda)}{n}\right) \|\Lambda\|_F \left\|\tilde{\Theta}_3\right\|_F^2 + \frac{d}{n}\|\Lambda\|_F^2 \left\|\tilde{\Theta}_4\right\|_F^2 + \left(1 + \frac{1}{n}\right)\|\Lambda\|_F^3 \left\|\tilde{\Theta}_4\right\|_F^2 \\
& + \frac{\operatorname{tr}(\Lambda)}{n}\|\Lambda\|_F^2 \left\|\tilde{\Theta}_4\right\|_F^2 + 2\left(1 + \frac{1}{n}\right)\|\Lambda\|_F^2 \left\|\tilde{\Theta}_3\right\|_F \left\|\tilde{\Theta}_4\right\|_F \\
\leq & \left(\left(d + 2 + \frac{\operatorname{tr}(\Lambda)}{n}\right)\|\Lambda\|_F + \left(1 + \frac{1}{n}\right)\|\Lambda\|_F^2\right)\left\|\tilde{\Theta}_3\right\|_F^2 \\
& + \left(\frac{d}{n}\|\Lambda\|_F^2 + \left(1 + \frac{1}{n}\right)\|\Lambda\|_F^3 + \frac{\operatorname{tr}(\Lambda)}{n}\|\Lambda\|_F^2 + \left(1 + \frac{1}{n}\right)\|\Lambda\|_F^2\right)\left\|\tilde{\Theta}_4\right\|_F^2 \\
\leq & \left(2\left(d + 2 + \frac{\operatorname{tr}(\Lambda)}{n} + \left(1 + \frac{1}{n}\right)\|\Lambda\|_F\right)\|\Lambda\|_F\right. \\
& + \left.\left(1 + \frac{d + 1 + \operatorname{tr}(\Lambda)}{n} + \left(1 + \frac{1}{n}\right)\|\Lambda\|_F\right)\|\Lambda\|_F^2\right) u_{-1}^2
\end{aligned}
\tag{113}
$$

The first inequality comes from Cauchy-Schwartz inequality and submultiplicativity of Frobenius norm. The second inequality is true because $2\|\tilde{\Theta}_3\|_F\|\tilde{\Theta}_4\|_F \leq \|\tilde{\Theta}_3\|_F^2 + \|\tilde{\Theta}_4\|_F^2$. The last inequality is because $\|\tilde{\Theta}_3\|_F^2 \leq 2(\|U_{12}\|_F^2 + \|U_{22}\|_F^2) \leq 2\|U\|_F^2$ and $\|\tilde{\Theta}_4\|_F^2 = \|U_{22}\|_F^2 \leq \|U\|_F^2$.

Now combining (113) and (112) we have

$$
\begin{aligned}
c_{1,2} + c_{2,2} \leq & 2\left(\left(\left(d + 4 + \frac{\operatorname{tr}(\Lambda)}{n}\right)(d+2) + \left(1 + \frac{1}{n}\right)(2+d)\|\Lambda\|_F + \frac{d + 2 + \operatorname{tr}(\Lambda)}{n}\|\Lambda\|_F\right.\right. \\
& + \left(1 + \frac{1}{n}\right)(2+d)\|\Lambda\|_F + \left(d + 2 + \frac{\operatorname{tr}(\Lambda)}{n} + \left(1 + \frac{1}{n}\right)\|\Lambda\|_F\right)\|\Lambda\|_F \\
& + \left.\left(1 + \frac{1}{n}\right)\|\Lambda\|_F^2 + \left(1 + \frac{d + 1 + \operatorname{tr}(\Lambda)}{n} + \left(1 + \frac{1}{n}\right)\|\Lambda\|_F\right)\|\Lambda\|_F^2\right) \cdot u_{-1}^2 \\
=: & c(d, n, \Lambda) u_{-1}^2.
\end{aligned}
\tag{114}
$$

**Upper bound $L_0(u_{-1}(0))$.** Now we can first upper bound $L_0(u_{-1}(0))$ by

$$
\begin{aligned}
L_0(u_{-1}(0)) &= (c_{1,2}(0) + c_{2,2}(0))\, u_{-1}^2(0) - (c_{1,1}(0) + c_{2,1}(0))\, u_{-1}(0) + \text{ constant} \\
&\leq c(d, n, \Lambda) \cdot u_{-1}^4(0) - (d+2)u_{-1}^2(0) + \text{ constant} \\
&\leq -\frac{d+2}{2c(d,n,\Lambda)}u_{-1}^2(0) + \text{ constant}.
\end{aligned}
\tag{115}
$$

The first inequality is simply combing the upper and lower bounds we just obtained. The second inequality is because by assumption we have $u_{-1}(0)^2 \leq \frac{d+2}{2c(d,n,\Lambda)}$.

**Lower bound $L_t(u_{-1}(t))$.** Then we lower bound $L_t(u_{-1}(t))$ by

$$
\begin{aligned}
L_t(u_{-1}(t)) &= (c_{1,2} + c_{2,2})\, u_{-1}^2 - (c_{1,1} + c_{2,1})\, u_{-1} + \text{ constant} \\
&\geq -(c_{1,1} + c_{2,1})\, u_{-1} + \text{ constant} \\
&\geq -(2 + \sqrt{2})\left(\|\Lambda\|_F^2 + 2\operatorname{tr}(\Lambda) + d^2 + 2d\right) u_{-1}^2 + \text{ constant}.
\end{aligned}
\tag{116}
$$

The first inequality is because $c_{1,2} + c_{2,2} > 0$. The second inequality is because we have the upper bound on $c_{1,1} + c_{2,1}$ and the fact that $u_{-1} > 0$. To see this, just noting that (115) implies that $L(U(0)) < $ constant and hence $L(U) < $ constant throughout training. When $u_{-1} = 0$, the loss is equal to the constant. This implies that $u_{-1}(t) \neq 0$ for all $t > 0$. Since $u_{-1}(0) > 0$ and $u_{-1}$ is continuous w.r.t. $t$, we have $u_{-1}(t) > 0$ for all $t$; otherwise there will be some $t_0$ such that $u_{-1}(t_0) = 0$, which is impossible.

**Combine both upper and lower bounds.** Combining (115) and (116) and using $L(U) \leq L(U(0))$, we have

$$
u_{-1} \geq \sqrt{\frac{d+2}{2c(d,n,\Lambda) \cdot (2 + \sqrt{2})\left(\|\Lambda\|_F^2 + 2\operatorname{tr}(\Lambda) + d^2 + 2d\right)}}\, u_{-1}(0),
\tag{117}
$$

which finished the proof. $\qquad\square$

### A.2.3 PROOF OF LEMMA A.10

It remains to prove Lemma A.10. This proof is inspired by Ahn et al. (2023). The high-level idea of this proof is to show the loss functions $L_1$ and $L_2$ are convex under a new parametrization, which is the product of the last row of $W^{PV}$ and the first two $d$ columns of $W^{KQ}$. This implies the global minimum is achieved at the critical point under the new parametrization. Note the value of the new parameters also depend on $u_{13}, u_{23}, u_{32}$ and $u_{31}$. We then show that there exists a critical point such that $u_{13} = u_{23} = u_{32} = u_{31} = 0_d$, which completes the proof. When the covariance $\Lambda$ is non-diagonal, we use the SVD decomposition of $\Lambda$ to reparametrize $L_1$ and further simplify the calculations on $L_1$.

*Proof.* **Reparametrize the loss function to be convex.** We will show that setting $u_{13} = u_{23} = u_{32} = u_{31} = 0_d$ does not lose optimality. Denote $a := (u_{13}^\top, u_{23}^\top, u_{-1})^\top \in \mathbb{R}^{2d+1}$. Then we can write prediction as

$$\hat{y}_{\tau,\text{query}} = \frac{1}{n} a^\top E M E^\top (W_1, W_2) \begin{pmatrix} \mu_\tau \\ x_{\tau,\text{query}} \end{pmatrix}.$$

Here $E$ is the input embedding matrix and with slight abuse of notation $W_1, W_2 \in \mathbb{R}^{(2d+1) \times d}$ denote the first $d$ and the second $d$ columns of $W^{KQ}$ respectively. Note that $u_{32}$ and $u_{31}$ are the last row of $W_1$ and $W_2$ respectively.

Now we again decompose the loss into variance and bias

$$
\begin{aligned}
L &= \frac{1}{2} \mathbb{E}_{z_i, z_{\text{query}}, \mu_\tau} \left[ \left( \hat{y}_{\tau,\text{query}} - w_\tau^\top x_{\tau,\text{query}} \right)^2 \right] \\
&= \frac{1}{2} \mathbb{E}_{z_i, \mu_\tau} \left[ \left( \frac{1}{n} a^\top E M E^\top W_1 u_\tau + \frac{1}{n} a^\top E M E^\top W_2 u_\tau - w_\tau^\top u_\tau \right)^2 \right] + \frac{1}{2} \mathbb{E} \left[ \text{tr} (A\Lambda) \right] \\
&= \underbrace{\frac{1}{2} \mathbb{E}_{z_i, \mu_\tau} \left[ \text{tr} \left( B u_\tau u_\tau^\top \right) \right]}_{L_2} + \underbrace{\frac{1}{2} \mathbb{E}_{z_i, \mu_\tau} \left[ \text{tr} (A\Lambda) \right]}_{L_1}
\end{aligned}
\tag{118}
$$

where

$$A := \left( \frac{1}{n} W_2^\top E M E^\top a - w_\tau \right) \left( \frac{1}{n} W_2^\top E M E^\top a - w_\tau \right)^\top$$

and

$$B := \left( \frac{1}{n} W_1^\top E M E^\top a + \frac{1}{n} W_2^\top E M E^\top a - w_\tau \right) \left( \frac{1}{n} W_1^\top E M E^\top a + \frac{1}{n} W_2^\top E M E^\top a - w_\tau \right)^\top.$$

Denote $W_{1i} \in \mathbb{R}^d$ and $W_{2i} \in \mathbb{R}^d$ the $i$-th column of $W_1$ and $W_2$ respectively. Denote $w_{\tau i}$ the $i$-th coordinate of $w_\tau$. Next we will see that $L$ is convex w.r.t. to a new parametrization $(aW_{11}^\top, \cdots, aW_{1d}^\top, aW_{21}^\top, \cdots, aW_{2d}^\top)$.

We first claim that loss $L$ is convex w.r.t. the parametrization $(aW_{11}^\top, \cdots, aW_{1d}^\top, aW_{21}^\top, \cdots, aW_{2d}^\top)$. With a slight abuse of notation we denote this parametrization as $aW$. To see this, we first note that $L_1 = \text{tr}(A\Lambda)$ is convex w.r.t. to the $d$-dimensional vector $\frac{1}{n} W_2^\top E M E^\top a - w_\tau$ since $\Lambda$ is a PSD matrix. Additionally, we have $\frac{1}{n} W_2^\top E M E^\top a - w_\tau$ is an affine function of $\left( aW_{21}^\top, \cdots, aW_{2d}^\top \right) \in \mathbb{R}^{(2d+1) \times (2d+1)d}$ since the $j$-th coordinate $\left[ \frac{1}{n} W_2^\top E M E^\top a - w_\tau \right]_j = \left\langle \frac{1}{n} E M E^\top, aW_{2j}^\top \right\rangle - w_{\tau j}$ is an affine function of $aW_{2j}^\top$ for $j \in [d]$. The desired convexity comes from the fact that the composition of a convex function and an affine map is a convex function. In the same way we can see that $L_2 = \text{tr}(B\mu_\tau \mu_\tau^\top)$ is convex w.r.t. the parametrization $aW$. Therefore, $L(aW)$ is a convex function.

**Further simplify $L_2$ for general covariance matrix.** We write $\Lambda := V \Sigma V^\top$ the SVD decomposition of covariance matrix $\Lambda$. Denote $\text{diag}(\sigma_1, \cdots, \sigma_d)$ the diagonal matrix $\Sigma$. Let $\alpha_1, \beta_1, \alpha_2$ and $\beta_2$ be $d$-dimensional vectors to be determined later. Denote $\text{diag}(a)$ the $d \times d$ diagonal matrix whose diagonals are the vector $a$.

Define $\tilde{w}_\tau := V^\top w_\tau$, $\tilde{W}_2 := \begin{pmatrix} V^\top & 0_{d\times d} & 0_d \\ 0_{d\times d} & V^\top & 0_d \\ 0_d^\top & 0_d^\top & 1 \end{pmatrix} W_2 V$ and $\tilde{a} := \begin{pmatrix} V^\top & 0_{d\times d} & 0_d \\ 0_{d\times d} & V^\top & 0_d \\ 0_d^\top & 0_d^\top & 1 \end{pmatrix} a$.

For each in-context example $i \in [n]$, we define $\tilde{x}_{\tau,i} = V^\top x_{\tau,i}$ and $\tilde{\mu}_\tau := V^\top \mu_\tau$. We can in turn define

$$\tilde{E} := \begin{pmatrix} V^\top & 0_{d\times d} & 0_d \\ 0_{d\times d} & V^\top & 0_d \\ 0_d^\top & 0_d^\top & 1 \end{pmatrix} E = \begin{pmatrix} \tilde{\mu}_\tau & \tilde{\mu}_\tau & \cdots & \tilde{\mu}_\tau & \tilde{\mu}_\tau \\ \tilde{x}_{\tau,1} & \tilde{x}_{\tau,2} & \cdots & \tilde{x}_{\tau,n} & x_{\tau,\,\text{query}} \\ y_{\tau,1} & y_{\tau,2} & \cdots & y_{\tau,n} & 0 \end{pmatrix}. \tag{119}$$

We can simplify $L_1$ by

$$L_1 = \frac{1}{2} \sum_{i=1}^d \sigma_i \mathbb{E}\left[\left[V^\top \left(\frac{1}{n} W_2^\top E M E^\top a - w_\tau\right)\right]_i^2\right]$$

$$= \frac{1}{2} \sum_{i=1}^d \sigma_i \mathbb{E}\left[\left[V^\top \left(\frac{1}{n} W_2^\top \begin{pmatrix} V & 0_{d\times d} & 0_d \\ 0_{d\times d} & V & 0_d \\ 0_d^\top & 0_d^\top & 1 \end{pmatrix} \tilde{E} M \tilde{E}^\top \begin{pmatrix} V^\top & 0_{d\times d} & 0_d \\ 0_{d\times d} & V^\top & 0_d \\ 0_d^\top & 0_d^\top & 1 \end{pmatrix} a - w_\tau\right)\right]_i^2\right]$$

$$= \frac{1}{2} \sum_{i=1}^d \sigma_i \mathbb{E}\left[\left[\frac{1}{n} \tilde{W}_2^\top \tilde{E} M \tilde{E}^\top \tilde{a} - \tilde{w}_\tau\right]_i^2\right]. \tag{120}$$

We can then write $L_1$ as a convex function of new parameters $\tilde{a}\tilde{W}_2 := \left(\tilde{a}\tilde{W}_{21}^\top, \cdots, \tilde{a}\tilde{W}_{2d}^\top\right)$. To this end, we have written $L_1$ as a convex function of $\tilde{a}\tilde{W}_2$ and $L_2$ as a convex function of $aW$.

**Show the existence of critical point such that** $u_{13} = u_{23} = u_{32} = u_{31} = 0_d$**.** Since the critical point for a convex function is a global minimizer, it is sufficient to show that there is a set of $a, W_1$ and $W_2$ satisfying $u_{13} = u_{23} = u_{32} = u_{31} = 0_d$, such that the corresponding $\tilde{a}\tilde{W}$ is a critical point of $L_1(\tilde{a}\tilde{W})$ and the corresponding $aW$ is a critical point of $L_2(aW)$. It is equivalent to set $u_{13} = u_{23} = u_{32} = u_{31} = 0_d$ and prove the existence of solutions to the equations $\nabla_{\tilde{a}\tilde{W}_{1i}^\top} L_1 = 0_{d\times d}$ and $\nabla_{aW_{1i}^\top} L_2 = 0_{d\times d}$ for all $i \in [d]$.

Now we construct such solution in the form of

$$a = e_{2d+1}, W_2 = \begin{pmatrix} V & 0_{d\times d} & 0_d \\ 0_{d\times d} & V & 0_d \\ 0_d^\top & 0_d^\top & 1 \end{pmatrix} \begin{pmatrix} -\text{diag}(\alpha_2) \\ \text{diag}(\beta_2) \\ 0_{2d+1}^\top \end{pmatrix} V^\top, W_1 = \begin{pmatrix} -\text{diag}(\alpha_1) \\ \text{diag}(\beta_1) \\ 0_{2d+1}^\top \end{pmatrix} - W_2$$

where $e_j \in \mathbb{R}^d$ is the vector such that the $j$-th coordinate is 1 and others are 0. Note that this construction is consistent with the restriction that $u_{13} = u_{23} = u_{32} = u_{31} = 0_d$ since we have made the first $2d$ elements of $a$ and the last row of $W_2$ and $W_1$ zero.

We compute the corresponding gradient

$$\nabla_{\tilde{a}\tilde{W}_{2j}} L_1 = \frac{\sigma_j}{n} \mathbb{E}\left[\left(\left\langle \frac{1}{n} \tilde{E} M \tilde{E}^\top, \tilde{a}\tilde{W}_{2j}\right\rangle - \tilde{w}_{\tau_j}\right) \tilde{E} M \tilde{E}^\top\right].$$

By direct computation we have

$$\mathbb{E}\left[\tilde{w}_{\tau j} \tilde{E} M \tilde{E}^\top\right] = \begin{pmatrix} 0 & 0 & e_j \\ 0 & 0 & (1+\sigma_j) e_j \\ e_j^\top & (1+\sigma_j) e_j^\top & 0 \end{pmatrix}$$

and

$$\mathbb{E}\left[\left\langle \frac{1}{n} \tilde{E} M \tilde{E}^\top, \tilde{a}\tilde{W}_{2j}\right\rangle \tilde{E} M \tilde{E}^\top\right] = \begin{pmatrix} 0 & 0 & p_j e_j \\ 0 & 0 & q_j e_j \\ p_j e_j^\top & q_j e_j^\top & 0 \end{pmatrix}$$

where $np_j = \beta_{2j}\sigma_j - \alpha_{2j}(d+2+\frac{\text{tr}(\Sigma)}{n})$ and $nq_j = \beta_{2j}\sigma_j(1+\sigma_j) - \alpha_{2j}\left(3 + (1+\frac{1}{n})\sigma_j + \frac{\text{tr}(\Sigma)}{n}\right)$.

We can now decide the values of $\alpha_2$ and $\beta_2$ by setting $p_j = 1$ and $q_j = 1 + \sigma_j$ for all $j \in [d]$. Since it is a full-rank linear system so the solution exists [5] and the corresponding $\nabla_{\tilde{a}\tilde{W}_{2j}} L_1 = 0$. Hence the corresponding $\alpha_2$ and $\beta_2$ is a global minimizer for $L_1$. Therefore, the corresponding $a$ and $W_2$ is a global minimizer for $L_1$.

Now we compute the gradient for $L_2$.

$$
\begin{aligned}
&\nabla_{aW_{2j}} L_2 \\
=&\nabla_{aW_{1j}} L_2 \\
=&\sum_{i \neq j} \frac{1}{2n} \mathbb{E}\left[\mu_{\tau i}\mu_{\tau j}\left(\left\langle \frac{1}{n}EME^\top, a\left(W_{1i} + W_{2i}\right)\right\rangle - w_{\tau i}\right)EME^\top\right] \\
&+ \frac{1}{n}\mathbb{E}\left[\mu_{\tau j}^2\left(\left\langle \frac{1}{n}EME^\top, a\left(W_{1j} + W_{2j}\right)\right\rangle - w_{\tau j}\right)EME^\top\right].
\end{aligned}
$$

We have the sum of minus terms as

$$
\mathbb{E}\left[\left(\mu_{\tau j}^2 w_{\tau j} + \frac{1}{2}\sum_{i \neq j}\mu_{\tau i}\mu_{\tau j}w_{\tau i}\right)EME^\top\right] = \begin{pmatrix} 0 & 0 & \frac{d+1}{2}e_j \\ 0 & 0 & \sigma_j e_j \\ \frac{d+1}{2}e_j^\top & \sigma_j e_j^\top & 0 \end{pmatrix}
$$

and the sum of plus terms as

$$
\mathbb{E}\left[\left(\mu_{\tau j}^2 + \frac{1}{2}\sum_{i \neq j}\mu_{\tau i}\mu_{\tau j}\right)\left\langle \frac{1}{n}EME^\top, a\left(W_{1i} + W_{2i}\right)\right\rangle EME^\top\right] = \begin{pmatrix} 0 & 0 & s_j e_j \\ 0 & 0 & t_j e_j \\ s_j e_j^\top & t_j e_j^\top & 0 \end{pmatrix}.
$$

Here

$$
s_j := -\frac{\alpha_{1j}}{n}\left(3d + 12 + \frac{\mathrm{tr}(\Sigma)}{n}\right) + \frac{\beta_{2j}}{n}\left(\mathrm{tr}(\Sigma) + 2\sigma_j\right)
$$

and

$$
\begin{aligned}
t_j := &-\frac{\alpha_{1j}}{n}\left(\left(d + 4 + \left(1 + \frac{1}{n}\right)\sigma_j + \frac{\mathrm{tr}(\Sigma)}{n}\right)d + 2d + 8 + 2\left(1 + \frac{1}{n}\right)\sigma_j + \frac{2\mathrm{tr}(\Sigma)}{n}\right) \\
&+ \frac{\beta_{1j}}{n}\left(\mathrm{tr}(\Sigma) + 2\sigma_j + \sigma_j^2\right).
\end{aligned}
$$

A sufficient condition for critical point is $s_j = \frac{d+1}{2}$ and $t_j = \sigma_j$, from which we can solve out $\alpha_1$ and $\beta_1$. Again it is a full-rank linear system so the solution exists. □

### A.3  Proof of Theorem A.8.

Theorem A.8 give a characterization of the optimal solution when $\Lambda = I_d$, which can simply be proved by two steps: (i) Computing the critical point of the $\Theta$-parametrized loss function; (ii) Translate the critical point condition back to the original parametrization along with the balancing condition.

*Proof.* Since $\Lambda = I_d$, we have

$$
L_1(\Theta_3, \Theta_4) = \frac{1}{2}\mathbb{E}_{z_i, \mu_\tau}\left[\left\|\hat{\mu}_\tau \mu_\tau^\top \Theta_3 + \left(\mu_\tau \hat{z}^\top + \hat{Z}\right)\Theta_4 - I_d\right\|_F^2\right],
$$

$$
L_2(\Theta_1, \Theta_2) = \frac{1}{2}\mathbb{E}_{z_i, \mu_\tau}\left[\left\|\left(\hat{\mu}_\tau \mu_\tau^\top \Theta_1 \mu_\tau + \left(\mu_\tau \hat{z}^\top + \hat{Z}\right)\Theta_2 \mu_\tau - \mu_\tau\right)\right\|^2\right].
$$

Lemma A.11 implies that $L(\Theta)$ is strongly-convex. Hence the global minima is its critical point. We write out the critical point condition $\nabla_{\Theta_1} L_2 = 0, \nabla_{\Theta_2} L_2 = 0, \nabla_{\Theta_3} L_1 = 0$ and $\nabla_{\Theta_4} L_1 = 0$

---

[5]Since we are only proving the existence of the solution, we do not need to solve $\alpha_2$ and $\beta_2$ out explicitly.

respectively:

$$\begin{cases} 2\left(d+4+\frac{d}{n}\right)\Theta_1 + 2\left(1+\frac{1}{n}\right)\Theta_2 + \left(\operatorname{tr}\left(\left(d+4+\frac{d}{n}\right)\Theta_1 + \left(1+\frac{1}{n}\right)\Theta_2\right) - 2 - d\right)\cdot I_d = 0 \\ 2\left(1+\frac{1}{n}\right)\Theta_1 + \left(1+\frac{2d+3}{n}\right)\Theta_2 + \left(-1 + \left(1+\frac{1}{n}\right)\operatorname{tr}\left(\Theta_1\right)\right)\cdot I_d = 0 \\ \left(1+\frac{1}{n}\right)\Theta_4 + \left(d+2+\frac{d}{n}\right)\Theta_3 - I_d = 0 \\ \left(1+\frac{1}{n}\right)\Theta_3 + \left(1+\frac{2d+1}{n}\right)\Theta_4 - I_d = 0. \end{cases} \quad (121)$$

Combining (121) with reparametrization formula and balancing condition in (122), we can solve out the optimal $u_{-1}, U_{11}, U_{12}, U_{21}, U_{22}$ in Theorem A.8.

$$\begin{cases} \Theta_1 = u_{-1}\operatorname{Sym}(U_{11} + U_{12} + U_{21} + U_{22}) \\ \Theta_2 = u_{-1}U_{21} + U_{22} \\ \Theta_3 = u_{-1}U_{12} + U_{22} \\ \Theta_4 = u_{-1}U_{22} \\ u_{-1}^2 = \|U_{11}\|_F^2 + \|U_{12}\|_F^2 + \|U_{21}\|_F^2 + \|U_{22}\|_F^2. \end{cases} \quad (122)$$

$\square$

# B ADDITIONAL EXPERIMENTS

In this section, we discuss some additional experiments.

## B.1 EXPERIMENTS FOR SEPARATE WEIGHTS MATRICES.

In this section, we report the results for separate $W^K, W^Q, W^P, W^V$ matrices.

We plot the loss curves for Transformers with separate weights matrices in Figure 4. Results are similar to those of merged-weights. We can see embedding with task descriptors have smaller loss values than embedding without descriptors for 1, 2, 3-layer Transformers.

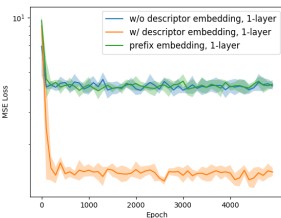 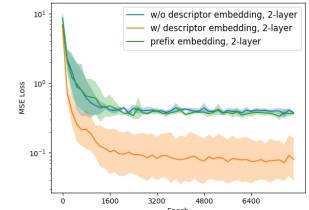 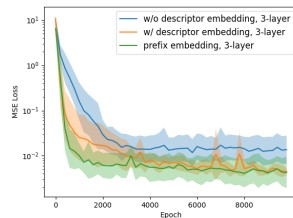

Figure 4: The MSE loss curves for Transformers with separate $W^K, W^Q, W^P, W^V$ matrices in different depths. We display mean and std of 5 random seeds. The number of samples in each sequence $n = 50$, data dimension $d = 5$ and $\Lambda = I_d$.

## B.2 EXPERIMENTS FOR HIGHER DIMENSION $d$ AND NON-SPHERICAL COVARIANCE MATRIX $\Lambda$.

In this section, we report some results on $d = 10$ and $\Lambda = \operatorname{diag}(0.2, 0.5, 1, 2, 5)$ respectively.

In Figure 5 we plot loss curves for Transformers trained on different embeddings with data dimension $d = 10$. In Figure 6 we plot loss curves for Transformers trained on different embeddings with covariance matrix $\Lambda = \operatorname{diag}(0.2, 0.5, 1, 2, 5)$. In both figures we can see that embedding with task descriptors have smaller loss values than embedding without descriptors for 1, 2, 3-layer Transformers.

To see how this performance gap will change across different dimension $d$, we further compare results on mean-varying linear regressions with different input dimensions $d = 5, 10, 15, 20$. In all experiments, Transformers trained with descriptors achieve lower MSE loss at convergence compared

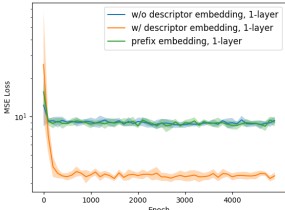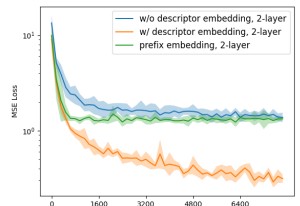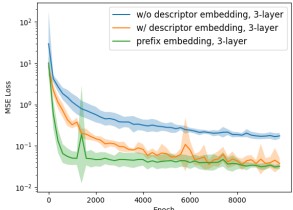

Figure 5: The MSE loss curves for Transformers in different depths. We display mean and std of 5 random seeds. The number of samples in each sequence $n = 50$, data dimension $d = 10$ and $\Lambda = I_d$.

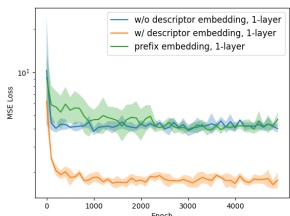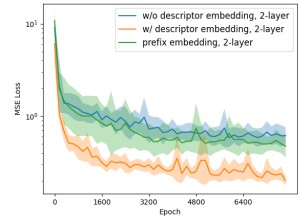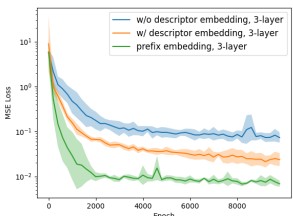

Figure 6: The MSE loss curves for Transformers in different depths. We display mean and std of 5 random seeds. The number of samples in each sequence $n = 50$, data dimension $d = 5$ and $\Lambda = \mathrm{diag}(0.2, 0.5, 1, 2, 5)$.

with those trained without descriptors, where we denote $L_{\text{descriptors}}$ and $L_{\text{no descriptors}}$ respectively. We report the performance gap $L_{\text{no descriptors}} - L_{\text{descriptors}}$ for LSA Transformers with different depths at Table 1. As we can see, as the ICL linear regression has higher dimension $d$, the performance gap widens between transformers trained with and without descriptors.

|         | $d = 5$ | $d = 10$ | $d = 15$ | $d = 20$ |
|---------|---------|----------|----------|----------|
| 1-layer | $2.725_{\pm 0.246}$ | $5.439_{\pm 0.361}$ | $7.857_{\pm 0.449}$ | $10.690_{\pm 0.742}$ |
| 2-layer | $0.473_{\pm 0.179}$ | $1.168_{\pm 0.098}$ | $3.548_{\pm 2.017}$ | $4.429_{\pm 1.657}$ |
| 3-layer | $0.045_{\pm 0.034}$ | $0.128_{\pm 0.030}$ | $0.184_{\pm 0.068}$ | $0.622_{\pm 0.447}$ |

Table 1: The performance gap $L_{\text{no descriptors}} - L_{\text{descriptors}}$ between Transformers trained with and without task descriptors. We report the mean and std of 5 random seeds.

### B.3 EXPERIMENTS FOR TWO-LAYER TRANSFORMERS TRAINED ON PREFIX EMBEDDING

In this section, we discuss some results on two-layer Transformers trained on prefix embedding.

We plot the attention matrices for a 2-layer Transformer trained on prefix embedding in Figure 7. One incomplete explanation for the results is that when a descriptor is added as a prefix, the Transformer initially uses one layer to attend to the descriptor and remove it from each in-context example. Subsequent layers can then simulate variants of GD to make predictions. As observed in the attention matrix, during the first layer, each in-context example attends to the descriptor in the prefix. However, there are additional attention patterns, such as the prefix attending to the in-context examples in the first layer, and the query token paying attention to the prefix token in the second layer during prediction. These attention dynamics may contribute to improved predictions in other ways. Notably, in Figure 2, the 2-layer Transformer trained on prefix embeddings outperforms[6] the 1-layer Transformer trained on embeddings with duplicated descriptors, suggesting that these additional attention patterns assist the Transformer in making more accurate predictions.

---

[6] 2-layer Transformers trained on prefix embedding converge to loss lower than $10^0$ in the center figure while 1-layer Transformers trained on $E_\tau$ converge to loss above $10^0$ in the left figure.

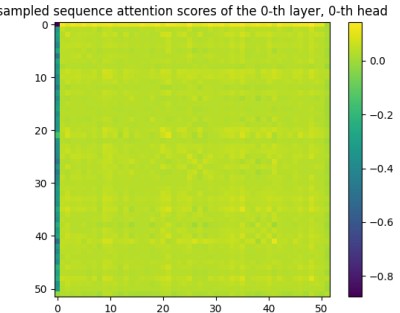 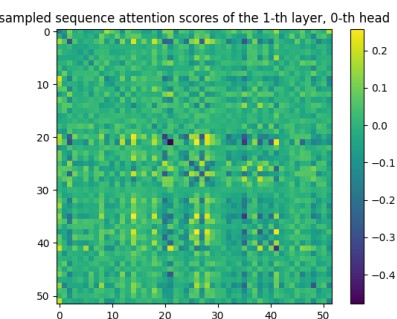

Figure 7: The attention matrices for a 2-layer Transformer trained on prefix embedding. The left one is the attention matrix of the 1-st layer and the right one is the attention matrix of the 2-nd layer.

## B.4 WEIGHTS PATTERNS FOR 3-LAYER TRANSFORMERS TRAINED ON $E_\tau$

In this section we report some patterns found in 3-layer Transformers trained on embedding with descriptors. We plot the heat maps of converged weights in Figure 8. We can roughly observe that they use descriptors to remove the mean from in-context examples and simulate GD++ (Von Oswald et al., 2023) as there are diagonals in $W^{PV}$ matrices at the first two layers.

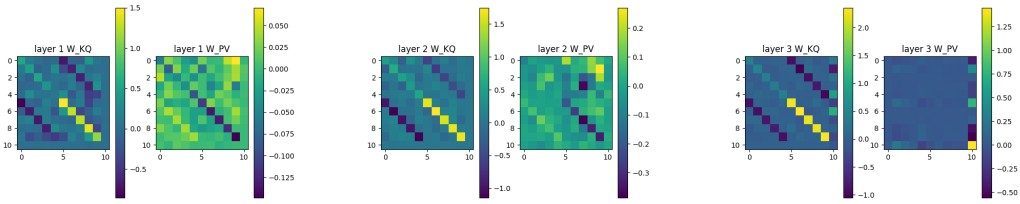

Figure 8: The heat maps of $W^{KQ}$ and $W^{PV}$ for a 3-layer Transformer trained on embedding with descriptors.

## B.5 EXPERIMENTS FOR ONE-HOT TASK DESCRIPTORS

We extend our experiments by introducing one-hot vectors as descriptors to indicate different distributions of $x$. We consider another mean-varying linear regression setting where there are $k$ spherical Gaussian distributions with different means $\mu_1, \ldots, \mu_k$. Each ICL linear regression task $\tau$ independently samples a $\mathcal{N}(\mu_i, I_d)$ from them as the input distribution of $x$. Instead of directly using the mean $\mu_i$ as the task descriptor, we assign an one-hot "task-identity" vector $e_i$ to each of the Gaussian distributions $\mathcal{N}(\mu_i, I_d)$ and use this vector as the task descriptor.

As a summary, there are two key modifications:

1. The mean $\mu_\tau$ for each task $\tau$ is uniformly drawn from a fixed set of $k$ vectors $\{\mu_1, \mu_2, \cdots, \mu_k\}$.

2. For each task $\tau$, if $\mu_\tau = \mu_j$ for some $1 \le j \le k$, we use the one-hot vector $e_j$ (1 at position $j$, 0 elsewhere) as the task descriptor. The corresponding input embedding matrix with task descriptors becomes

$$
E_\tau = \begin{pmatrix}
e_i & e_i & \ldots & e_i & e_i \\
x_{\tau,1} & x_{\tau,2} & \ldots & x_{\tau,n} & x_{\tau,\text{query}} \\
y_{\tau,1} & y_{\tau,2} & \ldots & y_{\tau,n} & 0
\end{pmatrix}.
$$

We can see from Figure 9 that one-hot descriptors also improve the ICL performance of trained Transformers.

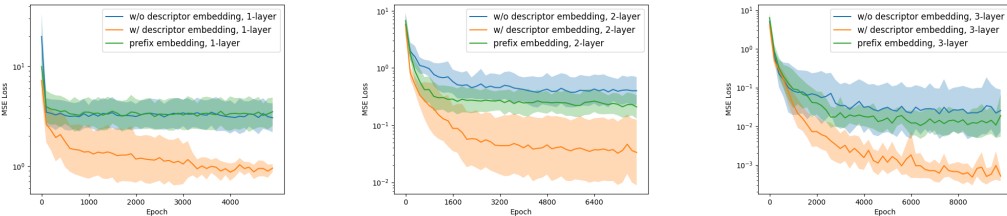

Figure 9: The MSE loss curves for Transformers in different depths with one-hot descriptors. We display mean and std of $5$ random seeds. The number of samples in each sequence $n = 50$, data dimension $d = 5$ and $\Lambda = I_d$.

