# OpenReview forum: "Task Descriptors Help Transformers Learn Linear Models In-Context"
_ICLR.cc/2025/Conference — ICLR 2025 Poster_

### Official Review · Reviewer_UsGc · 2024-10-27

**Soundness:** 3
**Presentation:** 3
**Contribution:** 3
**Rating:** 6
**Confidence:** 2

**Summary:**

This paper studies in-context learning with task descriptions. It uses a synthetic setup of linear regression, where the means from which the inputs are samples serve as the task description. Theory shows that a single-layer linear attention model can reach the optimal solution and the optimal solutions are characterized. Experiments show that a single-layer can indeed reach the optimal solution, but deeper models still perform better.

**Strengths:**

- Important research question on the effect of task descriptions in ICL
- Simple synthetic setup to study
- Theory showing optimality of a specific model class

**Weaknesses:**

- The biggest issue I have is whether the synthetic setup is a good proxy to study ICL, and specifically task descriptors. I understand similar synthetic setups were used to study ICL. But why is giving the mean a good task description? How does it simulate the natural language case, for example the one cited from Brown et al.?
- It would be helpful to explicitly highlight the insights drawn from the various lemmas and theorems throughout section 4. Unfortunately I had a hard time following this part and so I indicate this is a low-confidence review.
- The experiments are a nice edition to the theory, but I'm confused about the single-layer being inferior to the deeper models.
- See other comments and questions below.

**Questions:**

1. It's quite puzzling to understanding the embedding matrix formulation of the problem. I'm used to seeing in-context examples as running examples in text, which are made of words that are embedded. But here, the embedding matrix itself is the object to learn. More specifically, given this embedding matrix, the objective is to learn the bottom-right entry, which contains the prediction y_query. What are the trainable parameters? Is E itself updated during training? How does this formulation relate to the usual ICL setup of running text?

2. What are multiple tasks mentioned in equation 6 and does it make sense to take the expectation over all of them?  What is a task specifically?

3. What are the different sequences near equation 7? Are there multiple sequences per task? One sequence per task?

4. Can you motivate the initialization in page 4? any clearer motivation besides having two matrices have the same norm?

5. I am possibly missing some background to understand this, but what do you mean by "We run gradient flow on the population loss"? How is the optimization done exactly?

6. What is the significance of characterizing the optimal solutions in section 4? How should they be interpreted and what does it tell us about ICL with descriptions more broadly?

7. If a single linear layer transformer can reach the optimal solution, then why do the experimental results show deeper models to perform better? Is it because of training difficulties with the single layer case? Is it just a sample complexity issue, with the single-layer model not having enough samples?

8. Since there's no pre-training and fine-tuning going on here, it seems like all instances of "pretraining" could just be changed to "training".

---

> ### Author Response · Authors · 2024-11-21
> **Response to Reviewer UsGc (part 1)**
>
> Thank you for the thoughtful comments.
>
> > **Weakness 1: The biggest issue I have is whether the synthetic setup is a good proxy to study ICL, and specifically task descriptors. I understand similar synthetic setups were used to study ICL. But why is giving the mean a good task description? How does it simulate the natural language case, for example the one cited from Brown et al.?**
>
> In the example we cited from Brown et al., the description “English” implies x is drawn from the distribution of English words. Imagine the model is dealing with multilingual word translation and we can view each language as a distribution of words. Then the descriptors (like “English”) should imply which distribution x is drawn from. In this case, any vector that distinguishes the input distribution from other distributions could be considered as task descriptors and the mean $\mu_\tau$ is a natural choice of such descriptors in our setup. We also add some empirical results where we use one-hot vectors as descriptors to indicate different distributions of $x$. Please see 4. in our general response. In this experiment, the descriptors also improve the performance of ICL and we believe its theoretical analysis would not differ too much from current analysis.
>
>
> > **Weakness 2: It would be helpful to explicitly highlight the insights drawn from the various lemmas and theorems throughout section 4.**
>
> Sorry for making you confused. Theorem 4.1 gives a global convergence result for finite $n$ and also describes what the global minimizer looks like – its $W^{KQ}$ has two additional non-zero blocks (11 and 21 blocks) compared with $n=\infty$. We then try to explain why this transition appears by decomposing the loss into bias and variance. The subsequent simplifications and reparametrizations of loss shows that (i) transformers do not need these two blocks when $n=\infty$ since the 12 and 22 blocks are sufficient to achieve zero loss and (ii) when $n$ is finite, due to non-asymptotic $\hat z$ and $\hat Z$, transformers need to use all four blocks to minimize both bias and variance. We have added some similar discussions to Section 4 in the revision. The Lemma 4.2 is actually a bit orthogonal to the above discussions. We put it in Section 4 because it is used to prove the optimality of the proposed solution when $n$ is finite and it is unique to finite $n$ —- we don’t have a corresponding lemma for $n=\infty$ since it is easy to prove optimality with infinite samples. We have moved it to the appendix in the revision to make Section 4 more readable.
>
> > **Weakness 3: The experiments are a nice edition to the theory, but I'm confused about the single-layer being inferior to the deeper models.**
>
> For infinite samples single-layer models are optimal since one step of GD on infinite samples achieves zero loss. However, in experiments the number of samples is finite. In this case, one step of GD does not achieve zero loss and more steps of GD will give you a lower loss. More layers offer more capacity for transformers to simulate more complicated algorithms, for example, multiple steps of (variants of) GD.

---

> > ### Comment · Reviewer_UsGc · 2024-11-21
> > **Task description?**
> >
> > Thanks for your responses. The main issue that's still troubling me is how reasonable the setup is in simulating the ICL with task descriptors in natural language tasks. This is weakness 1 in my review and weakness 2 in yqsE review. I understand your answer that even in natural language we may have "distribution cues" like "English" signaling that inputs come from the English distribution. However, task descriptions in language models are more than that. The paper gives the example from Brown et al. of "Please translate English into French:" and as Reviewer yqsE says, "task descriptions in practical ICL applications typically convey semantic instructions".
> >
> > Given this, I think the paper is over-claiming in terms of title and general pitch and promises. I would have been more accepting if the paper were proposing to study how descriptions of the input distribution affect in-context learning. Then the simple synthetic setup would have been more faithful.
> >
> > Re weakness 3: Thanks. it would be good to add this clarification to the paper.

---

> > > ### Author Response · Authors · 2024-11-22
> > > **Edited the abstract and intro to avoid overclaiming**
> > >
> > > We appreciate your feedback. It was not our intention to overclaim. We edited the abstract to talk about the nature of the descriptor. We also added **"While the prefix before in-context examples could contain various information, this work focuses on descriptions containing *distributional* information about inputs. In this paper, we investigate whether such task descriptors can help in-context learning for linear models"** to main text to clarify. We welcome any additional suggestions if this revision does not fully address your concerns.

---

> > > > ### Comment · Reviewer_UsGc · 2024-11-22
> > > > **Thank you!**
> > > >
> > > > This is helpful, thank you!
> > > > If you have space, I suggest trying to motivate the initialization in the revision.

---

> ### Author Response · Authors · 2024-11-21
> **Response to Reviewer UsGc (part 2)**
>
> > **Question 1: It's quite puzzling to understanding the embedding matrix formulation of the problem..... What are the trainable parameters? Is E itself updated during training? How does this formulation relate to the usual ICL setup of running text?**
>
> $E$ is not trainable since the only trainable parameters are the weights matrices $W^{KQ}$ and $W^{PV}$. In the usual running text setup, $E$ embeds each token in $(x_1, y_1, x_2, y_2, \dots, x_n, y_n, x_{query})$. In our setup, we concatenate each pair of $(x_i, y_i)$ into a single token. For the query pair, we zero out the $y_{query}$ entry to obtain $(x_{query}, 0)$ where $0$ is waiting to be filled with the prediction. Therefore, the prediction can be read out from that entry after feeding $E$ to the transformers, which is the bottom-right entry of $f_{LSA}(E)$. Such input formats are used in other works such as [1, 2, 3, 4, 5].
>
> [1] Johannes Von Oswald, Eyvind Niklasson, Ettore Randazzo, Jo&atilde;o Sacramento, Alexander Mordvintsev, Andrey Zhmoginov, and Max Vladymyrov. Transformers learn in-context by gradient descent. ICML 2023.
>
> [2] Ekin Aky&uuml;rek, Dale Schuurmans, Jacob Andreas, Tengyu Ma, and Denny Zhou. What learning
> algorithm is in-context learning? investigations with linear models. ICLR 2023.
>
> [3] Kwangjun Ahn, Xiang Cheng, Hadi Daneshmand, and Suvrit Sra. Transformers learn to implement preconditioned gradient descent for in-context learning. NeurIPS 2023.
>
> [4] Arvind Mahankali, Tatsunori B. Hashimoto, and Tengyu Ma. One step of gradient descent is provably the optimal in-context learner with one layer of linear self-attention. ICLR 2024.
>
> [5] Ruiqi Zhang, Spencer Frei, and Peter L Bartlett. Trained transformers learn linear models in-context. JMLR 2024.
>
> > **Question 2: What are multiple tasks mentioned in equation 6 and does it make sense to take the expectation over all of them? What is a task specifically?**
>
> An in-context task $\tau$ is a linear regression task with input $x_{\tau, i}, y_{\tau,i}=w_{\tau}^\top x_{\tau, i}$ and a query $x_{\tau, query}$, where $x_{\tau, i}$ and $x_{\tau, query}$ are drawn from $\mathcal{N}(\mu_\tau, I)$. Different tasks have different $w_{\tau}$ and different mean $\mu_\tau$. We take expectation over all tasks because we are using the population loss, which assumes transformers are trained on infinite tasks drawn randomly. In experiments we sample $m=4096$ tasks and use the empirical loss.
>
> > **Question 3: What are the different sequences near equation 7? Are there multiple sequences per task? One sequence per task?**
>
> One sequence $(\mu_\tau, x_{\tau, 1},w_\tau^\top x_{\tau, 1}, \dots,x_{\tau, n},w_\tau^\top x_{\tau, n}, x_{\tau, query})$ per task $\tau$.
>
> >**Question 4: Can you motivate the initialization in page 4? any clearer motivation besides having two matrices have the same norm?**
>
> Sure. If we scale $W^{KQ}$ by factor $\kappa$ and $W^{PV}$ by factor $1/\kappa$, the output of the transformer does not change. Hence there are infinite optimal solutions since you can always scale a solution in this way without changing its loss. Infinite optimal solutions are actually troublesome for the convergence proof because we do not know which optimal solution will the gradient flow converge to. The balanced initialization addresses this issue. As we can see in Lemma 3.3, if the initializations of $W^{KQ}$ and $W^{PV}$ are balanced, they will keep balanced throughout the training. This invariance helps us exclude all optimal solutions that are not balanced. Actually the first 3 invariances in Lemma 3.3 together exclude all optimal solutions except the one in Theorem 3.1, which fully addresses our issue of non-unique optimal solutions.

---

> > ### Comment · Reviewer_UsGc · 2024-11-21
> >
> > Thank you. It would be good to add the clarifications to questions 2+3 to the paper and the motivation for initialization (question 4).

---

> ### Author Response · Authors · 2024-11-21
> **Response to Reviewer UsGc (part 3)**
>
> >**Question 5: I am possibly missing some background to understand this, but what do you mean by "We run gradient flow on the population loss"? How is the optimization done exactly?**
>
> In the theoretical analysis, the optimization is done by running gradient flow on the population loss (6), which is indeed the gradient descent with infinitesimal step size. In experiments, we use Adam optimizer.
>
> > **Question 6: What is the significance of characterizing the optimal solutions in section 4? How should they be interpreted and what does it tell us about ICL with descriptions more broadly?**
>
> The characterization of the optimal solution in section 4 implies that if the number of samples is finite, transformers learned a more delicate way to use the descriptors in not only keys but also queries to minimize both the bias and variance. This is different from the infinite-sample case in section 3 where transformers only use the descriptors in keys. This implies that the way transformers leverage the information in descriptions could depend on the number of in-context examples, and when you have fewer in-context examples, transformers might leverage descriptors more.
>
> > **Question 7: If a single linear layer transformer can reach the optimal solution, then why do the experimental results show deeper models to perform better? Is it because of training difficulties with the single layer case? Is it just a sample complexity issue, with the single-layer model not having enough samples?**
>
> Yes, if we have infinite in-context examples, the single-layer linear transformer would achieve zero loss and thus be optimal. Since we only have finite $n$ in experiments, more layers enable transformers to perform multiple steps of (variants of) GD to achieve lower loss. It is also worth noting that the global minimizer in our Theorem 4.1 means that transformers converge to the solution minimizing the population loss (6), where the model is constrained to be single-layer.
>
> >**Question 8: Since there's no pre-training and fine-tuning going on here, it seems like all instances of "pretraining" could just be changed to "training".**
>
> Yes, it wouldn’t hurt if changing from pretraining to training. We have modified them in the revision.

---

> > ### Comment · Reviewer_UsGc · 2024-11-21
> >
> > Thank you. The clarification for question 6 is very helpful and I'd suggest adding it to the paper.

---

### Official Review · Reviewer_yqsE · 2024-10-28

**Soundness:** 2
**Presentation:** 3
**Contribution:** 2
**Rating:** 5
**Confidence:** 4

**Summary:**

This paper presents a theory analysis of the task descriptor usage in in-context learning (ICL)  for Transformer models.
The analysis is conducted based on a simple linear regression problem (i.e., given $x$ drawn from Gaussian $(\mu, \mathcal{I})$ predict $y$  = $w^Tx$ where $w$ is the underlying generative weights), and the task descriptors here are the $\mu$ for the input distribution.

Theoretically, the analysis results show that the provided task descriptors could help the Transformer directly use the $\mu$ provided in specialized input embedding given infinite samples $n$. Furthermore, the authors show that the model guaranteed to converge to the global optima from a a reasonable initialization. Empirical results show that the task descriptor could help a 1-layer Transformer achieve significantly lower loss, and the findings generalize to multi-layer Transformer model as well.

**Strengths:**

- Overall solid analysis (theory and empirical ) for the task descriptor usage in ICL of Transformer models.

**Weaknesses:**

- The analysis and results are overly narrow, focusing solely on linear regression. This limited scope makes it difficult to generalize the findings to fundamental tasks like sentiment classification with in-context learning (ICL).

- The paper's restricted scope can be partially attributed to defining the task descriptor as the mean $\mu$. This is a non-standard approach, as task descriptions in practical ICL applications typically convey semantic instructions (e.g., `Classify the review as positive or negative.`) rather than characterizing the input distribution $x$ (e.g., the lexical and syntactic patterns of review text).

- (Minor) Figure 2 reveals that the performance gap between models with and without task descriptors narrows as model complexity increases (from 1-layer to 2-layer architectures). This raises questions about the utility of task descriptors in modern large-scale models. The diminishing effect of task descriptors with increasing model size may limit the long-term relevance of this analysis, particularly given the trend toward ever-larger models.

**Questions:**

- Would the loss curve comparison change with larger models and complex data samples (higher dimension than 5)? 12-Layer Transformer such as GPT-2?

---

> ### Author Response · Authors · 2024-11-21
> **Response to Reviewer yqsE**
>
> Thanks for your thoughtful comments.
>
> > **Weakness 1: The analysis and results are overly narrow, focusing solely on linear regression. This limited scope makes it difficult to generalize the findings to fundamental tasks like sentiment classification with in-context learning (ICL).**
>
> Simple settings such as ICL of linear regressions are important for theory researchers to start with and can still provide valuable insights. There is a line of theoretical research ([1, 2, 3, 4, 5, 6]) on linear regressions which sheds light on understanding the mechanism of ICL.
>
> [1] Johannes Von Oswald, Eyvind Niklasson, Ettore Randazzo, Jo&atilde;o Sacramento, Alexander Mordvintsev, Andrey Zhmoginov, and Max Vladymyrov. Transformers learn in-context by gradient descent. ICML 2023.
>
> [2] Ekin Aky&uuml;rek, Dale Schuurmans, Jacob Andreas, Tengyu Ma, and Denny Zhou. What learning
> algorithm is in-context learning? investigations with linear models. ICLR 2023.
>
> [3] Kwangjun Ahn, Xiang Cheng, Hadi Daneshmand, and Suvrit Sra. Transformers learn to implement preconditioned gradient descent for in-context learning. NeurIPS 2023.
>
> [4] Arvind Mahankali, Tatsunori B. Hashimoto, and Tengyu Ma. One step of gradient descent is provably the optimal in-context learner with one layer of linear self-attention. ICLR 2024.
>
> [5] Ruiqi Zhang, Spencer Frei, and Peter L Bartlett. Trained transformers learn linear models in-context. JMLR 2024.
>
>
>
> > **Weakness 2: The paper's restricted scope can be partially attributed to defining the task descriptor as the mean μ. This is a non-standard approach, as task descriptions in practical ICL applications typically convey semantic instructions (e.g., Classify the review as positive or negative.) rather than characterizing the input distribution x (e.g., the lexical and syntactic patterns of review text).**
>
> We understand that the prefix before in-context examples could contain various information – some of them imply knowledge of the input distribution while some of them don’t. In this paper, we want to focus on the former, which motivates the setup of our paper. Actually your example prefix also contains knowledge about the input data – “review” implies that input x is drawn from the distribution of reviews. The example used in our paper is the word translation example from Brown et al., where “English” implies that x is drawn from the distribution of English vocabulary.
>
> > **Weakness 3: Figure 2 reveals that the performance gap between models with and without task descriptors narrows as model complexity increases (from 1-layer to 2-layer architectures). This raises questions about the utility of task descriptors in modern large-scale models. The diminishing effect of task descriptors with increasing model size may limit the long-term relevance of this analysis, particularly given the trend toward ever-larger models.**
>
> Thanks for pointing it out! We believe the performance gap between models with and without descriptors depends on multiple factors – the size of the model, the data dimension $d$, etc. Though it is true that the gap decreases as the depth of models grows, it could increase again as the data dimension d increases. We have added this comparison in our revision. Please see 3. in our general response.
>
> > **Question 1: Would the loss curve comparison change with larger models and complex data samples (higher dimension than 5)? 12-Layer Transformer such as GPT-2?**
>
> Yes, the performance gap could grow as the dimension $d$ grows. Our experiments are still within the scope of ICL of linear regressions and are mostly to verify our theory. For simple linear regressions, we feel it is not necessary to use more complex transformers such as GPT-2 in experiments.

---

> > ### Comment · Reviewer_yqsE · 2024-11-21
> >
> > Thank you for your response. The revision makes the scope clearer and more focused. Although I still found the scope is rather limited (especially from an empirical perspective),  I understand the value of the investigated setup better after reading the rebuttals and checking other reviews. I will increase my Score to 5 accordingly.

---

> > > ### Author Response · Authors · 2024-11-22
> > >
> > > Thank you for raising your score. We appreciate your constructive feedback which helped improve our paper.

---

### Official Review · Reviewer_WNXG · 2024-10-31

**Soundness:** 3
**Presentation:** 3
**Contribution:** 2
**Rating:** 6
**Confidence:** 3

**Summary:**

This paper proposes a new mathematical setting for theoretical in-context learning analysis. It propose a setting which models the task description in in-context learning. The author discussed the optimal soluations for a 1-layer linear self-attention network in two situations, where samples are sufficient or limited, and prove that such optimal soluations can be achieved under gradient flow training with a reasonable initialization. Also, empirical verification for 1-layer transformers are also provided.

**Strengths:**

1.	The author models the task description part in in-context learning and provide comprehensive analysis.
2.	Both limited-sample and infinite-sample situations are considered.
3.	Experiments further strengthened their analysis.

**Weaknesses:**

1.	The conclusions based on a 1-layer linear self-attention model and gradient-flow might be a little weak.
2.	Though introducing task description is very interesting, the techniques used and the theorems got might be a little incremental.

**Questions:**

1.	Might the author provide some insights/comparisons about what real difference the task description brings for ICL? It would be very helpful.
2.	Proofs of more complex situation than gradient flow / 1-layer LSA would be very helpful, especially for cases where global optimal can not achieved (which I guess is more often in reality?)

---

> ### Author Response · Authors · 2024-11-21
> **Response to Reviewer WNXG**
>
> Thanks for your thoughtful comments.
>
> > **Weakness 1: The conclusions based on a 1-layer linear self-attention model and gradient-flow might be a little weak.**
>
> We agree the model we study is simple and stylized. The rigorous theoretical analysis we did in this paper is already highly nontrivial for this setting. Existing theoretical analysis for in-context learning/transformers have mostly focused on similar linear regression settings and other shallow models.
>
> > **Weakness 2: Though introducing task description is very interesting, the techniques used and the theorems got might be a little incremental**
>
> The high level structure of our analysis is indeed similar to previous papers (especially [1]). However, we want to highlight some challenges that task descriptors and the varying-mean introduce. For example, we need to deal with the symmetry of the 11 block of $W^{KQ}$ (please see line 318 for details) which is unique to task descriptors; For the finite sample setting, a priori it is even unclear what the optimal solution should be, and we rely on a new decomposition of loss functions into two $L_1+L_2$ and analyze them separately (note that the bias term $L_2$ vanishes when $\mu_\tau$ is fixed to be zero as in previous works).
>
> [1] Ruiqi Zhang, Spencer Frei, and Peter L Bartlett. Trained transformers learn linear models in-context. JMLR 2024.
>
> > **Question 1: Might the author provide some insights/comparisons about what real difference the task description brings for ICL? It would be very helpful.**
>
> The differences that task descriptors bring in general scenarios may depend on what the ICL task and the descriptors are. For our mean-varying linear regression setting, please see 1. in our general response.
>
> > **Question 2: Proofs of more complex situation than gradient flow / 1-layer LSA would be very helpful, especially for cases where global optimal can not achieved (which I guess is more often in reality?)**
>
> Note that in the finite sample setting, even though we achieve the global optimal solution, the optimal solution does not have a zero loss. We agree in more complicated settings it might even be difficult to achieve the global optimal solution, that would be a great direction for future work.

---

### Official Review · Reviewer_13Fd · 2024-11-03

**Soundness:** 3
**Presentation:** 2
**Contribution:** 3
**Rating:** 8
**Confidence:** 2

**Summary:**

The paper explores the role of task descriptors in enhancing the in-context learning (ICL) capabilities of Transformers. The authors show that incorporating task descriptors into the input prompt leads to improved ICL performance for linear models. They present theoretical findings indicating that gradient flow converges to a global minimum for simple linear models when task descriptors are used, and they empirically verify these results by demonstrating weight convergence to the predicted global minimum. The paper includes a theoretical analysis of how task descriptors facilitate in-context learning in linear regression settings, illustrates how Transformers can leverage these descriptors to enhance performance, and provides empirical validation of the theoretical findings through experiments with various Transformer configurations.

**Strengths:**

- The paper offers a novel perspective on how task descriptors can improve the ICL performance of Transformers.
- The paper presents a theoretical analysis of how task descriptors enhance in-context learning in linear regression settings, showing how Transformers can utilize these descriptors to boost performance.
- It also offers empirical validation of the theoretical findings through experiments conducted with different Transformer configurations.

**Weaknesses:**

- While the paper provides a theoretical analysis, it may lack a deeper exploration into the underlying mechanisms of how Transformers integrate task descriptors during the learning process.
- The experiments, although comprehensive, are limited to a specific setup. More diverse datasets and problem settings could provide a more robust evaluation.

**Questions:**

- What are the specific mechanisms through which task descriptors improve ICL in Transformers? Further analysis or experiments that shed light on this process would be valuable.

---

> ### Author Response · Authors · 2024-11-21
> **Response to Reviewer 13Fd**
>
> We thank the reviewer for the thoughtful comments.
>
> > **Weakness 1: While the paper provides a theoretical analysis, it may lack a deeper exploration into the underlying mechanisms of how Transformers integrate task descriptors during the learning process. Question 1: What are the specific mechanisms through which task descriptors improve ICL in Transformers? Further analysis or experiments that shed light on this process would be valuable.**
>
> The way that transformers use task descriptors in general scenarios may depend on what the ICL task and the descriptors are. For our mean-varying linear regression setting, please see 1. in our general response.
>
> > **Weakness 2: The experiments, although comprehensive, are limited to a specific setup. More diverse datasets and problem settings could provide a more robust evaluation.**
>
> Thanks for pointing it out. The experiments are mostly verification of our theory. Experiments for more general setups and more practical settings are interesting future directions. For the verification, we add new experiments varying the data dimension $d$. We also extend our experiments by using one-hot task descriptors. Please see 3. and 4. in our general response.

---

### Author Response · Authors · 2024-11-21
**Global Response**

We sincerely thank all reviewers for constructive comments. We revised our paper according to the issues and concerns that reviewers proposed. Here is a summary of the updates.

1.  We add descriptions of how transformers use task descriptors in our setting.
* For the setting with infinite in-context examples, the task descriptors enable transformers to do data standardization (subtract the mean from each $x_i$). Specifically,  the optimal solution given in Theorem 3.1 subtracts the mean $\mu_\tau$ from the $x_i$ in key values and do one step of preconditioned GD to output $$\begin{aligned}
    \hat y_{query}
&=x_{query}^{\top} \Lambda^{-1} \frac{1}{n} \sum_{i=1}^n  (x_i-\mu_\tau) y_i\\\\
&= x_{query}^{\top} \Lambda^{-1} \left(\frac{1}{n} \sum_{i=1}^n (x_i-\mu_\tau)x_i^\top\right) w \\\\
&\to x_{query}^{\top} w \text{ ~~as } n\to\infty,
\end{aligned}$$ which is correct. While without descriptors, one can only get $$\begin{aligned}
    \hat y_{query}
&=x_{query}^{\top} A \frac{1}{n} \sum_{i=1}^n  x_i y_i\\\\
&= x_{query}^{\top}A \left(\frac{1}{n} \sum_{i=1}^n x_i x_i^\top\right) w \\\\
&\to x_{query}^{\top} A (\Lambda + \mu_\tau\mu_\tau^\top)w \text{ ~~as } n\to\infty,
\end{aligned}$$ which would have a non-zero loss for any preconditioner $A$ in the presence of an unfixed $\mu_\tau$.
* For the setting with finite in-context examples, transformers learned a more delicate way to use the descriptors in not only keys but also queries to minimize both the bias and variance. This implies that the transformers trained on fewer in-context examples could leverage descriptors more.

2. We also emphasize in the intro that in the more general setting the transformer may have many other potential ways of leveraging the task descriptor, this paper only gives one example.

3. We add a new experiment varying the data dimension $d$. The experiment shows that as the ICL linear regression has higher dimension $d$, the performance gap widens between transformers trained with and without descriptors. Please see table 1 in Section B.2 at appendix for details.

4. We extend our experiments by introducing one-hot vectors as descriptors to indicate different distributions of $x$. We consider another mean-varying linear regression setting where there are $k$ spherical Gaussian distributions with different means $\mu_1,\dots,\mu_k$. Each ICL linear regression task $\tau$ independently samples a $\mathcal{N}(\mu_i, I_d)$ from them as the input distribution of $x$. Instead of directly using the mean $\mu_i$ as the task descriptor, we assign an one-hot ``task-identity'' vector $e_i$ to each of the Gaussian distributions $\mathcal{N}(\mu_i, I_d)$ and use this task identity vector as the task descriptor.
As a summary, there are two key modifications:
- The mean $\mu_\tau$ for each task $\tau$ is uniformly drawn from a fixed set of $k$ vectors ${\mu_1, \mu_2, \cdots, \mu_k}$.
- When $\mu_\tau = \mu_i$ for some $1 \leq i\leq k$, we use the one-hot vector $e_i$ (1 at position $i$, 0 elsewhere) as the task descriptor.
 The corresponding input embedding matrix becomes $$E_\tau=\left(\begin{array}{ccccc} e_i & e_i & \dots & e_i& e_i\\\\x_{\tau, 1} & x_{\tau, 2} & \dots & x_{\tau, n} & x_{\tau, \text { query }} \\\\y_{\tau,1} & y_{\tau,2} & \dots & y_{\tau,n} & 0\end{array}\right).$$
In this experiment, the descriptors also improve the performance of ICL. Please see Section B.5 at the appendix for details.

5. We move Lemma 4.2 to the appendix as it is a bit orthogonal to the main discussion in Section 4.

---

### Meta-Review · Area_Chair_fvCj · 2024-12-20

**Metareview:**

The authors study the linear attention, in-context learning dynamics problem, and ask whether task descriptions affect the gradient-descent like learning dynamics in these papers. The paper presents nice theoretical analysis of these linear ICL dynamics and also presents some simple empirical experiments to back this up. The reviewers are pretty consistent on both the strengths and the weaknesses, with the strength being the theory and toy experiments, and the weakness being the very limited setting of linear attention ICL.

The weakness is generally minor - there's been fairly extensive (both empirical and theory) interest in understanding the linear ICL setting, and the paper seems to be a solid contribution to this area.

**Additional Comments On Reviewer Discussion:**

There were some nice clarifications from the authors about the reviewer questions. The main issue (limited scope) is true but not that serious of an issue, and so the rebuttal was not critical to the final decision.

---

### Decision · Program_Chairs · 2025-01-22

Accept (Poster)